# PERMUTATION COMPRESSORS FOR PROVABLY FASTER DISTRIBUTED NONCONVEX OPTIMIZATION

**Rafał Szlendak**[*]
KAUST
Saudi Arabia

**Alexander Tyurin**
KAUST
Saudi Arabia

**Peter Richtárik**
KAUST
Saudi Arabia

## ABSTRACT

We study the MARINA method of Gorbunov et al. (2021) – the current state-of-the-art distributed non-convex optimization method in terms of theoretical communication complexity. Theoretical superiority of this method can be largely attributed to two sources: the use of a carefully engineered biased stochastic gradient estimator, which leads to a reduction in the number of communication rounds, and the reliance on *independent* stochastic communication compression operators, which leads to a reduction in the number of transmitted bits within each communication round. In this paper we i) extend the theory of MARINA to support a much wider class of potentially *correlated* compressors, extending the reach of the method beyond the classical independent compressors setting, ii) show that a new quantity, for which we coin the name *Hessian variance*, allows us to significantly refine the original analysis of MARINA without any additional assumptions, and iii) identify a special class of correlated compressors based on the idea of *random permutations*, for which we coin the term Perm$K$. The use of it leads to $O(\sqrt{n})$ (resp. $O(1 + d/\sqrt{n})$) improvement in the theoretical communication complexity of MARINA in the low Hessian variance regime when $d \geq n$ (resp. $d \leq n$), where $n$ is the number of workers and $d$ is the number of parameters describing the model we are learning. We corroborate our theoretical results with carefully engineered synthetic experiments with minimizing the average of nonconvex quadratics, and on autoencoder training with the MNIST dataset.

## 1 INTRODUCTION

The practice of modern supervised learning relies on highly sophisticated, high dimensional and data hungry deep neural network models (Vaswani et al., 2017; Brown et al., 2020) which need to be trained on specialized hardware providing fast distributed and parallel processing. Training of such models is typically performed using elaborate systems relying on specialized distributed stochastic gradient methods (Gorbunov et al., 2021). In distributed learning, communication among the compute nodes is typically a key bottleneck of the training system, and for this reason it is necessary to employ strategies alleviating the communication burden.

### 1.1 THE PROBLEM AND ASSUMPTIONS

Motivated by the need to design provably communication efficient *distributed stochastic gradient methods* in the *nonconvex* regime, in this paper we consider the optimization problem

$$\min_{x \in \mathbb{R}^d} \left[ f(x) := \frac{1}{n} \sum_{i=1}^{n} f_i(x) \right], \tag{1}$$

where $n$ is the number of workers/machines/nodes/devices working in parallel, and $f_i : \mathbb{R}^d \to \mathbb{R}$ is a (potentially *nonconvex*) function representing the loss of the model parameterized by weights $x \in \mathbb{R}^d$ on training data stored on machine $i$.

---

[*]The work of Rafał Szlendak was performed during a Summer research internship in the *Optimization and Machine Learning Lab* at KAUST led by Peter Richtárik. Rafał Szlendak is an undergraduate student at the University of Warwick, United Kingdom.

While we do *not* assume the functions $\{f_i\}$ to be convex, we rely on their differentiability, and on the well-posedness of problem (1):

**Assumption 1.** *The functions $f_1, \ldots, f_n : \mathbb{R}^d \to \mathbb{R}$ are differentiable. Moreover, $f$ is lower bounded, i.e., there exists $f^{\text{inf}} \in \mathbb{R}$ such that $f(x) \geq f^{\text{inf}}$ for all $x \in \mathbb{R}^d$.*

We are interested in finding an approximately stationary point of the nonconvex problem (1). That is, we wish to identify a (random) vector $\hat{x} \in \mathbb{R}^d$ such that

$$\mathrm{E}\left[\|\nabla f(\hat{x})\|^2\right] \leq \varepsilon \tag{2}$$

while ensuring that the volume of communication between the $n$ workers and the server is as small as possible. Without the lower boundedness assumption there might not be a point with a small gradient (e.g., think of $f$ being linear), which would render problem (2) unsolvable. However, lower boundedness ensures that the problem is well posed. Besides Assumption 1, we rely on the following smoothness assumption:

**Assumption 2.** *There exists a constant $L_+ > 0$ such that $\frac{1}{n}\sum_{i=1}^n \|\nabla f_i(x) - \nabla f_i(y)\|^2 \leq L_+^2 \|x - y\|^2$ for all $x, y \in \mathbb{R}^d$. To avoid ambiguity, let $L_+$ be the smallest such number.*

While this is a somewhat stronger assumption than mere $L_-$-Lipschitz continuity of the gradient of $f$ (the latter follows from the former by Jensen's inequality and we have $L_- \leq L_+$), it is weaker than $L_i$-Lipschitz continuity of the gradient of the functions $f_i$ (the former follows from the latter with $L_+^2 \leq \frac{1}{n}\sum_i L_i^2$). So, this is still a reasonably weak assumption.

## 1.2 A BRIEF OVERVIEW OF THE STATE OF THE ART

To the best of our knowledge, the state-of-the-art distributed method for finding a point $\hat{x}$ satisfying (2) for the nonconvex problem (1) in terms of the *theoretical communication complexity*[1] is the MA-RINA method of Gorbunov et al. (2021) (see Algorithm 1). MARINA relies on *worker-to-server communication compression*, and its power resides in the construction of a carefully designed sequence of *biased* gradient estimators which help the method obtain its superior communication complexity.

---

**Algorithm 1** MARINA

---
1: **Input:** starting point $x^0$, stepsize $\gamma$, probability $p \in (0, 1]$, number of iterations $T$
2: Initialize $g^0 = \nabla f(x^0)$
3: **for** $k = 0, 1, \ldots, T - 1$ **do**
4:     Sample $\theta_t \sim \mathrm{Be}(p)$
5:     Broadcast $g^t$ to all workers
6:     **for** $i = 1, \ldots, n$ in parallel **do**
7:         $x^{t+1} = x^t - \gamma g^t$
8:         Set $g_i^{t+1} = \nabla f_i(x^{t+1})$ if $\theta_t = 1$, and $g_i^{t+1} = g^t + \mathcal{C}_i\left(\nabla f_i(x^{t+1}) - \nabla f_i(x^t)\right)$ otherwise
9:     **end for**
10:    $g^{t+1} = \frac{1}{n}\sum_{i=1}^n g_i^{t+1}$
11: **end for**
12: **Output:** $\hat{x}^T$ chosen uniformly at random from $\{x^t\}_{k=0}^{T-1}$

---

The method uses *randomized compression operators* $\mathcal{C}_i : \mathbb{R}^d \to \mathbb{R}^d$ to compress messages (gradient differences) at the workers $i \in \{1, 2, \ldots, n\}$ before they are communicated to the server. It is assumed that these operators are unbiased, i.e., $\mathrm{E}[\mathcal{C}_i(a)] = a$ for all $a \in \mathbb{R}^d$, and that their variance is bounded as

$$\mathrm{E}\left[\|\mathcal{C}_i(a) - a\|^2\right] \leq \omega \|a\|^2$$

for all $a \in \mathbb{R}^d$ and some $\omega \geq 0$. For convenience, let $\mathbb{U}(\omega)$ be the class of such compressors. A key assumption in the analysis of MARINA is the *independence* of the compressors $\{\mathcal{C}_i\}_{i=1}^n$.

---
[1] For the purposes of this paper, by *communication complexity* we mean the product of the number of communication rounds sufficient to find $\hat{x}$ satisfying (2), and a suitably defined measure of the volume of communication performed in each round. As standard in the literature, we assume that the workers-to-server communication is the key bottleneck, and hence we do not count server-to-worker communication. For more details about this highly adopted and studied setup, see Appendix F.

In particular, MARINA solves the problem (1)–(2) in

$$T = \frac{2\Delta^0}{\varepsilon}\left(L_- + L_+\sqrt{\frac{1-p}{p}\frac{\omega}{n}}\right)$$

communication rounds[2], where $\Delta^0 := f(x^0) - f^{\text{inf}}$, $x^0 \in \mathbb{R}^d$ is the initial iterate, $p \in (0,1]$ is a parameter defining the probability with which full gradients of the local functions $\{f_i\}$ are communicated to the server, $L_- > 0$ is the Lipschitz constant of the gradient of $f$, and $L_+ \geq L_-$ is a certain smoothness constant associated with the functions $\{f_i\}$.

In each iteration of MARINA, all workers send (at most) $pd + (1-p)\zeta$ floats to the server in expectation, where $\zeta := \max_i \sup_{v \in \mathbb{R}^d} \text{size}(\mathcal{C}_i(v))$, where $\text{size}(\mathcal{C}_i(v))$ is the size of the message $v$ compressed by compressor $\mathcal{C}_i$. For an uncompressed vector $v$ we have $\text{size}(v) = d$ in the worst case, and if $\mathcal{C}_i$ is the RandK sparsifier, then $\text{size}(\mathcal{C}_i(v)) = K$. Putting the above together, the communication complexity of MARINA is $T(pd + (1-p)\zeta)$, i.e., the product of the number of communication rounds and the communication cost of each round. See Section B for more details on the method and its theoretical properties.

An alternative to the application of unbiased compressors is the practice of applying contractive compressors, such as TopK (Alistarh et al., 2018), together with an error feedback mechanism (Seide et al., 2014; Stich et al., 2018; Beznosikov et al., 2020). However, this approach is not competitive in theoretical communication complexity with MARINA; see Appendix G for details.

### 1.3 Summary of contributions

**(a) Correlated and permutation compressors.** We *generalize* the analysis of MARINA *beyond independence* by supporting arbitrary unbiased compressors, including compressors that are *correlated*. In particular, we construct new compressors based on the idea of a *random permutation* (we called them PermK) which provably reduce the variance caused by compression beyond what independent compressors can achieve. The properties of our compressors are captured by two quantities, $A \geq B \geq 0$, through a new inequality (which we call "AB inequality") bounding the variance of the *aggregated* (as opposed to individual) compressed message.

**(b) Refined analysis through the new notion of Hessian variance.** We *refine* the analysis of MARINA by identifying a new quantity, for which we coin the name *Hessian variance*, which plays an important role in our sharper analysis. To the best of our knowledge, Hessian variance is a new quantity proposed in this work and not used in optimization before. This quantity is well defined under the same assumptions as those used in the analysis of MARINA by Gorbunov et al. (2021).

**(c) Improved communication complexity results.** We prove iteration complexity and communication complexity results for MARINA, for smooth nonconvex (Theorem 4) and smooth Polyak-Łojasiewicz[3] (Theorem 5) functions. Our results hold for all unbiased compression operators, including the standard independent but also all *correlated* compressors. Most importantly, we show that in the low Hessian variance regime, and by using our PermK compressors, we can improve upon the current state-of-the-art communication complexity of MARINA due to Gorbunov et al. (2021) by up to the factor $\sqrt{n}$ in the $d \geq n$ case, and up to the factor $1 + d/\sqrt{n}$ in the $d \leq n$ case. The improvement factors degrade gracefully as Hessian variance grows, and in the worst case we recover the same complexity as those established by Gorbunov et al. (2021).

**(d) Experiments agree with our theory.** Our theoretical results lead to predictions which are corroborated through computational experiments. In particular, we perform proof-of-concept testing with carefully engineered synthetic experiments with minimizing the average of nonconvex quadratics, and also test on autoencoder training with the MNIST dataset.

## 2 Beyond Independence: The Power of Correlated Compressors

As mentioned in the introduction, MARINA was designed and analyzed to be used with compressors $\mathcal{C}_i \in \mathbb{U}(\omega)$ that are sampled *independently* by the workers. For example, if the RandK sparsification

---

[2]Gorbunov et al. (2021) *present* their result with $L_-$ replaced by the larger quantity $L_+$. However, after inspecting their proof, it is clear that they proved the improved rate we attribute to them here, and merely used the bound $L_- \leq L_+$ at the end for convenience of presentation only.

[3]The PŁ analysis is included in Appendix D.

operator is used by all workers, then each worker chooses the $K$ random coordinates to be communicated *independently* from the other workers. This independence assumption is crucial for MARINA to achieve its superior theoretical properties. Indeed, without independence, the rate would depend on $\omega$ instead[4] of $\omega/n$, which would mean no improvement as the number $n$ of workers grows, which is problematic because $\omega$ is typically very large. For this reason, independence is assumed in the analysis of virtually all distributed methods that use unbiased communication compression, including methods designed for convex or strongly convex problems (Khirirat et al., 2018; Mishchenko et al., 2019; Li et al., 2020; Philippenko & Dieuleveut, 2020).

In our work we first *generalize* the analysis of MARINA *beyond independence*, which provably extends its use to a much wider array of (still unbiased) compressors, some of which have interesting theoretical properties and are useful in practice.

## 2.1 AB INEQUALITY: A TOOL FOR A MORE PRECISE CONTROL OF COMPRESSION VARIANCE

We assume that all compressors $\{\mathcal{C}_i\}_{i=1}^n$ are unbiased, and that there exist constants $A, B \geq 0$ for which the compressors satisfy a certain inequality, which we call "AB inequality", bounding the variance of $\frac{1}{n} \sum_i \mathcal{C}_i(a_i)$ as a stochastic estimator of $\frac{1}{n} \sum_i a_i$.

**Assumption 3** (Unbiasedness). *The random operators* $\mathcal{C}_1, \ldots, \mathcal{C}_n : \mathbb{R}^d \to \mathbb{R}^d$ *are unbiased, i.e.,* $\mathrm{E}\left[\mathcal{C}_i(a)\right] = a$ *for all* $i \in \{1, 2, \ldots, n\}$ *and all* $a \in \mathbb{R}^d$.

**Assumption 4** (AB inequality). *There exist constants* $A, B \geq 0$ *such that the random operators* $\mathcal{C}_1, \ldots, \mathcal{C}_n : \mathbb{R}^d \to \mathbb{R}^d$ *satisfy the inequality*

$$\mathrm{E}\left[\left\|\frac{1}{n}\sum_{i=1}^n \mathcal{C}_i(a_i) - \frac{1}{n}\sum_{i=1}^n a_i\right\|^2\right] \leq A\left(\frac{1}{n}\sum_{i=1}^n \|a_i\|^2\right) - B\left\|\frac{1}{n}\sum_{i=1}^n a_i\right\|^2 \qquad (3)$$

*for all* $a_1, \ldots, a_n \in \mathbb{R}^d$. *If these conditions are satisfied, we will write* $\{\mathcal{C}_i\}_{i=1}^n \in \mathbb{U}(A, B)$.

It is easy to observe that whenever the AB inequality holds, it must necessarily be the case that $A \geq B$. Indeed, if we fix nonzero $a \in \mathbb{R}^d$ and choose $a_i = a$ for all $i$, then the right hand side of the AB inequality is equal to $A - B$ while the left hand side is nonnegative.

Our next observation is that whenever $\mathcal{C}_i \in \mathbb{U}(\omega_i)$ for all $i \in \{1, 2, \ldots, n\}$, the AB inequality holds without any assumption on the independence of the compressors. Furthermore, if independence is assumed, the $A$ constant is substantially improved.

**Lemma 1.** *If* $\mathcal{C}_i \in \mathbb{U}(\omega_i)$ *for* $i \in \{1, 2, \ldots, n\}$, *then* $\{\mathcal{C}_i\}_{i=1}^n \in \mathbb{U}(\max_i \omega_i, 0)$. *If we further assume that the compressors are independent, then* $\{\mathcal{C}_i\}_{i=1}^n \in \mathbb{U}(\frac{1}{n}\max_i \omega_i, 0)$.

In Table 1 we provide a list of several compressors that belong to the class $\mathbb{U}(A, B)$, and give values of the associated constants $A$ and $B$. While in the two examples captured by Lemma 1 we had $B = 0$, intuitively, we should want $B$ to be as large as possible.

## 2.2 INPUT VARIANCE COMPRESSORS

Due to the above considerations, compressors for which $A = B$ are special, and their construction and theoretical properties are a key contribution of our work. Moreover, as we shall see in Section 4, such compressors have favorable communication complexity properties. This leads to the following definition:

**Definition 1** (Input variance compressors). *We say that a collection* $\{\mathcal{C}_i\}_{i=1}^n$ *of unbiased operators form an* input variance *compressor system if the variance of* $\frac{1}{n}\sum_i \mathcal{C}_i(a_i)$ *is controlled by a multiple of the variance of the input vectors* $\{a_i\}_{i=1}^n$. *That is, if there exists a constant* $C \geq 0$ *such that*

$$\mathrm{E}\left[\left\|\frac{1}{n}\sum_{i=1}^n \mathcal{C}_i(a_i) - \frac{1}{n}\sum_{i=1}^n a_i\right\|^2\right] \leq C\mathrm{Var}(a_1, \ldots, a_n) \qquad (4)$$

*for all* $a_1, \ldots, a_n \in \mathbb{R}^d$. *If these conditions are satisfied, we will write* $\{\mathcal{C}_i\}_{i=1}^n \in \mathbb{IV}(C)$.

If $\{\mathcal{C}_i\}_{i=1}^n \in \mathbb{U}(A, B)$ and $A = B$, then $\{\mathcal{C}_i\}_{i=1}^n \in \mathbb{IV}(A)$.

---

[4]This is a consequence of the more general analysis from our paper; Gorbunov et al. (2021) do not consider the case of unbiased compressors without the independence assumption.

Table 1: Examples of compressors $\{\mathcal{C}_i\}_{i=1}^n \in \mathbb{U}(A, B)$. See the appendix for many more.

| Compressors | $A$ | $B$ | Calculation of $A, B$ | Reference |
|---|---|---|---|---|
| $\mathcal{C}_i \in \mathbb{U}(\omega_i)$ | $\max_i \omega_i$ | $0$ | Lemma 1 | standard |
| $\mathcal{C}_i \in \mathbb{U}(\omega_i)$, independent | $\frac{1}{n}\max_i \omega_i$ | $0$ | Lemma 1 | standard |
| Perm$K$ ($d \geq n$); Def 2 | $1$ | $1$ | Theorem 1 | **new** |
| Perm$K$ ($d \leq n$); Def 3 | $1 - \frac{n-d}{n-1}$ | $1 - \frac{n-d}{n-1}$ | Theorem 2 | **new** |

## 2.3 PERM$K$: PERMUTATION BASED SPARSIFIERS

We now define two input variance compressors based on a *random permutation* construction.[5] The first compressor handles the $d \geq n$ case, and the second handles the $d \leq n$ case. For simplicity of exposition, we assume that $d$ is divisible by $n$ in the first case, and that $n$ is divisible by $d$ in the second case.[6] Since both these new compressors are sparsification operators, in an analogy with the established notation Rand$K$ and Top$K$ for sparsification, we will write Perm$K$ for our permutation-based sparsifiers. To keep the notation simple, we chose to include simple variants which do not offer freedom in choosing $K$. Having said that, these simple compressors lead to state-of-the-art communication complexity results for MARINA, and hence not much is lost by focusing on these examples. Let $e_i$ be the $i^{\text{th}}$ standard unit basis vector in $\mathbb{R}^d$. That is, for any $x = (x_1, \ldots, x_d) \in \mathbb{R}^d$ we have $x = \sum_i x_i e_i$.

**Definition 2** (Perm$K$ for $d \geq n$). *Assume that $d \geq n$ and $d = qn$, where $q \geq 1$ is an integer. Let $\pi = (\pi_1, \ldots, \pi_d)$ be a random permutation of $\{1, \ldots, d\}$. Then for all $x \in \mathbb{R}^d$ and each $i \in \{1, 2, \ldots, n\}$ we define*

$$\mathcal{C}_i(x) := n \cdot \sum_{j=q(i-1)+1}^{qi} x_{\pi_j} e_{\pi_j}. \tag{5}$$

Note that $\mathcal{C}_i$ is a sparsifier: we have $(\mathcal{C}_i(x))_l = nx_l$ if $l \in \{\pi_j \; : \; q(i-1)+1 \leq j \leq qi\}$ and $(\mathcal{C}_i(x))_l = 0$ otherwise. So, $\|\mathcal{C}_i(x)\|_0 \leq q := K$, which means that $\mathcal{C}_i$ offers compression by the factor $n$. Note that we do not have flexibility to choose $K$; we have $K = q = d/n$. See Appendix J for implementation details.

**Theorem 1.** *The Perm$K$ compressors from Definition 2 are unbiased and belong to $\mathbb{IV}(1)$.*

In contrast with the collection of independent Rand$K$ sparsifiers, which satisfy the AB inequality with $A = \frac{d/K-1}{n}$ and $B = 0$ (this follows from Lemma 1 since $\omega_i = d/K - 1$ for all $i$), Perm$K$ satisfies the AB inequality with $A = B = 1$. While both are sparsifiers, the permutation construction behind Perm$K$ introduces a favorable correlation among the compressors: we have $\langle \mathcal{C}_i(a_i), \mathcal{C}_j(a_j) \rangle = 0$ for all $i \neq j$.

**Definition 3** (Perm$K$ for $n \geq d$). *Assume that $n \geq d$, $n > 1$ and $n = qd$, where $q \geq 1$ is an integer. Define the multiset $S := \{1, \ldots, 1, 2, \ldots, 2, \ldots, d, \ldots, d\}$, where each number occurs precisely $q$ times. Let $\pi = (\pi_1, \ldots, \pi_n)$ be a random permutation of $S$. Then for all $x \in \mathbb{R}^d$ and each $i \in \{1, 2, \ldots, n\}$ we define*

$$\mathcal{C}_i(x) := dx_{\pi_i} e_{\pi_i}. \tag{6}$$

Note that for each $i$, $\mathcal{C}_i$ from Definition 3 *is* the Rand1 sparsifier, offering compression factor $d$. However, the sparsifiers $\{\mathcal{C}_i\}_{i=1}^n$ are *not* mutually independent. Note that, again, we do not have a choice of $K$ in Definition 3: we have $K = 1$.

**Theorem 2.** *The Perm$K$ compressors from Definition 3 are unbiased and belong to $\mathbb{IV}(A)$ with $A = 1 - \frac{n-d}{n-1}$.*

**Combining Perm$K$ with quantization.** It is easy to show that if $\{\mathcal{C}_i\}_{i=1}^n \in \mathbb{U}(A, B)$, and $\mathcal{Q}_i \in \mathbb{U}(\omega_i)$ are chosen independently of $\{\mathcal{C}_i\}_{i=1}^n$ (we do not require mutual independence of $\{\mathcal{Q}_i\}$), then $\{\mathcal{C}_i \circ \mathcal{Q}_i\}_{i=1}^n \in \mathbb{U}((\max_i \omega_i + 1)A, B)$ (see Lemma 9). This allows us to combine our compression techniques with quantization (Alistarh et al., 2017; Horváth et al., 2019).

---

[5]More examples of input variance compressors are given in the appendix.
[6]The general situation is handled in Appendix I.

Table 2: Value of $L_\pm^2$ in cases when $f_i(x) = \phi(x) + \phi_i(x)$, where $\phi : \mathbb{R}^d \to \mathbb{R}$ is an arbitrary differentiable function and $\phi_i : \mathbb{R}^d \to \mathbb{R}$ is twice continuously differentiable. The matrices $\boldsymbol{A}_i \in \mathbb{R}^{d \times d}$ are assumed (without loss of generality) to be symmetric. The matrix-valued function $\boldsymbol{L}_\pm(x, y)$ is defined in Theorem 3.

| $\phi(x)$ | $\phi_i(x)$ | Hessian variance $L_\pm^2$ |
|:---:|:---:|:---:|
| any | $0$ | $0$ |
| any | $b_i^\top x + c_i$ | $0$ |
| $0$ | $\frac{1}{2} x^\top \boldsymbol{A}_i x + b_i^\top x + c_i$ | $\lambda_{\max} \left( \frac{1}{n} \sum_{i=1}^n \boldsymbol{A}_i^2 - \left( \frac{1}{n} \sum_{i=1}^n \boldsymbol{A}_i \right)^2 \right)$ |
| $0$ | smooth | $\displaystyle\sup_{x,y \in \mathbb{R}^d, x \neq y} \frac{(x-y)^\top \boldsymbol{L}_\pm(x,y)(x-y)}{\|x-y\|^2}$ |

## 3  HESSIAN VARIANCE

Working under the *same* assumptions on the problem (1)–(2) as Gorbunov et al. (2021) (i.e., Assumptions 1 and 2), in this paper we study the complexity of MARINA under the influence of a new quantity, which we call *Hessian variance*.

**Definition 4** (Hessian variance). *Let $L_\pm \geq 0$ be the smallest quantity such that*

$$\frac{1}{n} \sum_{i=1}^n \|\nabla f_i(x) - \nabla f_i(y)\|^2 - \|\nabla f(x) - \nabla f(y)\|^2 \leq L_\pm^2 \|x - y\|^2, \quad \forall x, y \in \mathbb{R}^d. \qquad (7)$$

*We refer to the quantity $L_\pm^2$ by the name* Hessian variance.

Recall that in this paper we have so far mentioned four "smoothness" constants: $L_i$ (Lipschitz constant of $\nabla f_i$), $L_-$ (Lipschitz constant of $\nabla f$), $L_+$ (see Assumption 2) and $L_\pm$ (Definition 4). To avoid ambiguity, let all be defined as the smallest constants for which the defining inequalities hold. In case the defining inequality does not hold, the value is set to $+\infty$. This convention allows us to formulate the following result summarizing the relationships between these quantities.

**Lemma 2.** $L_- \leq L_+$, $L_- \leq \frac{1}{n} \sum_{i=1}^n L_i$, $L_+^2 \leq \frac{1}{n} \sum_{i=1}^n L_i^2$, *and* $L_+^2 - L_-^2 \leq L_\pm^2 \leq L_+^2$.

It follows that if $L_i$ is finite for all $i$, then $L_-, L_+$ and $L_\pm$ are all finite as well. Similarly, if $L_+$ is finite (i.e., if Assumption 2 holds), then $L_-$ and $L_\pm$ are finite, and $L_\pm \leq L_+$. We are not aware of any prior use of this quantity in the analysis of any optimization methods. Importantly, there are situations when $L_-$ is large, and yet the Hessian variance $L_\pm^2$ is small, or even zero. This is important as the improvements we obtain in our analysis of MARINA are most pronounced in the regime when the Hessian variance is small. We also wish to stress that the finiteness of $L_\pm$ is *not* an additional assumption – it follows directly from Assumption 2.

### 3.1  HESSIAN VARIANCE CAN BE ZERO

We now illustrate on a few examples that there are situations when the values of $L_-$ and $L_i$ are large and the Hessian variance is zero. The simplest such example is the identical functions regime.

**Example 1** (Identical functions). *Assume that $f_1 = f_2 = \cdots = f_n$. Then $L_\pm = 0$.*

This follows by observing that the left hand side in (7) is zero. Note that while $L_\pm = 0$, it is possible for $L_-$ and $L_+$ to be arbitrarily large! Note that methods based on the $\mathrm{Top}K$ compressor (including all error feedback methods) suffer in this regime. Indeed, EF21 in this simple scenario is the same method for any value of $n$, and hence can't possibly improve as $n$ grows. This is because when $a_i = a_j$ for all $i, j$, $\frac{1}{n} \sum_i \mathrm{Top}K(a_i) = \mathrm{Top}K(a_i)$. As the next example shows, Hessian variance is zero even if we perturb the local functions via *arbitrary* linear functions.

**Example 2** (Identical functions + arbitrary linear perturbation). *Assume that $f_i(x) = \phi(x) + b_i^\top x + c_i$, for some differentiable function $\phi : \mathbb{R}^d \to \mathbb{R}$ and arbitrary $b_i \in \mathbb{R}^d$ and $c_i \in \mathbb{R}$. Then $L_\pm = 0$.*

This follows by observing that the left hand side in (7) is zero in this case as well. Note that in this example it is possible for the functions $\{f_i\}$ to have *arbitrarily different minimizers*. So, this

example does *not* correspond to the overparameterized machine learning regime, and is in general challenging for standard methods.

## 3.2 SECOND ORDER CHARACTERIZATION

To get an insight into when the Hessian variance may be small but not necessarily zero, we establish a useful second order characterization.

**Theorem 3.** *Assume that for each $i \in \{1, 2, \ldots, n\}$, the function $f_i$ is twice continuously differentiable. Fix any $x, y \in \mathbb{R}^d$ and define $\boldsymbol{H}_i(x, y) := \int_0^1 \nabla^2 f_i(x + t(y - x)) \, dt$, $\boldsymbol{H}(x, y) := \frac{1}{n} \sum_{i=1}^n \boldsymbol{H}_i(x, y)$. Then the matrices $\boldsymbol{L}_i(x, y) := \boldsymbol{H}_i^2(x, y)$, $\boldsymbol{L}_-(x, y) := \boldsymbol{H}^2(x, y)$, $\boldsymbol{L}_+(x, y) := \frac{1}{n} \sum_{i=1}^n \boldsymbol{H}_i^2(x, y)$ and $\boldsymbol{L}_\pm(x, y) := \boldsymbol{L}_+(x, y) - \boldsymbol{L}_-(x, y)$ are symmetric and positive semidefinite. Moreover,*

$$L_i^2 = \sup_{x,y \in \mathbb{R}^d, x \neq y} \frac{(x-y)^\top \boldsymbol{L}_i(x,y)(x-y)}{\|x-y\|^2}, \quad L_-^2 = \sup_{x,y \in \mathbb{R}^d, x \neq y} \frac{(x-y)^\top \boldsymbol{L}_-(x,y)(x-y)}{\|x-y\|^2},$$

$$L_+^2 = \sup_{x,y \in \mathbb{R}^d, x \neq y} \frac{(x-y)^\top \boldsymbol{L}_+(x,y)(x-y)}{\|x-y\|^2}, \quad L_\pm^2 = \sup_{x,y \in \mathbb{R}^d, x \neq y} \frac{(x-y)^\top \boldsymbol{L}_\pm(x,y)(x-y)}{\|x-y\|^2}.$$

While $L_\pm^2$ is obviously well defined through Definition 4 even when the functions $\{f_i\}$ are *not* twice differentiable, the term "Hessian variance" comes from the interpretation of $L_\pm^2$ in the case of quadratic functions.

**Example 3** (Quadratic functions). *Let $f_i(x) = \frac{1}{2} x^\top \boldsymbol{A}_i x + b_i^\top x + c_i$, where $\boldsymbol{A}_i \in \mathbb{R}^{d \times d}$ are symmetric. Then $L_\pm^2 = \lambda_{\max}(\frac{1}{n} \sum_{i=1}^n \boldsymbol{A}_i^2 - \left(\frac{1}{n} \sum_{i=1}^n \boldsymbol{A}_i\right)^2)$, where $\lambda_{\max}(\cdot)$ denotes the largest eigenvalue.*

Indeed, note that the matrix $\frac{1}{n} \sum_{i=1}^n \boldsymbol{A}_i^2 - \left(\frac{1}{n} \sum_{i=1}^n \boldsymbol{A}_i\right)^2$ can be interpreted as a matrix-valued variance of the Hessians $\boldsymbol{A}_1, \ldots, \boldsymbol{A}_n$, and $L_\pm^2$ measures the size of this matrix in terms of its largest eigenvalue.

See Table 2 for a summary of the examples mentioned above. As we shall explain in Section 4, the data/problem regime when the Hessian variance is small is of key importance to the improvements we obtain in this paper.

## 4 IMPROVED ITERATION AND COMMUNICATION COMPLEXITY

The key contribution of our paper is a more general and more refined analysis of MARINA. In particular, we i) extend the reach of MARINA to the general class of unbiased and possibly correlated compressors $\{\mathcal{C}_i\}_{i=1}^n \in \mathbb{U}(A, B)$ while ii) providing a more refined analysis in that we take the Hessian variance $L_\pm^2$ into account. [7]

**Theorem 4.** *Let Assumptions 1, 2, 3 and 4 be satisfied. Let the stepsize in MARINA be chosen as $0 < \gamma \leq \frac{1}{M}$, where $M = L_- + \sqrt{\frac{1-p}{p} \left((A - B)L_+^2 + B L_\pm^2\right)}$. Then after $T$ iterations, MARINA finds a random point $\hat{x}^T \in \mathbb{R}^d$ for which*

$$\mathrm{E}\left[\left\|\nabla f(\hat{x}^T)\right\|^2\right] \leq \frac{2\Delta^0}{\gamma T}.$$

In particular, by choosing the maximum stepsize allowed by Theorem 4, MARINA converges in $T$ communication rounds, where $T$ is shown in the first row Table 3. If in this result we replace $L_\pm^2$ by the coarse estimate $L_\pm^2 \leq L_+^2$, and further specialize to independent compressors satisfying $\mathcal{C}_i \in \mathbb{U}(\omega)$ for all $i \in \{1, 2, \ldots, n\}$, then since $\{\mathcal{C}_i\}_{i=1}^n \in \mathbb{U}(\omega/n, 0)$ (recall Lemma 1), our general rate specializes to the result of Gorbunov et al. (2021), which we show in the second row of Table 3.

---

[7]At this point, the authors wish to point out an independent work by Yun et al. (2022) which appeared online shortly after ours. While the problem addressed by Yun et al. (2022) is different, the spirit remains the same – improving the performance of a method of interest by designing carefully correlated randomness.

Table 3: The number of communication rounds for solving (1)–(2) by MARINA and EF21.

| Method + Compressors | $T = \#$ Communication Rounds |
|---|---|
| MARINA $\bigcap \{\mathcal{C}_i\}_{i=1}^n \in \mathbb{U}(A, B)$ (this paper, 2021) | $\mathcal{O}\left(\frac{\Delta^0}{\varepsilon}\left(L_- + \sqrt{\frac{1-p}{p}\left((A-B)L_+^2 + BL_\pm^2\right)}\right)\right)$ |
| MARINA $\bigcap \mathcal{C}_i \in \mathbb{U}(\omega)$ and independent (Gorbunov et al., 2021) | $\mathcal{O}\left(\frac{\Delta^0}{\varepsilon}\left(L_- + \sqrt{\frac{1-p}{p}\frac{\omega}{n}}L_+\right)\right)$ |
| EF21 $\bigcap \mathcal{C}_i$ are $\alpha$-contractive (Richtárik et al., 2021) | $\mathcal{O}\left(\frac{\Delta^0}{\varepsilon}\left(L_- + \left(\frac{1+\sqrt{1-\alpha}}{\alpha} - 1\right)L_+\right)\right)$ |

Table 4: Optimized communication complexity of MARINA and EF21 with particular compressors.

| | Communication Complexity | |
|---|---|---|
| Method + Compressor | $d \geq n$ (Lemma 13) | $d \leq n$ (Lemma 14) |
| MARINA $\bigcap \mathrm{Perm}K$ | $\mathcal{O}\left(\frac{\Delta^0}{\varepsilon}\min\left\{dL_-, \frac{d}{n}L_- + \frac{d}{\sqrt{n}}L_\pm\right\}\right)$ | $\mathcal{O}\left(\frac{\Delta^0}{\varepsilon}\min\left\{dL_-, L_- + \frac{d}{\sqrt{n}}L_\pm\right\}\right)$ |
| MARINA $\bigcap \mathrm{Rand}K$ | $\mathcal{O}\left(\frac{\Delta^0}{\varepsilon}\min\left\{dL_-, \frac{d}{\sqrt{n}}L_+\right\}\right)$ | $\mathcal{O}\left(\frac{\Delta^0}{\varepsilon}\min\left\{dL_-, L_- + \frac{d}{\sqrt{n}}L_+\right\}\right)$ |
| EF21 $\bigcap \mathrm{Top}K$ | $\mathcal{O}\left(\frac{\Delta^0}{\varepsilon}dL_-\right)$ | $\mathcal{O}\left(\frac{\Delta^0}{\varepsilon}dL_-\right)$ |

(a) Note, that $L_- \leq L_+$ and $L_\pm \leq L_+$.

*However, and this is a key finding of our work, in the regime when the Hessian variance $L_\pm^2$ is very small, the original result of Gorbunov et al. (2021) can be vastly suboptimal!* To show this, in Table 4 we compare the *communication complexity*, i.e., the # of communication rounds multiplied by the maximum # of floats transmitted by a worker to the sever in a single communication round. We compare the communication complexity of MARINA with the $\mathrm{Rand}K$ and $\mathrm{Perm}K$ compressors, and the state-of-the-art error-feedback method EF21 of Richtárik et al. (2021) with the $\mathrm{Top}K$ compressor. In all cases we do not consider the communication complexity of the initial step equal to $\mathcal{O}(d)$. In each case we optimized over the parameters of the methods (e.g., $p$ for MARINA and $K$ in all cases; for details see Appendix L). Our results for MARINA with $\mathrm{Perm}K$ are better than the competing methods (recall Lemma 2).

### 4.1 Improvements in the ideal zero-Hessian-variance regime

To better understand the improvements our analysis provides, let us consider the ideal regime characterized by zero Hessian variance: $L_\pm^2 = 0$. If we now use compressors $\{\mathcal{C}_i\}_{i=1}^n \in \mathbb{U}(A, B)$ for which $A = B$, which is the case for $\mathrm{Perm}K$, then the dependence on the potentially very large quantity $L_+^2$ is eliminated completely.

**Big model case ($d \geq n$).** In this case, and using the $\mathrm{Perm}K$ compressor, MARINA has communication complexity $\mathcal{O}(L_- \Delta^0 \varepsilon^{-1} d/n)$, while using the $\mathrm{Rand}K$ compressor, the communication complexity of MARINA is no better than $\mathcal{O}(L_- \Delta^0 \varepsilon^{-1} d/\sqrt{n})$. Hence, we get an improvement by *at least* the factor $\sqrt{n}$. Moreover, note that this is an $n\times$ improvement over gradient descent (GD) (Khaled & Richtárik, 2020) and EF21, both of which have communication complexity $\mathcal{O}(L_- \Delta^0 \varepsilon^{-1} d)$. In Appendix M, we discuss how we can get the same theoretical improvement even if $L_\pm^2 > 0$.

**Big data case ($d \leq n$).** In this case, and using the $\mathrm{Perm}K$ compressor, MARINA achieves communication complexity $\mathcal{O}(L_- \Delta^0 \varepsilon^{-1})$, while using the $\mathrm{Rand}K$ compressor, the communication complexity of MARINA is no better than $\mathcal{O}(L_- \Delta^0 \varepsilon^{-1}(1 + d/\sqrt{n}))$. Hence, we get an improvement by *at least* the factor $1 + d/\sqrt{n}$. Moreover, note that this is a $d\times$ improvement over gradient descent (GD) and EF21, both of which have communication complexity $\mathcal{O}(L_- \Delta^0 \varepsilon^{-1} d)$.

## 5 Experiments

We compare MARINA using $\mathrm{Rand}K$ and $\mathrm{Perm}K$, and EF21 with $\mathrm{Top}K$, in two experiments. In the first experiment, we construct quadratic optimization tasks with different $L_\pm$ to capture the dependencies that our theory predicts. In the second experiment, we consider practical machine learning

task MNIST (LeCun et al., 2010) to support our assertions. Each plot represents the dependence between the norm of gradient (or function value) and the total number of transmitted bits by a node.

## 5.1 TESTING THEORETICAL PREDICTIONS ON A SYNTHETIC QUADRATIC PROBLEM

To test the predictive power of our theory in a controlled environment, we first consider a synthetic (strongly convex) quadratic function $f = \frac{1}{n} \sum f_i$ composed of nonconvex quadratics $f_i(x) := \frac{1}{2} x^\top \boldsymbol{A}_i x - x^\top b_i$, where $b_i \in \mathbb{R}^d$, $\boldsymbol{A}_i \in \mathbb{R}^{d \times d}$, and $\boldsymbol{A}_i = \boldsymbol{A}_i^\top$. We enforced that $f$ is $\lambda$–strongly convex, i.e., $\frac{1}{n} \sum_{i=1}^{n} \boldsymbol{A}_i \succcurlyeq \lambda \boldsymbol{I}$ for $\lambda > 0$. We fix $\lambda = 1\mathrm{e}{-6}$, and dimension $d = 1000$ (see Figure 1). We then generated optimization tasks with the number of nodes $n \in \{10, 1000, 10000\}$ and $L_\pm \in \{0, 0.05, 0.1, 0.21, 0.91\}$. We take MARINA's and EF21's parameters prescribed by the theory and performed a grid search for the step sizes for each compressor by multiplying the theoretical ones with powers of two. For simplicity, we provide one plot for each compressor with the best convergence rate. First, we see that $\mathrm{Perm}K$ outperforms $\mathrm{Rand}K$, and their differences in the plots reproduce dependencies from Table 4. Moreover, when $n \in \{1000, 10000\}$ and $L_\pm \leq 0.21$, EF21 with $\mathrm{Top}K$ has worse performance than MARINA with $\mathrm{Perm}K$, while in heterogeneous regime, when $L_\pm = 0.91$, $\mathrm{Top}K$ is superior except when $n = 10000$. See Appendix A for detailed experiments.

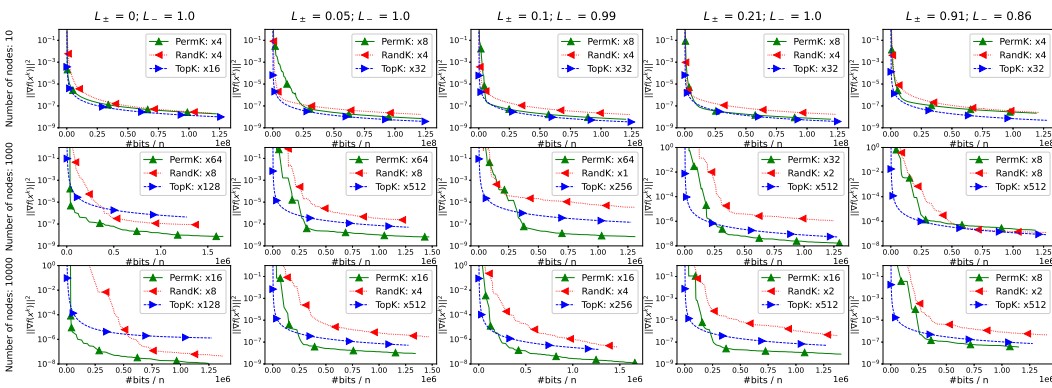

Figure 1: Comparison of algorithms on synthetic quadratic optimization tasks with nonconvex $\{f_i\}$.

## 5.2 TRAINING AN AUTOENCODER WITH MNIST

Now we compare compressors from Section 5.1 on the MNIST dataset (LeCun et al., 2010). Our current goal is to learn the linear autoencoder, $f(\boldsymbol{D}, \boldsymbol{E}) := \frac{1}{N} \sum_{i=1}^{N} \|\boldsymbol{D}\boldsymbol{E} a_i - a_i\|^2$, where $\boldsymbol{D} \in \mathbb{R}^{d_f \times d_e}$, $\boldsymbol{E} \in \mathbb{R}^{d_e \times d_f}$, $a_i \in \mathbb{R}^{d_f}$ are MNIST images, $d_f = 784$ is the number of features, $d_e = 16$ is the size of encoding space. Thus the dimension of the problem $d = 25088$, and compressors send at most 26 floats in each communication round since we take $n = 1000$. We use parameter $\hat{p}$ to control the homogeneity of MNIST split among $n$ nodes: if $\hat{p} = 1$, then all nodes store the same data, and if $\hat{p} = 0$, then nodes store different splits (see Appendix A.5). In Figure 2, one plot for each compressor with the best convergence rate is provided for $\hat{p} \in \{0, 0.5, 0.75, 0.9, 1.0\}$. We choose parameters of algorithms prescribed by the theory except for the step sizes, where we performed a grid search as before. In all experiments, $\mathrm{Perm}K$ outperforms $\mathrm{Rand}K$. Moreover, we see that in the more homogeneous regimes, when $\hat{p} \in \{0.9, 1.0\}$, $\mathrm{Perm}K$ converges faster than $\mathrm{Top}K$. When $\hat{p} = 0.75$, both compressors have almost the same performance. In the heterogenous regime, when $\hat{p} \in \{0, 0.5\}$, $\mathrm{Top}K$ is faster than $\mathrm{Perm}K$, but they both significantly outperform $\mathrm{Rand}K$.

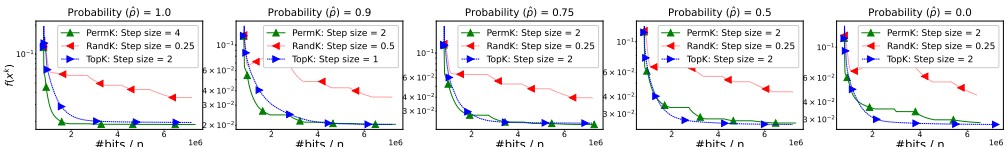

Figure 2: Comparison of algorithms on the encoding learning task for the MNIST dataset.

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

# Appendix

CONTENTS

## A    EXTRA EXPERIMENTS

In this section, we provide more detailed experiments and explanations.

### A.1    EXPERIMENTS SETUP

All methods are implemented in Python 3.6 and run on a machine with 24 Intel(R) Xeon(R) Gold 6146 CPU @ 3.20GHz cores with 32-bit precision. Communication between master and nodes is emulated in one machine.

In all experiments, we compare MARINA algorithm with $\text{Rand}K$ compressor and $\text{Perm}K$ compressor and EF21 with $\text{Top}K$. In $\text{Rand}K$ and $\text{Top}K$, we take $K = \lceil d/n \rceil$; we show in Lemma 13 that $K = \lceil d/n \rceil$ is optimal for $\text{Rand}K$. For $\text{Top}K$, the optimal rate predicted by the current state-of-the-art theory is obtained when $K = d$ (however, in practice, $\text{Top}K$ works much better when $K \ll d$). Lastly, we make the pessimistic assumption that $L_\pm^2$ and $L_+^2$ are equal to their upper bound $\frac{1}{n}\sum_{i=1}^n L_i^2$.

### A.2    EXPERIMENT WITH QUADRATIC OPTIMIZATION TASKS: FULL DESCRIPTION

First, we present Algorithm 2 which is used in the experiments of Section 5.1. The algorithm is designed to generate sparse quadratic optimization tasks where we can control $L_\pm$ using the noise scale. Furthermore, it can be seen that the procedure generates strongly convex quadratic optimization tasks; thus, all assumptions from this paper are fulfilled to use theoretical results.

---

**Algorithm 2** Quadratic optimization task generation

1: **Parameters:** number nodes $n$, dimension $d$, regularizer $\lambda$, and noise scale $s$.
2: **for** $i = 1, \ldots, n$ **do**
3:    Generate random noises $\nu_i^s = 1 + s\xi_i^s$ and $\nu_i^b = s\xi_i^b$, i.i.d. $\xi_i^s, \xi_i^b \sim \mathcal{N}(0,1)$
4:    Take vector $b_i = \frac{\nu_i^s}{4}(-1 + \nu_i^b, 0, \cdots, 0) \in \mathbb{R}^d$
5:    Take the initial tridiagonal matrix

$$\boldsymbol{A}_i = \frac{\nu_i^s}{4}\begin{pmatrix} 2 & -1 & & 0 \\ -1 & \ddots & \ddots & \\ & \ddots & \ddots & -1 \\ 0 & & -1 & 2 \end{pmatrix} \in \mathbb{R}^{d \times d}$$

6: **end for**
7: Take the mean of matrices $\boldsymbol{A} = \frac{1}{n}\sum_{i=1}^n \boldsymbol{A}_i$
8: Find the minimum eigenvalue $\lambda_{\min}(\boldsymbol{A})$
9: **for** $i = 1, \ldots, n$ **do**
10:    Update matrix $\boldsymbol{A}_i = \boldsymbol{A}_i + (\lambda - \lambda_{\min}(\boldsymbol{A}))\boldsymbol{I}$
11: **end for**
12: Take starting point $x^0 = (\sqrt{d}, 0, \cdots, 0)$
13: **Output:** matrices $\boldsymbol{A}_1, \cdots, \boldsymbol{A}_n$, vectors $b_1, \cdots, b_n$, starting point $x^0$

---

Homogeneity of optimizations tasks is controlled by noise scale $s$; indeed, with noise scale equal to zero, all matrices are equal, and, by increasing noise scale, functions become less "similar" and $L_\pm^2$ grows. In Section 5.1, we take noise scales $s \in \{0, 0.05, 0.1, 0.2, 0.8\}$.

In Figure 3, we provide the same experiments as in Section 5.1 but with $\lambda = 0.0001$ to capture dependencies under PŁ condition. Here, we also see that $\text{Perm}K$ has better performance when the number of nodes $n \geq 1000$ and $L_\pm \leq 0.21$.

### A.3    COMPARISON OF MARINA WITH $\text{Rand}K$ AND MARINA WITH $\text{Perm}K$ ON QUADRATIC OPTIMIZATION PROBLEMS

In this section, we provide detailed experiments from Section 5.1 and comparisons between $\text{Rand}K$ and $\text{Perm}K$ using different step sizes (see Figure 4 and Figure 5). We omitted plots where algorithms

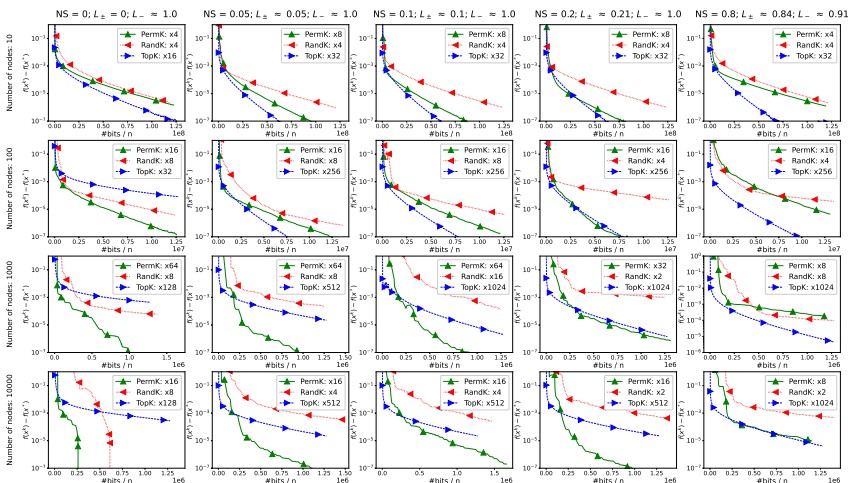

Figure 3: Comparison of algorithms under PŁ condition on synthetic quadratic optimization tasks. Each row corresponds to a fixed number of nodes; each column corresponds to a fixed noise scale. In the legends, we provide compressor names and fine-tuned multiplicity factors of step sizes relative to theoretical ones. Abbreviations: NS = noise scale. Axis $x$ represents the number of bits that every node has sent. Dimension $d = 1000$.

diverged. We can see that in all experiments, Perm$K$ behaves better than Rand$K$ and tolerates larger step sizes. The improvement becomes more significant when $n$ increases.

### A.4  COMPARISON OF EF21 WITH TOP$K$ AND MARINA WITH PERM$K$ ON QUADRATIC OPTIMIZATION PROBLEMS

In this section, we provide detailed experiments from Section 5.1 and comparisons of EF21 with Top$K$ and MARINA with Perm$K$ with different step sizes (see Figure 6 and Figure 7). We omitted plots where algorithms diverged. As we can see, when $L_{\pm} \leq 0.21$ and $n \geq 10000$, Perm$K$ converges faster than Top$K$. While in heterogeneous regimes, when $L_{\pm}$ is large, Top$K$ has better performance except when $n = 10000$. When $n > d$, we see that Perm$K$ converges faster in all experiments.

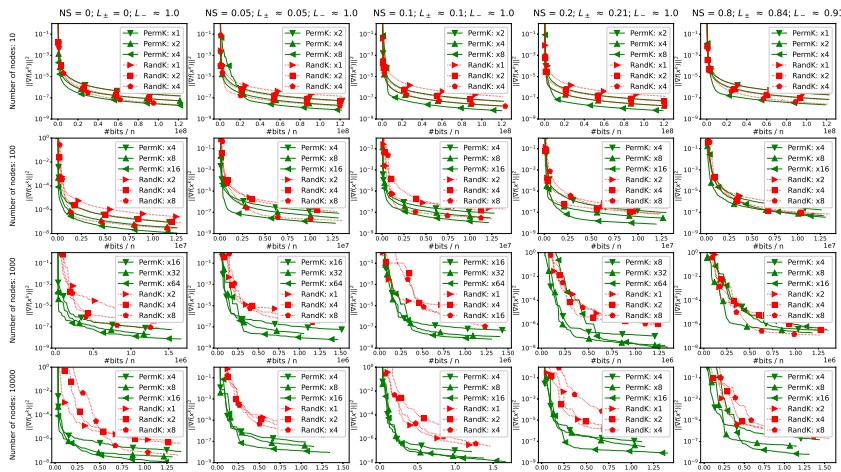

Figure 4: Comparison of Rand$K$ and Perm$K$ on synthetic quadratic optimization tasks. Each row corresponds to a fixed number of nodes; each column corresponds to a fixed noise scale. In the legends, we provide compressor names and fine-tuned multiplicity factors of step sizes relative to theoretical ones. Abbreviations: NS = noise scale. Axis $x$ represents the number of bits that every node has sent. Dimension $d = 1000$.

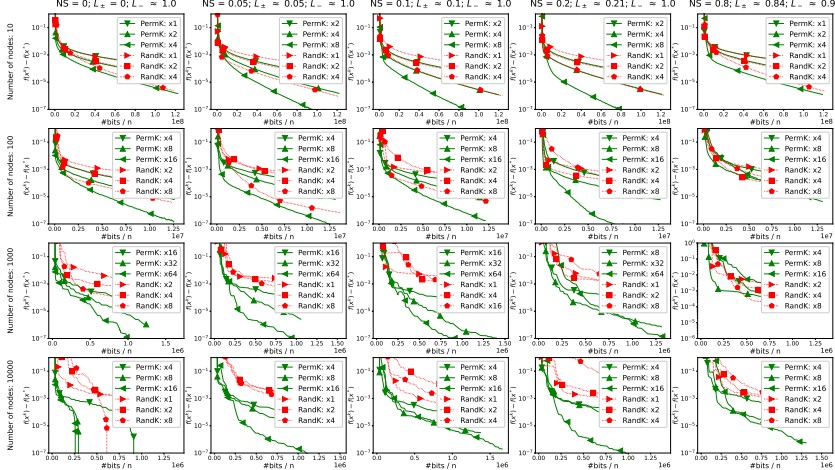

Figure 5: Comparison of Rand$K$ and Perm$K$ under PŁ condition on synthetic quadratic optimization tasks. Each row corresponds to a fixed number of nodes; each column corresponds to a fixed noise scale. In the legends, we provide compressor names and fine-tuned multiplicity factors of step sizes relative to theoretical ones. Abbreviations: NS = noise scale. Axis $x$ represents the number of bits that every node has sent. Dimension $d = 1000$.

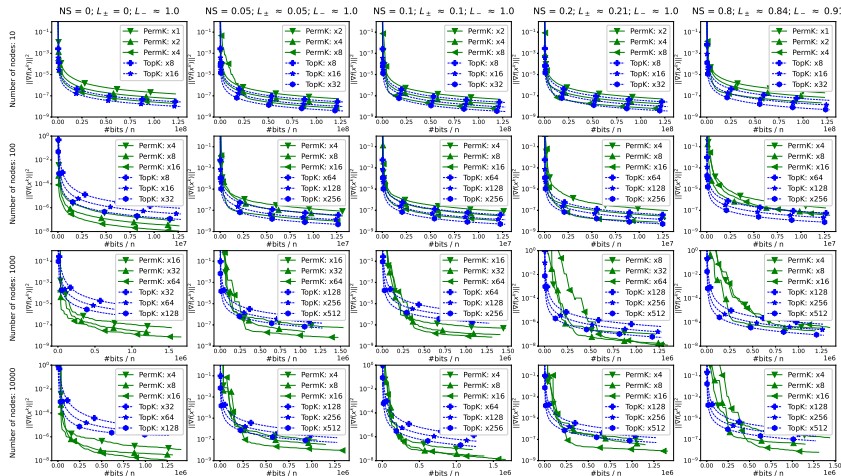

Figure 6: Comparison of TopK and PermK on synthetic quadratic optimization tasks. Each row corresponds to a fixed number of nodes; each column corresponds to a fixed noise scale. In the legends, we provide compressor names and fine-tuned multiplicity factors of step sizes relative to theoretical ones. Abbreviations: NS = noise scale. Axis $x$ represents the number of bits that every node has sent. Dimension $d = 1000$.

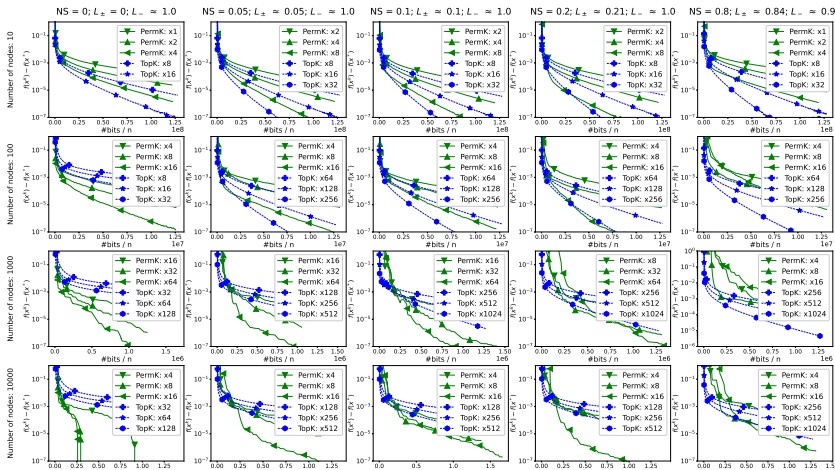

Figure 7: Comparison of TopK and PermK under PŁ condition on synthetic quadratic optimization tasks. Each row corresponds to a fixed number of nodes; each column corresponds to a fixed noise scale. In the legends, we provide compressor names and fine-tuned multiplicity factors of step sizes relative to theoretical ones. Abbreviations: NS = noise scale. Axis $x$ represents the number of bits that every node has sent. Dimension $d = 1000$.

## A.5 EXPERIMENT WITH MNIST: FULL DESCRIPTION

We introduce parameter $\widehat{p}$. Initially, we randomly split MNIST into $n + 1$ parts: $D_0, D_1, \cdots, D_n$, where $n = 1000$ is the number of nodes. Then, for all $i \in \{1, \ldots, n\}$, the $i^{\text{th}}$ node takes split $D_0$ with probability $\widehat{p}$, or split $D_i$ with probability $1 - \widehat{p}$. We define the chosen split as $\widehat{D_i}$. Using probability $\widehat{p}$, we control the homogeneity of our distribution optimization task. Note that if $\widehat{p} = 1$, all nodes store the same data $D_0$, and if $\widehat{p} = 0$, nodes store different splits $D_i$.

Let us consider the more general optimization problem than in Section 5.2. We optimize the following non-convex loss with regularization:

$$\min_{\boldsymbol{D} \in \mathbb{R}^{d_f \times d_e}, \boldsymbol{E} \in \mathbb{R}^{d_e \times d_f}} \left[ f(\boldsymbol{D}, \boldsymbol{E}) := \frac{1}{N} \sum_{i=1}^{N} \|\boldsymbol{DE}a_i - a_i\|^2 + \frac{\lambda}{2} \|\boldsymbol{DE} - \boldsymbol{I}\|_F^2 \right],$$

where $a_i \in \mathbb{R}^{d_f}$ are MNIST images, $d_f = 784$ is the number of features, $d_e = 16$ is the size of encoding space. regularizer $\lambda \geq 0$.

Each node stores function

$$f_i(\boldsymbol{D}, \boldsymbol{E}) := \frac{1}{|\widehat{D_i}|} \sum_{j \in \widehat{D_i}} \|\boldsymbol{DE}a_j - a_j\|^2 + \frac{\lambda}{2} \|\boldsymbol{DE} - \boldsymbol{I}\|_F^2, \quad \forall i \in \{1, \ldots, n\}.$$

In Figure 8, one plot for each compressor with the best convergence rate is provided for $\lambda = \{0, 0.00001, 0.001\}$ and $\widehat{p} = \{0, 0.5, 0.75, 0.9, 1.0\}$.

We see that in homogeneous regimes, when $\widehat{p} \in \{0.9, 1.0\}$, PermK outperforms other compressors for any $\lambda$. And the larger the regularization parameter $\lambda$, the faster PermK convergences compared to rivals.

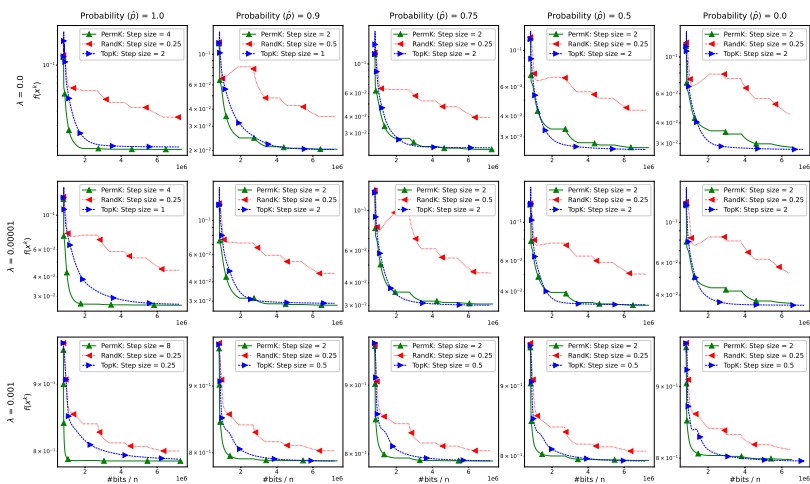

Figure 8: Comparison of algorithms on the encoding learning task for the MNIST dataset. Each row corresponds to a fixed regularization parameter $\lambda$; each column corresponds to a fixed probability $\widehat{p}$. In the legends, we provide compressor names and fine-tuned step sizes. Axis $x$ represents the number of bits that every node has sent.

## A.6 COMPARISON OF MARINA WITH RANDK AND MARINA WITH PERMK ON MNIST DATASET

In this section, we provide detailed experiments from Section 5.2 and comparisons of RandK and PermK with different step sizes (see Figure 9). We omitted plots where algorithms diverged. We see that in all experiments, PermK is better than RandK. Practical experiments on MNIST fully reproduce dependencies from our theory and experiments with synthetic quadratic optimization tasks from Section 5.1.

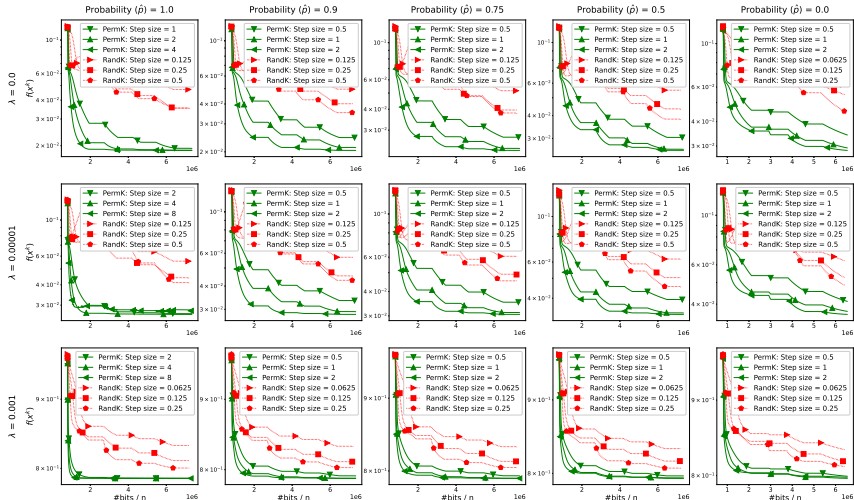

Figure 9: Comparison of Rand$K$ and Perm$K$ on the encoding learning task for the MNIST dataset. Each row corresponds to a fixed regularization parameter $\lambda$; each column corresponds to a fixed probability $\widehat{p}$. In the legends, we provide compressor names and fine-tuned step sizes. Axis $x$ represents the number of bits that every node has sent.

## A.7 COMPARISON OF EF21 WITH TOP$K$ AND MARINA WITH PERM$K$ ON MNIST DATASET

In this section, we provide detailed experiments from Section 5.2 and comparisons of Rand$K$ and Perm$K$ with different step sizes (see Figure 10). We omitted plots where algorithms diverged. We see that, when $\widehat{p} \in \{0.9, 1.0\}$, Perm$K$ tolerates larger step sizes and convergences faster than Top$K$. When $\widehat{p} \in \{0, 0.5\}$, both compressors approximately tolerate the same step sizes, but Top$K$ has a better performance when $\lambda \in \{0, 0.00001\}$.

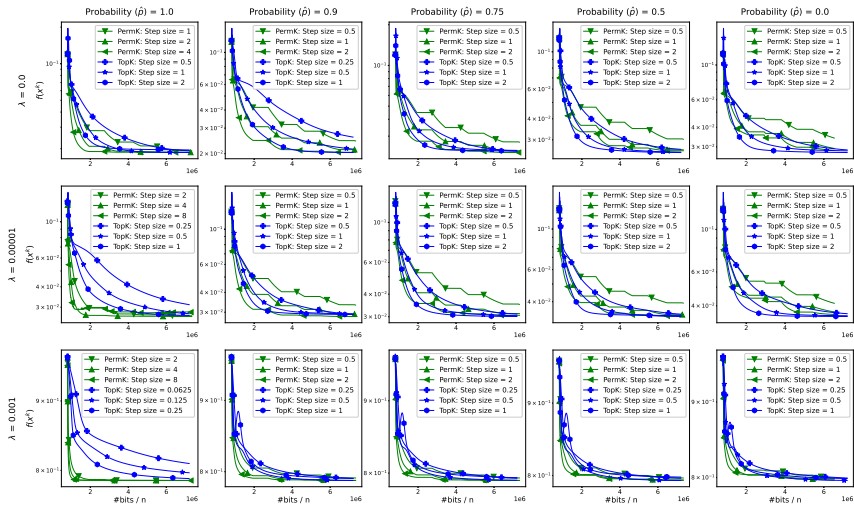

Figure 10: Comparison of Top$K$ and Perm$K$ on the encoding learning task for the MNIST dataset. Each row corresponds to a fixed regularization parameter $\lambda$; each column corresponds to a fixed probability $\widehat{p}$. In the legends, we provide compressor names and fine-tuned step sizes. Axis $x$ represents the number of bits that every node has sent.

Table 5: The "adaptive" ratio $\left(L_+^t\right)^2 / \left(L_\pm^t\right)^2$ calculated at points $x^t$ from experiments with MARINA and Perm$K$ on MNIST (see Figure 2).

|  |  | $t :=$ Iteration number | | | |
| --- | --- | --- | --- | --- | --- |
|  |  | 10 | 300 | 600 | 900 |
| $\left(L_+^t\right)^2 / \left(L_\pm^t\right)^2$ | Homogeneous ($p = 0.9$) | 104.73 | 9.96 | 5.69 | 3.61 |
| $\left(L_+^t\right)^2 / \left(L_\pm^t\right)^2$ | Heterogeneous ($p = 0.0$) | 18.83 | 2.31 | 1.69 | 1.30 |

## A.8 BEHAVIOR OF THE HESSIAN VARIANCE IN EXPERIMENTS

Now, we investigate the behavior of the Hessian Variance $L_\pm$ in practical machine learning tasks and compare it with $L_+$. Our goal is to show that $L_\pm$ can be much smaller than $L_+$ in some tasks.

### A.8.1 LINEAR REGRESSION WITH 2 NODES

Let us consider the linear regression problem distributed between 2 machines:

$$\min_{w \in \mathbb{R}^d} \left[ f(w) := \tfrac{1}{2}||\boldsymbol{X}_1 w - y_1||^2 + \tfrac{1}{2}||\boldsymbol{X}_2 w - y_2||^2 \right],$$

where $(\boldsymbol{X}_1, y_1)$ and $(\boldsymbol{X}_2, y_2)$ refer to a random split of the "mg" dataset from LIBSVM ($1,385$ samples and $d = 6$ features). Using the results from Example 3, we get $L_+ = 3609.6653$ while $L_\pm = 34.58$. Clearly, $L_\pm$ two orders smaller than $L_+$.

### A.8.2 LOCAL HESSIAN VARAINCE IN MNIST EXPERIMENTS

Now, we estimate $L_\pm$ and $L_+$ in the following way. While MARINA running, we capture points $x^t$ and calculate the "adaptive" $L_\pm^t$ and $L_+^t$, that satisfy

$$\frac{1}{n} \sum_{i=1}^{n} ||\nabla f_i(x^t) - \nabla f_i(x^{t+1})||^2 - ||\nabla f(x^t) - \nabla f(x^{t+1})||^2 = \left(L_\pm^t\right)^2 ||x^t - x^{t+1}||^2$$

and

$$\frac{1}{n} \sum_{i=1}^{n} ||\nabla f_i(x^t) - \nabla f_i(x^{t+1})||^2 = \left(L_+^t\right)^2 ||x^t - x^{t+1}||^2.$$

In experiments from Section 5.2 (see also Figure 2), we evaluate $\left(L_+^t\right)^2 / \left(L_\pm^t\right)^2$ in MARINA with Perm$K$. Results are presented in Table 5. We see that in the homogeneous case, $\left(L_\pm^t\right)^2$ is about $10\times$ times smaller than $\left(L_+^t\right)^2$ (on average), and $\left(L_+^t\right)^2$ is much larger at the start of the process. On the other hand, in the heterogeneous case, the difference is smaller. The above experiments provide further numerical justification that $L_\pm^2$ can be effectively much smaller than $L_+^2$, and that this drives the improvement of our new Perm$K$ compressor.

## B MARINA ALGORITHM

To the best of our knowledge, the state-of-the-art method for solving the nonconvex problem (1) in terms of the theoretical communication efficiency is MARINA (Gorbunov et al., 2021) (see Algorithm 1). In its simplest variant, MARINA performs iterations of the form

$$x^{k+1} = x^k - \gamma g^k, \qquad g^k = \frac{1}{n} \sum_{i=1}^{n} g_i^k, \tag{8}$$

where $g_i^k$ is a carefully designed *biased* estimator of the gradient $\nabla f_i(x^k)$, and $\gamma > 0$ is a learning rate. The gradient estimators used in MARINA are initialized to the full gradients, i.e., $g_i^0 = \nabla f_i(x^0)$, for $i \in \{1, \ldots, n\}$, and subsequently updated as

$$g_i^{k+1} = \begin{cases} \nabla f_i(x^{k+1}) & \text{if} \quad \theta_k = 1 \\ g^k + \mathcal{C}_i^k(\nabla f_i(x^{k+1}) - \nabla f_i(x^k)) & \text{if} \quad \theta_k = 0 \end{cases},$$

where $\theta_k$ is a Bernoulli random variable sampled at iteration $k$ (equal to 1 with probability $p \in (0, 1]$, and equal to 0 with probability $1 - p$), and $\mathcal{C}_i : \mathbb{R}^d \to \mathbb{R}^d$ is a randomized compression operator sampled at iteration $k$ on node $i$ *independently* from other nodes. In particular, Gorbunov et al. (2021) assume that the compression operators $\mathcal{C}_i$ are *unbiased*, and that their variance is proportional to squared norm of the input vector:

$$\mathrm{E}\left[\mathcal{C}_i(x)\right] = x, \qquad \mathrm{E}\left[\|\mathcal{C}_i(x) - x\|^2\right] \leq \omega_i \|x\|^2, \qquad \forall x \in \mathbb{R}^d.$$

In each iteration of MARINA, the gradient estimator is reset to the true gradient with (small) probability $p$. Otherwise, each worker $i$ compresses the difference of the last two local gradients, and communicates the compressed message

$$m_i^k = \mathcal{C}_i^k(\nabla f_i(x^{k+1}) - \nabla f_i(x^k))$$

to the server. These messages are then aggregated by the server to form the new gradient estimator via

$$g^{k+1} = g^k + \frac{1}{n} \sum_{i=1}^{n} m_i^k.$$

Note that i) this preserves the second relation in (8), ii) the server *can* compute $g^{k+1}$ since it has access to $g^k$, which is the case (via a recursive argument) if $g^0$ is known by the server at the start of the iterative process[8].

Further, note that the *expected communication cost* in each iteration of MARINA is equal to

$$\mathrm{Comm} = pd + (1 - p)\zeta, \qquad \zeta = \max_i \zeta_i,$$

where $d$ is the cost of communicating a (possibly dense) vector in $\mathbb{R}^d$, and $\zeta_i \leq d$ is the expected cost of communicating a vector compressed by $\mathcal{C}_i$.

MARINA one of the very few examples in stochastic optimization where the use of a biased estimator leads to a better theoretical complexity than the use of an unbiased estimator, with the other example being optimal SGD methods for single-node problems SARAH (Nguyen et al., 2017), SPIDER (Fang et al., 2018), PAGE (Li et al., 2021).

---

[8]This is done by each worker sending the full gradient $g_i^0 = \nabla f_i(x^0)$ to the server at initialization.

## C  MISSING PROOFS

### C.1  PROOF OF LEMMA 1

**Lemma 1.** *If* $\mathcal{C}_i \in \mathbb{U}(\omega_i)$ *for* $i \in \{1, 2, \ldots, n\}$, *then* $\{\mathcal{C}_i\}_{i=1}^n \in \mathbb{U}(\max_i \omega_i, 0)$. *If we further assume that the compressors are independent, then* $\{\mathcal{C}_i\}_{i=1}^n \in \mathbb{U}(\frac{1}{n} \max_i \omega_i, 0)$.

*Proof.* Let us first assume unbiasedness only. By Jensen's inequality,

$$\left\| \frac{1}{n} \sum_{i=1}^n \mathcal{C}_i(a_i) - \frac{1}{n} \sum_{i=1}^n a_i \right\|^2 \leq \frac{1}{n} \sum_{i=1}^n \| \mathcal{C}_i(a_i) - a_i \|^2 .$$

It remains to apply expectation on both sides and then use inequality

$$\mathrm{E}\left[ \| \mathcal{C}_i(a_i) - a_i \|^2 \right] \leq \omega_i \| a_i \|^2 , \forall i \in \{1, \ldots, n\},$$

to conclude that $\{\mathcal{C}_i\}_{i=1}^n \in \mathbb{U}(\max_i \omega_i, 0)$.

Let us now add the assumption of independence.

$$\mathrm{E}\left[ \left\| \frac{1}{n} \sum_{i=1}^n \mathcal{C}_i(a_i) - \frac{1}{n} \sum_{i=1}^n a_i \right\|^2 \right]$$

$$= \mathrm{E}\left[ \left\| \frac{1}{n} \sum_{i=1}^n (\mathcal{C}_i(a_i) - a_i) \right\|^2 \right]$$

$$= \frac{1}{n^2} \sum_{i=1}^n \mathrm{E}\left[ \| \mathcal{C}_i(a_i) - a_i \|^2 \right] + \frac{1}{n^2} \sum_{i \neq j} \mathrm{E}\left[ \langle \mathcal{C}_i(a_i) - a_i, \mathcal{C}_j(a_j) - a_j \rangle \right]$$

$$\leq \frac{\max_i \omega_i}{n^2} \sum_{i=1}^n \| a_i \|^2 ,$$

by independence, thus, $A = \max_i \omega_i / n$, $B = 0$. $\qquad\square$

### C.2  PROOF OF LEMMA 2

**Lemma 2.** $L_- \leq L_+$, $L_- \leq \frac{1}{n} \sum_{i=1}^n L_i$, $L_+^2 \leq \frac{1}{n} \sum_{i=1}^n L_i^2$, *and* $L_+^2 - L_-^2 \leq L_\pm^2 \leq L_+^2$.

*Proof.* Let us define

$$\mathcal{L}_-(x, y) \quad := \quad \| \nabla f(x) - \nabla f(y) \|^2 ,$$

$$\mathcal{L}_+(x, y) \quad := \quad \frac{1}{n} \sum_{i=1}^n \| \nabla f_i(x) - \nabla f_i(y) \|^2 ,$$

$$\mathcal{L}_\pm(x, y) \quad := \quad \mathcal{L}_+(x, y) - \mathcal{L}_-(x, y).$$

The inequalities are now established as follows:

1. By Jensen's inequality and the definition of $L_+$,

$$\mathcal{L}_-(x, y) \leq \mathcal{L}_+(x, y) \leq L_+^2 \| x - y \|^2 ,$$

   thus, $L_-$ is at most $L_+$.

2. By the triangle inequality, we have

$$\| \nabla f(x) - \nabla f(y) \| \leq \frac{1}{n} \sum_{i=1}^n \| \nabla f_i(x) - \nabla f_i(y) \| \leq \frac{1}{n} \sum_{i=1}^n L_i \| x - y \| ,$$

   thus $L_-$ is at most $\frac{1}{n} \sum_{i=1}^n L_i$.

3. From the definition of $L_i$, we have

$$\mathcal{L}_+(x, y) = \frac{1}{n} \sum_{i=1}^{n} \|\nabla f_i(x) - \nabla f_i(y)\|^2 \le \frac{1}{n} \sum_{i=1}^{n} L_i^2 \|x - y\|^2,$$

and $L_+^2$ is at most $\frac{1}{n} \sum_{i=1}^{n} L_i^2$.

4. The right inequality follows from $\mathcal{L}_-(x, y) \ge 0$ and

$$\mathcal{L}_\pm(x, y) \le \mathcal{L}_+(x, y) \le L_+^2 \|x - y\|^2.$$

Now, we prove the left inequality. From the definition of $L_\pm$, we have

$$\mathcal{L}_\pm(x, y) \le L_\pm^2 \|x - y\|^2,$$

and

$$\mathcal{L}_\pm(x, y) = \mathcal{L}_+(x, y) - \mathcal{L}_-(x, y),$$

hence,

$$\mathcal{L}_+(x, y) \le L_\pm^2 \|x - y\|^2 + \mathcal{L}_-(x, y) \le (L_-^2 + L_\pm^2) \|x - y\|^2,$$

thus $L_+^2 \le L_-^2 + L_\pm^2$.

$\square$

## C.3 Proof of Theorem 1

**Theorem 1.** *The PermK compressors from Definition 2 are unbiased and belong to $\mathbb{IV}(1)$.*

*Proof.*

We fix any $x \in \mathbb{R}^d$ and prove unbiasedness:

$$\mathrm{E}\left[\mathcal{C}_i(x)\right] = n \sum_{j=q(i-1)+1}^{qi} \mathrm{E}\left[x_{\pi_j} e_{\pi_j}\right] = n \left( \sum_{j=q(i-1)+1}^{qi} \frac{1}{d} \sum_{i=1}^{d} x_i e_i \right) = \frac{nq}{d} x = x.$$

Next, we find the second moment:

$$\mathrm{E}\left[\|\mathcal{C}_i(x)\|^2\right] = n^2 \sum_{j=q(i-1)+1}^{qi} \mathrm{E}\left[|x_{\pi_j}|^2\right] = n^2 \sum_{j=q(i-1)+1}^{qi} \frac{1}{d} \sum_{i=1}^{d} |x_i|^2 = n^2 \frac{q}{d} \|x\|^2 = n \|x\|^2.$$

For all $a_1, \ldots, a_n \in \mathbb{R}^d$, the following inequality holds:

$$
\begin{aligned}
\mathrm{E}\left[\left\|\frac{1}{n} \sum_{i=1}^{n} \mathcal{C}_i(a_i)\right\|^2\right] &= \frac{1}{n^2} \sum_{i=1}^{n} \mathrm{E}\left[\|\mathcal{C}_i(a_i)\|^2\right] + \sum_{i \ne j} \mathrm{E}\left[\langle \mathcal{C}_i(a_i), \mathcal{C}_j(a_j)\rangle\right] \\
&= \frac{1}{n^2} \sum_{i=1}^{n} \mathrm{E}\left[\|\mathcal{C}_i(a_i)\|^2\right] \\
&= \frac{1}{n} \sum_{i=1}^{n} \|a_i\|^2.
\end{aligned}
$$

Hence, Assumption 4 is fulfilled with $A = B = 1$.

$\square$

## C.4 PROOF OF THEOREM 2

**Theorem 2.** *The PermK compressors from Definition 3 are unbiased and belong to $\mathbb{IV}(A)$ with $A = 1 - \frac{n-d}{n-1}$.*

*Proof.*

We fix any $x \in \mathbb{R}^d$ and prove unbiasedness:

$$\mathrm{E}\left[\mathcal{C}_i(x)\right] = d\mathrm{E}\left[x_{\pi_i} e_{\pi_i}\right] = d\frac{1}{d}\sum_{i=1}^{d} x_i e_i = x.$$

Next, we find the second moment:

$$\mathrm{E}\left[\|\mathcal{C}_i(x)\|^2\right] = \frac{1}{d}\sum_{i=1}^{d} d^2 |x_i|^2 = d\|x\|^2.$$

For all $i \neq j \in \{1, 2, \ldots, n\}, x, y \in \mathbb{R}^d$, we have

$$
\begin{aligned}
\mathrm{E}\left[\langle\mathcal{C}_i(x), \mathcal{C}_j(y)\rangle\right] &= \mathrm{E}\left[\langle\mathcal{C}_i(x), \mathcal{C}_j(y)\rangle \mid \pi_i = \pi_j\right]\mathbf{Prob}\left(\pi_i = \pi_j\right) \\
&= \frac{(q-1)}{(n-1)d}\sum_{q=1}^{d}\mathrm{E}\left[\langle\mathcal{C}_i(x), \mathcal{C}_j(y)\rangle \mid \pi_i = q, \pi_j = q\right] \\
&= \frac{(q-1)}{(n-1)d}\sum_{q=1}^{d} d^2 x_q y_q \\
&= \frac{(q-1)d}{n-1}\langle x, y\rangle.
\end{aligned}
$$

For all $a_1, \ldots, a_n \in \mathbb{R}^d$, the following inequality holds:

$$
\begin{aligned}
\mathrm{E}\left[\left\|\frac{1}{n}\sum_{i=1}^{n}\mathcal{C}_i(a_i)\right\|^2\right] &= \frac{1}{n^2}\sum_{i=1}^{n}\mathrm{E}\left[\|\mathcal{C}_i(a_i)\|^2\right] + \frac{1}{n^2}\sum_{i\neq j}\mathrm{E}\left[\langle C_i(a_i), C_j(a_j)\rangle\right] \\
&= \frac{d}{n^2}\sum_{i=1}^{n}\|a_i\|^2 + \frac{1}{n^2}\sum_{i\neq j}\mathrm{E}\left[\langle\mathcal{C}_i(a_i), \mathcal{C}_j(a_j)\rangle\right] \\
&= \frac{d}{n^2}\sum_{i=1}^{n}\|a_i\|^2 + \frac{(q-1)d}{n^2(n-1)}\sum_{i\neq j}\langle a_i, a_j\rangle \\
&= \left(\frac{d}{n} - \frac{(q-1)d}{n(n-1)}\right)\frac{1}{n}\sum_{i=1}^{n}\|a_i\|^2 + \frac{(q-1)d}{n-1}\left\|\frac{1}{n}\sum_{i=1}^{n}a_i\right\|^2 \\
&= \left(1 - \frac{n-d}{n-1}\right)\frac{1}{n}\sum_{i=1}^{n}\|a_i\|^2 + \frac{n-d}{n-1}\left\|\frac{1}{n}\sum_{i=1}^{n}a_i\right\|^2.
\end{aligned}
$$

Hence, Assumption 4 is fulfilled with $A = B = 1 - \frac{n-d}{n-1}$. $\qquad\square$

## C.5 PROOF OF THEOREM 3

**Theorem 3.** *Assume that for each $i \in \{1, 2, \ldots, n\}$, the function $f_i$ is twice continuously differentiable. Fix any $x, y \in \mathbb{R}^d$ and define $\boldsymbol{H}_i(x, y) := \int_0^1 \nabla^2 f_i(x + t(y - x))\, dt$, $\boldsymbol{H}(x, y) := \frac{1}{n}\sum_{i=1}^{n}\boldsymbol{H}_i(x, y)$. Then the matrices $\boldsymbol{L}_i(x, y) := \boldsymbol{H}_i^2(x, y)$, $\boldsymbol{L}_-(x, y) := \boldsymbol{H}^2(x, y)$, $\boldsymbol{L}_+(x, y) :=$*

$\frac{1}{n}\sum_{i=1}^{n}\boldsymbol{H}_i^2(x,y)$ and $\boldsymbol{L}_{\pm}(x,y) := \boldsymbol{L}_+(x,y) - \boldsymbol{L}_-(x,y)$ are symmetric and positive semidefinite. Moreover,

$$L_i^2 = \sup_{x,y\in\mathbb{R}^d,x\neq y}\frac{(x-y)^\top \boldsymbol{L}_i(x,y)(x-y)}{\|x-y\|^2}, \quad L_-^2 = \sup_{x,y\in\mathbb{R}^d,x\neq y}\frac{(x-y)^\top \boldsymbol{L}_-(x,y)(x-y)}{\|x-y\|^2},$$

$$L_+^2 = \sup_{x,y\in\mathbb{R}^d,x\neq y}\frac{(x-y)^\top \boldsymbol{L}_+(x,y)(x-y)}{\|x-y\|^2}, \quad L_\pm^2 = \sup_{x,y\in\mathbb{R}^d,x\neq y}\frac{(x-y)^\top \boldsymbol{L}_\pm(x,y)(x-y)}{\|x-y\|^2}.$$

*Proof.* The fundamental theorem of calculus says that for any continuously differentiable function $\psi : \mathbb{R} \to \mathbb{R}$ we have

$$\psi(1) - \psi(0) = \int_0^1 \psi'(t)dt.$$

Choose $i \in \{1,2,\ldots,n\}$, $j \in \{1,2,\ldots,d\}$, distinct vectors $x, y \in \mathbb{R}^d$, and let

$$\psi_{ij}(t) := \langle \nabla f_i(x + t(y-x)), e_j \rangle,$$

where $e_j \in \mathbb{R}^d$ is the $j$th standard unit basis vector. Since $f_i$ is twice continuously differentiable, $\psi_{ij}$ is continuously differentiable, and by the chain rule,

$$\psi_{ij}'(t) = \langle \nabla^2 f_i(x + t(y-x))(y-x), e_j \rangle.$$

Applying the fundamental theorem of calculus, we get

$$\psi_{ij}(1) - \psi_{ij}(0) = \int_0^1 \langle \nabla^2 f_i(x + t(y-x))(y-x), e_j \rangle dt. \tag{9}$$

Let $\psi_i : \mathbb{R} \to \mathbb{R}^d$ be defined by $\psi_i(t) := \nabla f_i(x + t(y-x)) = (\psi_{i1}(t),\ldots,\psi_{id}(t))$. Combining equations 9 for $j = 1,2,\ldots,d$ into a vector form using the fact that

$$\int_0^1 \langle \nabla^2 f_i(x + t(y-x))(y-x), e_j \rangle dt = \left\langle \left(\int_0^1 \nabla^2 f_i(x + t(y-x))dt\right)(y-x), e_j \right\rangle$$

we arrive at the identity

$$
\begin{aligned}
\nabla f_i(y) - \nabla f_i(x) &= \psi_i(1) - \psi_i(0) \\
&\stackrel{(9)}{=} \left(\int_0^1 \nabla^2 f_i(x + t(y-x))dt\right)(y-x) \\
&= \boldsymbol{H}_i(x,y)(y-x). \tag{10}
\end{aligned}
$$

Next, since $\nabla^2 f_i(x + t(y-x))$ is symmetric for all $t$, so is $\boldsymbol{H}_i(x,y)$, and hence $\boldsymbol{L}_i(x,y) := \boldsymbol{H}_i^2(x,y) = \boldsymbol{H}_i^\top(x,y)\boldsymbol{H}_i(x,y)$, which also means that $\boldsymbol{L}_i(x,y)$ is symmetric and positive semidefinite. Combining these observations, we obtain

$$\|\nabla f_i(x) - \nabla f_i(y)\|^2 \stackrel{(10)}{=} (x-y)^\top \boldsymbol{L}_i(x,y)(x-y). \tag{11}$$

Clearly,

$$L_i^2 = \sup_{x,y\in\mathbb{R}^d,x\neq y}\frac{\|\nabla f_i(x) - \nabla f_i(y)\|^2}{\|x-y\|^2} \stackrel{(11)}{=} \sup_{x,y\in\mathbb{R}^d,x\neq y}\frac{(x-y)^\top \boldsymbol{L}_i(x,y)(x-y)}{\|x-y\|^2}.$$

Using the same reasoning, we have $\nabla f(y) - \nabla f(x) = \boldsymbol{H}(x,y)(y-x)$, and

$$L_-^2 = \sup_{x,y\in\mathbb{R}^d,x\neq y}\frac{\|\nabla f(x) - \nabla f(y)\|^2}{\|x-y\|^2} = \sup_{x,y\in\mathbb{R}^d,x\neq y}\frac{(x-y)^\top \boldsymbol{L}_-(x,y)(x-y)}{\|x-y\|^2},$$

where $\boldsymbol{L}_-(x,y) := \boldsymbol{H}^2(x,y) = \boldsymbol{H}^\top(x,y)\boldsymbol{H}(x,y)$ is symmetric and positive semidefinite, since $\boldsymbol{H}_i(x,y)$ are symmetric and positive semidefinite. Finally,

$$
\begin{aligned}
L_+^2 &= \sup_{x,y\in\mathbb{R}^d,x\neq y}\frac{\frac{1}{n}\sum_{i=1}^{n}\|\nabla f_i(x) - \nabla f_i(y)\|^2}{\|x-y\|^2} \\
&= \sup_{x,y\in\mathbb{R}^d,x\neq y}\frac{(x-y)^\top \left(\frac{1}{n}\sum_{i=1}^{n}\boldsymbol{H}_i^2(x,y)\right)(x-y)}{\|x-y\|^2} \\
&= \sup_{x,y\in\mathbb{R}^d,x\neq y}\frac{(x-y)^\top \boldsymbol{L}_+(x,y)(x-y)}{\|x-y\|^2},
\end{aligned}
$$

and

$$
\begin{aligned}
L_\pm^2 &= \sup_{x,y\in\mathbb{R}^d, x\neq y} \frac{\frac{1}{n}\sum_{i=1}^n \|\nabla f_i(x) - \nabla f_i(y)\|^2 - \|\nabla f(x) - \nabla f(y)\|^2}{\|x-y\|^2} \\
&= \sup_{x,y\in\mathbb{R}^d, x\neq y} \frac{(x-y)^\top \left(\frac{1}{n}\sum_{i=1}^n \boldsymbol{H}_i^2(x,y) - \boldsymbol{H}^2(x,y)\right)(x-y)}{\|x-y\|^2} \\
&= \sup_{x,y\in\mathbb{R}^d, x\neq y} \frac{(x-y)^\top \boldsymbol{L}_\pm(x,y)(x-y)}{\|x-y\|^2}.
\end{aligned}
$$

Note, that $\boldsymbol{L}_+(x,y)$ inherits symmetry and positive semidefiniteness from $\boldsymbol{H}_i^2(x,y)$. Symmetry of $\boldsymbol{L}_\pm(x,y)$ is trivial. To prove positive semidefiniteness of $\boldsymbol{L}_\pm(x,y)$, note that

$$
\begin{aligned}
\boldsymbol{L}_\pm(x,y) &= \frac{1}{n}\sum_{i=1}^n \boldsymbol{H}_i^2(x,y) - \boldsymbol{H}^2(x,y) \\
&= \frac{1}{n}\sum_{i=1}^n (\boldsymbol{H}_i(x,y) - \boldsymbol{H}(x,y) + \boldsymbol{H}(x,y))^2 - \boldsymbol{H}^2(x,y) \\
&= \frac{1}{n}\sum_{i=1}^n (\boldsymbol{H}_i(x,y) - \boldsymbol{H}(x,y))^2 + \frac{1}{n}\boldsymbol{H}(x,y)\sum_{i=1}^n (\boldsymbol{H}_i(x,y) - \boldsymbol{H}(x,y)) \\
&\quad + \frac{1}{n}\sum_{i=1}^n (\boldsymbol{H}_i(x,y) - \boldsymbol{H}(x,y))\,\boldsymbol{H}(x,y) \\
&= \frac{1}{n}\sum_{i=1}^n (\boldsymbol{H}_i(x,y) - \boldsymbol{H}(x,y))^2,
\end{aligned}
$$

which is positive semidefinite. $\qquad\square$

### C.6 PROOF OF THEOREM 4

**Theorem 4.** *Let Assumptions 1, 2, 3 and 4 be satisfied. Let the stepsize in* MARINA *be chosen as* $0 < \gamma \leq \frac{1}{M}$*, where* $M = L_- + \sqrt{\frac{1-p}{p}\left((A-B)L_+^2 + BL_\pm^2\right)}$*. Then after* $T$ *iterations,* MARINA *finds a random point* $\hat{x}^T \in \mathbb{R}^d$ *for which*

$$
\mathrm{E}\left[\|\nabla f(\hat{x}^T)\|^2\right] \leq \frac{2\Delta^0}{\gamma T}.
$$

*Proof.* In the proof, we follow closely the analysis of Gorbunov et al. (2021) and adapt it to utilize the power of Hessian variance (Definition 4) and AB assumption (Assumption 4). We bound the term $\mathrm{E}\left[\|g^{t+1} - \nabla f(x^{t+1})\|^2\right]$ in a similar fashion to Gorbunov et al. (2021), but make use of the AB assumption. Other steps are essentially identical, but refine the existing analysis through Hessian variance.

First, we recall the following lemmas.

**Lemma 3** (Li et al. (2021)). *Suppose that* $L_-$ *is finite and let* $x^{t+1} = x^t - \gamma g^t$*. Then for any* $g^t \in \mathbb{R}^d$ *and* $\gamma > 0$*, we have*

$$
f(x^{t+1}) \leq f(x^t) - \frac{\gamma}{2}\|\nabla f(x^t)\|^2 - \left(\frac{1}{2\gamma} - \frac{L_-}{2}\right)\|x^{t+1} - x^t\|^2 + \frac{\gamma}{2}\|g^t - \nabla f(x^t)\|^2. \quad (12)
$$

**Lemma 4** (Richtárik et al. (2021)). *Let* $a, b > 0$*. If* $0 \leq \gamma \leq \frac{1}{\sqrt{a}+b}$*, then* $a\gamma^2 + b\gamma \leq 1$*. Moreover, the bound is tight up to the factor of 2 since* $\frac{1}{\sqrt{a}+b} \leq \min\left\{\frac{1}{\sqrt{a}}, \frac{1}{b}\right\} \leq \frac{2}{\sqrt{a}+b}$

Next, we get an upper bound of $\mathrm{E}\left[\|g^{t+1} - \nabla f(x^{t+1})\|^2 \mid x^{t+1}\right]$.

**Lemma 5.** *Let us consider $g^{t+1}$ from Line 8 of Algorithm 1 and assume, that Assumptions 2, 3 and 4 hold, then*

$$
\begin{aligned}
\mathrm{E}\left[\left\|g^{t+1} - \nabla f(x^{t+1})\right\|^2 \Big| x^{t+1}\right] \leq \ & (1-p)\left((A-B)L_+^2 + BL_\pm^2\right)\left\|x^{t+1} - x^t\right\|^2 \\
& + (1-p)\left\|g^t - \nabla f(x^t)\right\|^2.
\end{aligned} \tag{13}
$$

*Proof.* In the view of definition of $g^{t+1}$, we get

$$
\begin{aligned}
& \mathrm{E}\left[\left\|g^{t+1} - \nabla f(x^{t+1})\right\|^2 \Big| x^{t+1}\right] \\
& = (1-p)\mathrm{E}\left[\left\|g^t + \frac{1}{n}\sum_{i=1}^n \mathcal{C}_i\left(\nabla f_i(x^{t+1}) - \nabla f_i(x^t)\right) - \nabla f(x^{t+1})\right\|^2 \Bigg| x^{t+1}\right] \\
& = (1-p)\mathrm{E}\left[\left\|\frac{1}{n}\sum_{i=1}^n \mathcal{C}_i\left(\nabla f_i(x^{t+1}) - \nabla f_i(x^t)\right) - \nabla f(x^{t+1}) + \nabla f(x^t)\right\|^2 \Bigg| x^{t+1}\right] \\
& \quad + (1-p)\left\|g^t - \nabla f(x^t)\right\|^2.
\end{aligned}
$$

In the last equality we used unbiasedness of $\mathcal{C}_i$. Next, from AB inequality, we have

$$
\begin{aligned}
& \mathrm{E}\left[\left\|g^{t+1} - \nabla f(x^{t+1})\right\|^2 \Big| x^{t+1}\right] \\
& \leq (1-p)\mathrm{E}\left[\left\|\frac{1}{n}\sum_{i=1}^n \mathcal{C}_i\left(\nabla f_i(x^{t+1}) - \nabla f_i(x^t)\right) - \nabla f(x^{t+1}) + \nabla f(x^t)\right\|^2 \Bigg| x^{t+1}\right] \\
& \quad + (1-p)\left\|g^t - \nabla f(x^t)\right\|^2. \\
& \leq (1-p)\left(A\left(\frac{1}{n}\sum_{i=1}^n \left\|\nabla f_i(x^{t+1}) - \nabla f_i(x^t)\right\|^2\right) - B\left\|\nabla f(x^{t+1}) - \nabla f(x^t)\right\|^2\right) \\
& \quad + (1-p)\left\|g^t - \nabla f(x^t)\right\|^2 \\
& = (1-p)\Bigg((A-B)\left(\frac{1}{n}\sum_{i=1}^n \left\|\nabla f_i(x^{t+1}) - \nabla f_i(x^t)\right\|^2\right) \\
& \quad + B\left(\frac{1}{n}\sum_{i=1}^n \left\|\nabla f_i(x^{t+1}) - \nabla f_i(x^t)\right\|^2 - \left\|\nabla f(x^{t+1}) - \nabla f(x^t)\right\|^2\right)\Bigg) \\
& \quad + (1-p)\left\|g^t - \nabla f(x^t)\right\|^2.
\end{aligned}
$$

Using Assumption 2 and Definition 4, we obtain (13). $\qquad\square$

We are ready to prove Theorem 4. Defining

$$
\begin{aligned}
\Phi^t &:= f(x^t) - f^{\mathrm{inf}} + \frac{\gamma}{2p}\left\|g^t - \nabla f(x^t)\right\|^2, \\
\widehat{L}^2 &:= (A-B)L_+^2 + BL_\pm^2,
\end{aligned}
$$

and using inequalities (12) and (13), we get

$$
\begin{aligned}
\mathrm{E}\left[\Phi^{t+1}\right] \\
&\leq \mathrm{E}\left[f(x^t) - f^{\mathrm{inf}} - \frac{\gamma}{2}\left\|\nabla f(x^t)\right\|^2 - \left(\frac{1}{2\gamma} - \frac{L_-}{2}\right)\left\|x^{t+1} - x^t\right\|^2 + \frac{\gamma}{2}\left\|g^t - \nabla f(x^t)\right\|^2\right] \\
&\quad + \frac{\gamma}{2p}\mathrm{E}\left[(1-p)\widehat{L}^2\left\|x^{t+1} - x^t\right\|^2 + (1-p)\left\|g^t - \nabla f(x^t)\right\|^2\right] \\
&= \mathrm{E}\left[\Phi^t\right] - \frac{\gamma}{2}\mathrm{E}\left[\left\|\nabla f(x^t)\right\|^2\right] \\
&\quad + \left(\frac{\gamma(1-p)\widehat{L}^2}{2p} - \frac{1}{2\gamma} + \frac{L_-}{2}\right)\mathrm{E}\left[\left\|x^{t+1} - x^t\right\|^2\right] \\
&\leq \mathrm{E}\left[\Phi^t\right] - \frac{\gamma}{2}\mathrm{E}\left[\left\|\nabla f(x^t)\right\|^2\right]
\end{aligned}
$$

where in the last inequality we use

$$
\frac{\gamma(1-p)\widehat{L}^2}{2p} - \frac{1}{2\gamma} + \frac{L}{2} \leq 0
$$

following from the stepsize choice and Lemma 4.

Summing up inequalities $\mathrm{E}\left[\Phi^{t+1}\right] \leq \mathrm{E}\left[\Phi^t\right] - \frac{\gamma}{2}\mathrm{E}\left[\left\|\nabla f(x^t)\right\|^2\right]$ for $t = 0, 1, \ldots, T - 1$ and rearranging the terms, we get

$$
\frac{1}{T}\sum_{t=0}^{T-1}\mathrm{E}\left[\left\|\nabla f(x^t)\right\|^2\right] \leq \frac{2}{\gamma T}\sum_{t=0}^{T-1}\left(\mathrm{E}\left[\Phi^t\right] - \mathrm{E}\left[\Phi^{t+1}\right]\right) = \frac{2\left(\mathrm{E}\left[\Phi^0\right] - \mathrm{E}\left[\Phi^T\right]\right)}{\gamma T} \leq \frac{2\Delta^0}{\gamma T},
$$

since $g^0 = \nabla f(x^0)$ and $\Phi^T \geq 0$. Finally, using the tower property and the definition of $\hat{x}^T$ (see Section B), we obtain the desired result. $\qquad\square$

## D    POLYAK-ŁOJASIEWICZ ANALYSIS

In this section, we analyze the algorithm under Polyak-Łojasiewicz (PŁ) condition. We show that MARINA algorithm with Assumption 5 enjoys a linear convergence rate. Now, we state the assumption and the convergence rate theorem.

**Assumption 5** (PŁ condition). *Function $f$ satisfies Polyak-Łojasiewicz (PŁ) condition, i.e.,*

$$\|\nabla f(x)\|^2 \geq 2\mu(f(x) - f^\star), \quad \forall x \in \mathbb{R}^d, \tag{14}$$

*where $\mu > 0$ and $f^\star := \inf_x f(x)$.*

**Lemma 6.** *For $L_- > 0$ and $\mu$ from Assumption 5 holds that $L_- \geq \mu$.*

**Theorem 5.** *Let Assumptions 1, 2, 3, 4 and 5 be satisfied and*

$$\gamma \leq \min\left\{ \left(L_- + \sqrt{\frac{2(1-p)}{p}\left((A-B)L_+^2 + BL_\pm^2\right)}\right)^{-1}, \frac{p}{2\mu}\right\}, \tag{15}$$

*then for $x^T$ from* MARINA *algorithm the following inequality holds:*

$$\mathrm{E}\left[f(x^T) - f^\star\right] \leq (1 - \gamma\mu)^T \Delta^0.$$

We provide the proof to Theorem 5 in Section D.2.

In Table 6, we provide communication complexity of MARINA with Perm$K$ and Rand$K$, and EF21 with Top$K$, optimized w.r.t. parameters of the methods. As in Section 4, we see that MARINA with Perm$K$ is not worse than MARINA with Rand$K$ (recall Lemma 2).

Let us consider zero Hessian variance regime: $L_\pm = 0$. When $d \geq n$, Perm$K$ compressor has communication complexity $\mathcal{O}\left(\max\left\{dL_-/n\mu, d\right\}\right)$, while Rand$K$ compressor has communication complexity $\mathcal{O}\left(\max\left\{dL_-/\sqrt{n}\mu, d\right\}\right)$. And the communication complexity of Perm$K$ is strictly better when $dL_-/\sqrt{n}\mu > d$. Moreover, if $d \leq n$ and $(1 + d/\sqrt{n})L_-/\mu > d$, then we get the strict improvement of the communication complexity from $\mathcal{O}\left(\max\left\{(1 + d/\sqrt{n})L_-/\mu, d\right\}\right)$ to $\mathcal{O}\left(\max\left\{L_-/\mu, d\right\}\right)$ over MARINA with Rand$K$.

Table 6: Optimized communication complexity of MARINA and EF21 with particular compressors under PŁ condition (up to a logarithmic factor).

| Method | Communication complexity | |
|---|---|---|
| | $d \geq n$ (Lemma 15) | $d \leq n$ (Lemma 16) |
| MARINA $\bigcap$ Perm$K$ | $\mathcal{O}\left(\max\left\{\frac{1}{\mu}\min\left\{dL_-, \frac{d}{n}L_- + \frac{d}{\sqrt{n}}L_\pm\right\}, d\right\}\right)$ | $\mathcal{O}\left(\max\left\{\frac{1}{\mu}\min\left\{dL_-, L_- + \frac{d}{\sqrt{n}}L_\pm\right\}, d\right\}\right)$ |
| MARINA $\bigcap$ Rand$K$ | $\mathcal{O}\left(\max\left\{\frac{1}{\mu}\min\left\{dL_-, \frac{d}{\sqrt{n}}L_+\right\}, d\right\}\right)$ | $\mathcal{O}\left(\max\left\{\frac{1}{\mu}\min\left\{dL_-, L_- + \frac{d}{\sqrt{n}}L_+\right\}, d\right\}\right)$ |
| EF21 $\bigcap$ Top$K$ | $\mathcal{O}\left(\frac{dL_-}{\mu}\right)$ | $\mathcal{O}\left(\frac{dL_-}{\mu}\right)$ |

### D.1    PROOF OF LEMMA 6

**Lemma 6.** *For $L_- > 0$ and $\mu$ from Assumption 5 holds that $L_- \geq \mu$.*

*Proof.* We can define $L_-$ using the following inequality:

$$f(x) \leq f(y) + \langle \nabla f(y), x - y \rangle + \frac{L_-}{2}\|x - y\|^2, \quad \forall x, y, \in \mathbb{R}^d.$$

Let us take $x = y - 1/L_-\nabla f(y)$. Then,

$$f\left(y - \frac{1}{L_-}\nabla f(y)\right) \leq f(y) - \frac{1}{2L_-}\|\nabla f(y)\|^2, \quad \forall y \in \mathbb{R}^d.$$

Rearranging the terms and using Definition 14, we have

$$\frac{1}{2L_-}\|\nabla f(y)\|^2 \leq f(y) - f\left(y - \frac{1}{L_-}\nabla f(y)\right) \leq f(y) - f^{\inf} \leq \frac{1}{2\mu}\|\nabla f(y)\|^2,$$

thus, $\mu \leq L_-$. $\qquad\square$

## D.2 PROOF OF THEOREM 5

**Theorem 5.** *Let Assumptions 1, 2, 3, 4 and 5 be satisfied and*

$$\gamma \le \min \left\{ \left( L_- + \sqrt{\frac{2(1-p)}{p}\left((A-B)L_+^2 + BL_\pm^2\right)} \right)^{-1}, \frac{p}{2\mu} \right\}, \tag{15}$$

*then for $x^T$ from* MARINA *algorithm the following inequality holds:*

$$\mathrm{E}\left[f(x^T) - f^\star\right] \le (1 - \gamma\mu)^T \Delta^0.$$

*Proof.* The analysis is almost the same as in Gorbunov et al. (2021), but we include it for completeness. Let us define

$$\Phi^t := f(x^t) - f^{\mathrm{inf}} + \frac{\gamma}{p}\left\|g^t - \nabla f(x^t)\right\|^2, \quad \widehat{L}^2 := \left((A-B)L_+^2 + BL_\pm^2\right).$$

As in Appendix C.6, we use (12) and (13) to get that

$$\mathrm{E}\left[\Phi^{t+1}\right]$$
$$\le \mathrm{E}\left[f(x^t) - f^{\mathrm{inf}} - \frac{\gamma}{2}\left\|\nabla f(x^t)\right\|^2 - \left(\frac{1}{2\gamma} - \frac{L_-}{2}\right)\left\|x^{t+1} - x^t\right\|^2 + \frac{\gamma}{2}\left\|g^t - \nabla f(x^t)\right\|^2\right]$$
$$\quad + \frac{\gamma}{p}\mathrm{E}\left[(1-p)\widehat{L}^2\left\|x^{t+1} - x^t\right\|^2 + (1-p)\left\|g^t - \nabla f(x^t)\right\|^2\right]$$
$$\overset{(14)}{\le} \mathrm{E}\left[(1-\gamma\mu)(f(x^t) - f^{\mathrm{inf}}) - \left(\frac{1}{2\gamma} - \frac{L_-}{2}\right)\left\|x^{t+1} - x^t\right\|^2 + \frac{\gamma}{2}\left\|g^t - \nabla f(x^t)\right\|^2\right]$$
$$\quad + \frac{\gamma}{p}\mathrm{E}\left[(1-p)\widehat{L}^2\left\|x^{t+1} - x^t\right\|^2 + (1-p)\left\|g^t - \nabla f(x^t)\right\|^2\right]$$
$$= \mathrm{E}\left[(1-\gamma\mu)(f(x^t) - f^{\mathrm{inf}}) + \left(\frac{\gamma}{2} + \frac{\gamma}{p}(1-p)\right)\left\|g^t - \nabla f(x^t)\right\|^2\right]$$
$$\quad + \mathrm{E}\left[\left(\frac{\gamma}{p}(1-p)\widehat{L}^2 - \frac{1}{2\gamma} + \frac{L_-}{2}\right)\left\|x^{t+1} - x^t\right\|^2\right]$$
$$\le (1-\gamma\mu)\mathrm{E}\left[\Phi^t\right].$$

In the last inequality, we used $\frac{\gamma}{p}(1-p)\widehat{L}^2 - \frac{1}{2\gamma} + \frac{L_-}{2} \le 0$ and $\frac{\gamma}{2} + \frac{\gamma}{p}(1-p) \le (1-\gamma\mu)\frac{\gamma}{p}$, that follow from (15) and Lemma 4. Unrolling $\mathrm{E}\left[\Phi^{t+1}\right] \le (1-\gamma\mu)\mathrm{E}\left[\Phi^t\right]$ and using $g^0 = \nabla f(x^0)$, we have

$$\mathrm{E}\left[f(x^T) - f^{\mathrm{inf}}\right] \le \mathrm{E}\left[\Phi^T\right] \le (1-\gamma\mu)^T \Phi^0 = (1-\gamma\mu)^T\left(f(x^0) - f^{\mathrm{inf}}\right).$$

This concludes the proof. □

# E  EF21 ANALYSIS

We provide convergence proofs of EF21 algorithm from Richtárik et al. (2021) for non-convex and PŁ regimes. They will be almost identical to the one by Richtárik et al. (2021) (indeed, the only change is the constant $L_+$ instead of $\widetilde{L}$), but we have decided to include it for the sake of clarity.

## E.1  EF21 RATE IN THE NON-CONVEX REGIME

We will be using the following lemmas, the proofs of which are in their corresponding papers.

**Lemma 7** (Richtárik et al. (2021)). *Let $\mathcal{C}$ to be $\alpha$-contractive for $0 < \alpha \leq 1$. Define $G_i^t := \left\|g_i^t - \nabla f_i(x^t)\right\|^2$ and $W^t := \{g_1^t, \ldots, g_n^t, x^t, x^{t+1}\}$. For any $s > 0$ we have*

$$\mathrm{E}\left[G_i^{t+1} \mid W^t\right] \leq (1 - \theta(s))G_i^t + \beta(s)\left\|\nabla f_i(x^{t+1}) - \nabla f_i(x^t)\right\|^2, \tag{16}$$

*where*

$$\theta(s) := 1 - (1 - \alpha)(1 + s), \quad and \quad \beta(s) := (1 - \alpha)\left(1 + s^{-1}\right). \tag{17}$$

**Lemma 8** (Richtárik et al. (2021)). *Let $0 < \alpha \leq 1$ and for $s > 0$ let $\theta(s)$ and $\beta(s)$ be as in equation 17. Then the solution of the optimization problem*

$$\min_s \left\{ \frac{\beta(s)}{\theta(s)} \; : \; 0 < s < \frac{\alpha}{1 - \alpha} \right\} \tag{18}$$

*is given by $s^* = \frac{1}{\sqrt{1-\alpha}} - 1$. Furthermore, $\theta(s^*) = 1 - \sqrt{1-\alpha}$, $\beta(s^*) = \frac{1-\alpha}{1-\sqrt{1-\alpha}}$ and*

$$\sqrt{\frac{\beta(s^*)}{\theta(s^*)}} = \frac{1}{\sqrt{1-\alpha}} - 1 = \frac{1}{\alpha} + \frac{\sqrt{1-\alpha}}{\alpha} - 1 \leq \frac{2}{\alpha} - 1. \tag{19}$$

We are now ready to conduct the proof.

**Theorem 6.** *Let Assumptions 1 and 2 hold, and let the stepsize be set as*

$$0 < \gamma \leq \left(L_- + L_+\sqrt{\frac{\beta}{\theta}}\right)^{-1}. \tag{20}$$

*Fix $T \geq 1$ and let $\hat{x}^T$ be chosen from the iterates $x^0, x^1, \ldots, x^{T-1}$ uniformly at random. Then*

$$\mathrm{E}\left[\left\|\nabla f(\hat{x}^T)\right\|^2\right] \leq \frac{2\left(f(x^0) - f^{inf}\right)}{\gamma T} + \frac{\mathrm{E}\left[G^0\right]}{\theta T}. \tag{21}$$

*Proof.* **STEP 1.** Recall that Lemma 7 says that

$$\mathrm{E}\left[\left\|g_i^{t+1} - \nabla f_i(x^{t+1})\right\|^2 \mid W^t\right] \leq (1 - \theta)\left\|g_i^t - \nabla f_i(x^t)\right\|^2 + \beta\left\|\nabla f_i(x^{t+1}) - \nabla f_i(x^t)\right\|^2, \tag{22}$$

where $\theta = \theta(s^*)$ and $\beta = \beta(s^*)$ are given by Lemma 8. Averaging inequalities equation 22 over $i \in \{1, 2, \ldots, n\}$ gives

$$
\begin{aligned}
\mathrm{E}\left[G^{t+1} \mid W^t\right] &= \frac{1}{n}\sum_{i=1}^n \mathrm{E}\left[\left\|g_i^{t+1} - \nabla f_i(x^{t+1})\right\|^2 \mid W^t\right] \\
&\leq (1 - \theta)\frac{1}{n}\sum_{i=1}^n \left\|g_i^t - \nabla f_i(x^t)\right\|^2 + \beta\frac{1}{n}\sum_{i=1}^n \left\|\nabla f_i(x^{t+1}) - \nabla f_i(x^t)\right\|^2 \\
&= (1 - \theta)G^t + \beta\frac{1}{n}\sum_{i=1}^n \left\|\nabla f_i(x^{t+1}) - \nabla f_i(x^t)\right\|^2 \\
&\leq (1 - \theta)G^t + \beta L_+ \left\|x^{t+1} - x^t\right\|^2. 
\end{aligned}
\tag{23}
$$

Using Tower property and $L$-smoothness in equation 23, we proceed to

$$\mathrm{E}\left[G^{t+1}\right] = \mathrm{E}\left[\mathrm{E}\left[G^{t+1} \mid W^t\right]\right] \leq (1 - \theta)\mathrm{E}\left[G^t\right] + \beta L_+^2 \mathrm{E}\left[\left\|x^{t+1} - x^t\right\|^2\right]. \tag{24}$$

**STEP 2.** Next, using Lemma 3 and Jensen's inequality applied to the function $x \mapsto \|x\|^2$, we obtain the bound

$$
\begin{aligned}
f(x^{t+1}) \quad &\leq \quad f(x^t) - \frac{\gamma}{2} \|\nabla f(x^t)\|^2 - \left(\frac{1}{2\gamma} - \frac{L_-}{2}\right) \|x^{t+1} - x^t\|^2 + \frac{\gamma}{2} \left\|\frac{1}{n}\sum_{i=1}^{n} \left(g_i^t - \nabla f_i(x^t)\right)\right\|^2 \\
&\leq \quad f(x^t) - \frac{\gamma}{2} \|\nabla f(x^t)\|^2 - \left(\frac{1}{2\gamma} - \frac{L_-}{2}\right) \|x^{t+1} - x^t\|^2 + \frac{\gamma}{2} G^t. \quad (25)
\end{aligned}
$$

Subtracting $f^{\mathrm{inf}}$ from both sides of equation 25 and taking expectation, we get

$$
\begin{aligned}
\mathrm{E}\left[f(x^{t+1}) - f^{\mathrm{inf}}\right] \quad &\leq \quad \mathrm{E}\left[f(x^t) - f^{\mathrm{inf}}\right] - \frac{\gamma}{2}\mathrm{E}\left[\|\nabla f(x^t)\|^2\right] \\
&\qquad - \left(\frac{1}{2\gamma} - \frac{L_-}{2}\right)\mathrm{E}\left[\|x^{t+1} - x^t\|^2\right] + \frac{\gamma}{2}\mathrm{E}\left[G^t\right]. \quad (26)
\end{aligned}
$$

**COMBINING STEP 1 AND STEP 2.** Let $\delta^t := \mathrm{E}\left[f(x^t) - f^{\mathrm{inf}}\right]$, $s^t := \mathrm{E}\left[G^t\right]$ and $r^t := \mathrm{E}\left[\|x^{t+1} - x^t\|^2\right]$. Then by adding equation 26 with a $\frac{\gamma}{2\theta}$ multiple of equation 24 we obtain

$$
\begin{aligned}
\delta^{t+1} + \frac{\gamma}{2\theta}s^{t+1} \quad &\leq \quad \delta^t - \frac{\gamma}{2}\|\nabla f(x^t)\|^2 - \left(\frac{1}{2\gamma} - \frac{L_-}{2}\right)r^t + \frac{\gamma}{2}s^t + \frac{\gamma}{2\theta}\left(\beta L_+^2 r^t + (1-\theta)s^t\right) \\
&= \quad \delta^t + \frac{\gamma}{2\theta}s^t - \frac{\gamma}{2}\|\nabla f(x^t)\|^2 - \left(\frac{1}{2\gamma} - \frac{L_-}{2} - \frac{\gamma}{2\theta}\beta L_+^2\right)r^t \\
&\leq \quad \delta^t + \frac{\gamma}{2\theta}s^t - \frac{\gamma}{2}\|\nabla f(x^t)\|^2.
\end{aligned}
$$

The last inequality follows from the bound $\gamma^2 \frac{\beta L_+^2}{\theta} + L_-\gamma \leq 1$, which holds from our assumption on the stepsize and Lemma 4. By summing up inequalities for $t = 0, \ldots, T-1$, we get

$$
0 \leq \delta^T + \frac{\gamma}{2\theta}s^T \leq \delta^0 + \frac{\gamma}{2\theta}s^0 - \frac{\gamma}{2}\sum_{t=0}^{T-1}\mathrm{E}\left[\|\nabla f(x^t)\|^2\right].
$$

Multiplying both sides by $\frac{2}{\gamma T}$, after rearranging we get

$$
\sum_{t=0}^{T-1}\frac{1}{T}\mathrm{E}\left[\|\nabla f(x^t)\|^2\right] \leq \frac{2\delta^0}{\gamma T} + \frac{s^0}{\theta T}.
$$

It remains to notice that the left hand side can be interpreted as $\mathrm{E}\left[\|\nabla f(\hat{x}^T)\|^2\right]$, where $\hat{x}^T$ is chosen from $x^0, x^1, \ldots, x^{T-1}$ uniformly at random. $\qquad\square$

### E.2 EF21 in PŁ regime

**Theorem 7.** *Let Assumptions 1, 2 and 5 hold, and let the stepsize in* EF21 *be set as*

$$
0 < \gamma \leq \min\left\{\left(L_- + L_+\sqrt{\frac{2\beta}{\theta}}\right)^{-1}, \frac{\theta}{2\mu}\right\}. \quad (27)
$$

*Let $\Psi^t := f(x^t) - f^{\mathrm{inf}} + \frac{\gamma}{\theta}G^t$. Then for any $T \geq 0$, we have*

$$
\mathrm{E}\left[\Psi^T\right] \leq (1 - \gamma\mu)^T\mathrm{E}\left[\Psi^0\right]. \quad (28)
$$

*Proof.* Again, this follows Richtárik et al. (2021) almost verbatim.

We proceed as in the previous proof, but use the PŁ inequality and subtract $f^{\mathrm{inf}}$ from both sides of equation 25 to get

$$
\begin{aligned}
\mathrm{E}\left[f(x^{t+1}) - f^{\mathrm{inf}}\right] \quad &\leq \quad \mathrm{E}\left[f(x^t) - f^{\mathrm{inf}}\right] - \frac{\gamma}{2}\|\nabla f(x^t)\|^2 - \left(\frac{1}{2\gamma} - \frac{L_-}{2}\right)\|x^{t+1} - x^t\|^2 + \frac{\gamma}{2}G^t \\
&\leq \quad (1 - \gamma\mu)\mathrm{E}\left[f(x^t) - f^{\mathrm{inf}}\right] - \left(\frac{1}{2\gamma} - \frac{L_-}{2}\right)\|x^{t+1} - x^t\|^2 + \frac{\gamma}{2}G^t.
\end{aligned}
$$

Let $\delta^t := \mathrm{E}\left[f(x^t) - f^{\mathrm{inf}}\right]$, $s^t := \mathrm{E}\left[G^t\right]$ and $r^t := \mathrm{E}\left[\left\|x^{t+1} - x^t\right\|^2\right]$. Then by adding the above inequality with a $\frac{\gamma}{\theta}$ multiple of equation 24, we obtain

$$
\begin{aligned}
\delta^{t+1} + \frac{\gamma}{\theta} s^{t+1} &\leq (1 - \gamma\mu)\delta^t - \left(\frac{1}{2\gamma} - \frac{L_-}{2}\right) r^t + \frac{\gamma}{2} s^t + \frac{\gamma}{\theta}\left((1-\theta)s^t + \beta L_+^2 r^t\right) \\
&= (1 - \gamma\mu)\delta^t + \frac{\gamma}{\theta}\left(1 - \frac{\theta}{2}\right) s^t - \left(\frac{1}{2\gamma} - \frac{L_-}{2} - \frac{\beta L_+^2 \gamma}{\theta}\right) r^t.
\end{aligned}
$$

Note that our assumption on the stepsize implies that $1 - \frac{\theta}{2} \leq 1 - \gamma\mu$ and $\frac{1}{2\gamma} - \frac{L_-}{2} - \frac{\beta L_+^2 \gamma}{\theta} \geq 0$. The last inequality follows from the bound $\gamma^2 \frac{2\beta L_+^2}{\theta} + \gamma L_- \leq 1$, which holds because of Lemma 4 and our assumption on the stepsize. Thus,

$$
\delta^{t+1} + \frac{\gamma}{\theta} s^{t+1} \leq (1 - \gamma\mu)\left(\delta^t + \frac{\gamma}{\theta} s^t\right).
$$

It remains to unroll the recurrence. $\qquad\square$

## F   COMMUNICATION MODEL

As mentioned in the introduction, *we consider the regime where the worker-to-server communication is the bottleneck of the system so that the server-to-workers communication can be neglected.* While this is a standard model used in many prior works, we include a brief explanation of why and when this regime is useful.

1. **Peer-to-peer communication.** First, this regime makes sense when the server is merely an abstraction, and does not exist physically. Indeed, from the point of view of each worker, the server may merely represent "all other nodes" combined. In this model, "a worker sending a message to the server" should be *interpreted* as this worker sending the message to all other workers. Clearly, in this model there is *no need* for the "server" to communicate the aggregated message back to the workers since aggregation is performed on all workers independently, and the aggregated message is immediately available to all workers without the need for any additional communication.

2. **Fast broadcast.** Second, the above regime makes sense in situations where the server exists physically, but is able to broadcast to the workers at a much higher speed compared to the worker-to-server communication. This happens in several distributed systems, e.g., on certain supercomputers (Mishchenko et al., 2019). Virtually all theoretical works on communication efficient distributed algorithms assume that the server-to-worker communication is cheap, and in this work we follow in their footsteps.

Having said that, our work can be extended to the more difficult regime where the server-to-worker communication is also costly (Horváth et al., 2019; Tang et al., 2019; Philippenko & Dieuleveut, 2020; Gorbunov et al., 2020). However, for simplicity, we do not explore this extension in this work.

## G   ON CONTRACTIVE COMPRESSORS AND ERROR FEEDBACK

### G.1   ON CONTRACTIVE COMPRESSORS

The most successful algorithmic solutions to solving the nonconvex distributed optimization problem (1) in a communication-efficient manner under the communication model described in Appendix F involve stochastic gradient descent (SGD) methods with *communication compression*. There are two large classes of such methods, depending on the type of compression operator involved: (i) methods that work with contractive (and possibly biased stochastic) compression operators, such as $\mathrm{Top}K$ or $\mathrm{Rank}K$, and (ii) methods that work with unbiased and independent (across the workers) stochastic compression operators, such as $\mathrm{Rand}K$.

A (randomized) compression operator $\mathcal{C} : \mathbb{R}^d \to \mathbb{R}^d$ is $\alpha$-contractive (we write $\mathcal{C} \in \mathbb{C}(\alpha)$), where $0 < \alpha \leq 1$, if

$$\mathrm{E}\left[\|\mathcal{C}(x) - x\|^2\right] \leq (1 - \alpha) \|x\|^2, \quad x \in \mathbb{R}^d. \tag{29}$$

A canonical example is the (deterministic) $\mathrm{Top}K$ compressor, which outputs the $K$ largest (in absolute value) entries of the input vector $x$, and zeroes out the rest. $\mathrm{Top}K$ is $\alpha$-contractive with $\alpha = K/d$. Another example is the $\mathrm{Rank}K$ compressor based on the best rank-$K$ approximation of $x$ represented as an $a \times b = d$ matrix. It can be shown that $\mathrm{Rank}K$ is $\alpha$-contractive with $\alpha = K/\min\{a,b\}$ (Safaryan et al., 2021, Section A.3.2). We refer to the work of Vogels et al. (2019) for a practical communication-efficient method PowerSGD based on low-rank approximations.

Of special importance are $\alpha$-compressors arising from *unbiased* compressors via appropriate scaling. Let $\mathcal{Q} : \mathbb{R}^d \to \mathbb{R}^d$ be an unbiased operator with variance proportional to the square norm of the input vector. That is, assume that $\mathrm{E}\left[\mathcal{Q}(x)\right] = x$ for all $x \in \mathbb{R}^d$ and that there exists $\omega \geq 0$ such that

$$\mathrm{E}\left[\|\mathcal{Q}(x) - x\|^2\right] \leq \omega \|x\|^2, \quad \forall x \in \mathbb{R}^d. \tag{30}$$

We will write $\mathcal{Q} \in \mathbb{U}(\omega)$ for brevity. It is well known that the operator $\mathcal{C} = (\omega + 1)^{-1}\mathcal{Q}$ is $\alpha$-contractive with $\alpha = (\omega + 1)^{-1}$. An example of a contractive compressor arising this way is

$(\omega + 1)^{-1}\text{Rand}K$, which keeps a subset of $K$ entries of the input vector $x$ chosen uniformly at random, and zeroes out the rest. As $\text{Top}K$, $(\omega + 1)^{-1}\text{Rand}K$ is $\alpha$-contractive, with $\alpha = K/d$.

Distributed SGD methods relying on general contractive compressors, i.e., on contractive which do *not* arise from unbiased compressors from scaling, need to rely on the error-feedback / error-compensation mechanism to avoid divergence.

### G.2 ON ERROR FEEDBACK

An alternative approach to the one represented by MARINA is to seek more aggressive compression, even at the cost of abandoning unbiasedness, in the hope that this will lead to better communication complexity in practice. This is the idea behind the class of *contractive compressors*, defined in (29), which have studied at least since the work of Seide et al. (2014). Example of such compressors are the $\text{Top}K$ (Alistarh et al., 2018) and $\text{Rank}K$ (Vogels et al., 2019; Safaryan et al., 2021) compressors.

While such compressors are indeed often very successful in practice, their theoretical impact on the methods using them is dramatically less understood than is the case with unbiased compressors. One of the key reasons for this that a naive use of biased compressors may lead to (exponential) divergence, even in simple problems (Beznosikov et al., 2020). Because of this, Seide et al. (2014) proposed the *error feedback* framework for controlling the error introduced by compression, and thus taming the method to convergence. While it has been successfully used by practitioners for many years, error feedback yielded the first convergence results only relatively recently (Stich et al., 2018; Stich & Karimireddy, 2019; Wu et al., 2018; Koloskova et al., 2019; Tang et al., 2019; Karimireddy et al., 2019; Qian et al., 2020; Beznosikov et al., 2020; Gorbunov et al., 2020).

The current best theoretical communication complexity results for error feedback belong to the EF21 method of Richtárik et al. (2021) who achieved their improvements by redesigning the original error feedback mechanism using the construction of a Markov compressor. However, even EF21 currently enjoys substantially weaker iteration and communication complexity than MARINA. For instance, we show in Appendix L that EF21 with $\text{Top}K$ is only proved to have the communication complexity of the gradient descent without any compression.

# H    COMPOSITION OF COMPRESSORS WITH AB ASSUMPTION AND UNBIASED COMPRESSORS

**Lemma 9.** *If $\{\mathcal{C}_i\}_{i=1}^n \in \mathbb{U}(A, B)$ and $\mathcal{Q}_i \in \mathbb{U}(\omega_i)$ for $i \in \{1, 2, \dots, n\}$, then $\{\mathcal{C}_i \circ \mathcal{Q}_i\}_{i=1}^n \in \mathbb{U}((\max_i \omega_i + 1)\, A, B)$.*

*Proof.* By the tower property, for all $a_1, \dots, a_n \in \mathbb{R}^d$, we have

$$
\mathrm{E}\left[\left\|\frac{1}{n}\sum_{i=1}^n \mathcal{C}_i(\mathcal{Q}_i(a_i)) - \frac{1}{n}\sum_{i=1}^n \mathcal{Q}_i(a_i)\right\|^2\right]
$$

$$
= \mathrm{E}\left[\mathrm{E}\left[\left.\left\|\frac{1}{n}\sum_{i=1}^n \mathcal{C}_i(\mathcal{Q}_i(a_i)) - \frac{1}{n}\sum_{i=1}^n \mathcal{Q}_i(a_i)\right\|^2 \right| \mathcal{Q}_1(a_1), \cdots, \mathcal{Q}_n(a_n)\right]\right]
$$

$$
\leq \mathrm{E}\left[A\frac{1}{n}\sum_{i=1}^n \|\mathcal{Q}_i(a_i)\|^2 - B\left\|\frac{1}{n}\sum_{i=1}^n \mathcal{Q}_i(a_i)\right\|^2\right].
$$

Since $\mathcal{Q}_i \in \mathbb{U}(\omega_i)$ for $i \in \{1, 2, \dots, n\}$, we get

$$
\mathrm{E}\left[\left\|\frac{1}{n}\sum_{i=1}^n \mathcal{C}_i(\mathcal{Q}_i(a_i)) - \frac{1}{n}\sum_{i=1}^n \mathcal{Q}_i(a_i)\right\|^2\right] \leq \left(\max_i \omega_i + 1\right) A \frac{1}{n}\sum_{i=1}^n \|a_i\|^2 - B\,\mathrm{E}\left[\left\|\frac{1}{n}\sum_{i=1}^n \mathcal{Q}_i(a_i)\right\|^2\right].
$$

Using Jensen's inequality, we derive inequalities:

$$
\mathrm{E}\left[\left\|\frac{1}{n}\sum_{i=1}^n \mathcal{Q}_i(a_i)\right\|^2\right] \geq \left\|\frac{1}{n}\sum_{i=1}^n \mathrm{E}\left[\mathcal{Q}_i(a_i)\right]\right\|^2 = \left\|\frac{1}{n}\sum_{i=1}^n a_i\right\|^2,
$$

and

$$
\mathrm{E}\left[\left\|\frac{1}{n}\sum_{i=1}^n \mathcal{C}_i(\mathcal{Q}_i(a_i)) - \frac{1}{n}\sum_{i=1}^n \mathcal{Q}_i(a_i)\right\|^2\right] \leq \left(\max_i \omega_i + 1\right) A \frac{1}{n}\sum_{i=1}^n \|a_i\|^2 - B\left\|\frac{1}{n}\sum_{i=1}^n a_i\right\|^2.
$$

The last inequality completes the proof. $\qquad\square$

# I  GENERAL EXAMPLES OF PERM$K$

For the sake of clarity, in the main part of our paper, we assumed that $n \mid d$ or $d \mid n$, and provided corresponding examples of Perm$K$ (see Definition 2 and Definition 3). Now, we provide two examples of Perm$K$ that work with any $n$ and $d$ and generalize the previous examples.

## I.1  CASE $d \geq n$

The following example generalizes for the case when $n$ does not divide $d$. Let us assume that $d = kn + r$ and $0 \leq r < n$. As in Definition 2, we permute coordinates and split them into the blocks of sizes $\{k, \ldots, k, r\}$. The first $n$ block of size $k$ we assign to nodes. Next, we take the last block of size $r$ and randomly assign each coordinate from this block to one node. As the size of the last block of size $r$ is less than $n$, some nodes will send one coordinate less.

**Definition 5** (Perm$K$ ($d \geq n$))**.** *Let us assume that $d \geq n$, $d = kn + r$, $0 \leq r < n$, $\pi^d = (\pi_1^d, \ldots, \pi_d^d)$ is a random permutation of $\{1, \cdots, d\}$, and $\pi^n = (\pi_1^n, \ldots, \pi_n^n)$ is a random permutation of $\{1, \cdots, n\}$. We define the tuple of vectors $S(x) = (x_{\pi_{kn+1}^d} e_{\pi_{kn+1}^d}, \ldots, x_{\pi_{kn+r}^d} e_{\pi_{kn+r}^d}, 0, \ldots, 0)$ of size $n$. Then,*

$$\mathcal{C}_i(x) := n \left( \sum_{j=k(i-1)+1}^{ki} x_{\pi_j^d} e_{\pi_j^d} + (S(x))_{\pi_i^n} \right).$$

**Theorem 8.** *Compressors $\{\mathcal{C}_i\}_{i=1}^n$ from Definition 5 belong to $\mathbb{IV}(1)$.*

*Proof.* We fix any $x \in \mathbb{R}^d$ and prove unbiasedness:

$$
\begin{aligned}
\mathrm{E}\left[\mathcal{C}_i(x)\right] &= n \left( \sum_{j=k(i-1)+1}^{ki} \mathrm{E}\left[x_{\pi_j^d} e_{\pi_j^d}\right] + \mathrm{E}\left[(S(x))_{\pi_i^n}\right] \right) \\
&= n \left( \frac{k}{d} x + \left(1 - \frac{r}{n}\right) 0 + \frac{r}{n} \frac{1}{d} x \right) \\
&= n \left( \frac{kn + r}{nd} \right) x \\
&= x,
\end{aligned}
$$

for all $i \in \{1, \ldots, n\}$.

Next, we derive the second moment:

$$
\begin{aligned}
\mathrm{E}\left[\left\|\mathcal{C}_i(x)\right\|^2\right] &= n^2 \left( \sum_{j=k(i-1)+1}^{ki} \mathrm{E}\left[\left\|x_{\pi_j^d} e_{\pi_j^d}\right\|^2\right] + \mathrm{E}\left[\left\|(S(x))_{\pi_i^n}\right\|^2\right] \right) \\
&= n^2 \left( \frac{k}{d} \left\|x\right\|^2 + \left(1 - \frac{r}{n}\right) \left\|0\right\|^2 + \frac{r}{n} \frac{1}{d} \left\|x\right\|^2 \right) \\
&= n^2 \left( \frac{kn + r}{nd} \right) \left\|x\right\|^2 \\
&= n \left\|x\right\|^2,
\end{aligned}
$$

We fix $x, y \in \mathbb{R}^d$. For all $i \neq l \in \{1, \ldots, n\}$, we have

$$
\begin{aligned}
&\mathrm{E}\left[\langle \mathcal{C}_i(x), \mathcal{C}_l(y) \rangle\right] \\
&= n^2 \left\langle \sum_{j=k(i-1)+1}^{ki} x_{\pi_j^d} e_{\pi_j^d} + (S(x))_{\pi_i^n}, \sum_{j=k(l-1)+1}^{kl} y_{\pi_j^d} e_{\pi_j^d} + (S(y))_{\pi_l^n} \right\rangle = 0,
\end{aligned}
$$

due to orthogonality of vectors $e_p$, for all $p \in \{1, \ldots, d\}$, and the fact that $i \neq l$.

Thus, for all $a_1, \ldots, a_n \in \mathbb{R}^d$, the following equality holds:

$$\mathrm{E}\left[\left\|\frac{1}{n}\sum_{i=1}^n \mathcal{C}_i(a_i)\right\|^2\right] = \frac{1}{n^2}\sum_{i=1}^n \mathrm{E}\left[\|\mathcal{C}_i(a_i)\|^2\right] + \frac{1}{n^2}\sum_{i\neq j}\mathrm{E}\left[\langle C_i(a_i), C_j(a_j)\rangle\right] = \frac{1}{n}\sum_{i=1}^n \|a_i\|^2.$$

Hence, Assumption 4 is fulfilled with $A = B = 1$.

$\square$

## I.2 CASE $n \geq d$

The following definition generalizes Definition 3 for the case when $d$ does not divide $n$. Let us assume that $n = qd + r$ and $0 \leq r < d$. As in Definition 3, we permute the multiset, where each coordinate occures $q$ times. Then, we randomly assign each element from the multiset of size $qd$ to one node. Note that $r$ randomly chosen nodes are idle.

**Definition 6** (Perm$K$, $(n \geq d)$). *Let us assume that $n \geq d$, $n = qd + r$, $0 \leq r < d$. Let us fix point $x \in \mathbb{R}^d$, that we want to compress. Define the tuple of vectors $\widehat{S}(x) = (x_1e_1, \ldots, x_1e_1, x_2e_2, \ldots, x_2e_2, \ldots, x_de_d, \ldots, x_de_d)$, where each vector occurs $q$ times. Concat $r$ zero vectors to $\widehat{S}(x)$: $S(x) = \widehat{S}(x) \oplus (0, \ldots, 0)$. Let $\pi = (\pi_1, \ldots, \pi_n)$ be a random permutation of $\{1, \ldots, n\}$. Define*

$$\mathcal{C}_i(x) := \frac{n}{q}\left(S(x)\right)_{\pi_i}.$$

**Theorem 9.** *Compressors $\{\mathcal{C}_i\}_{i=1}^n$ from Definition 6 belong to $\mathbb{IV}(A)$ with $A = 1 - \frac{n(q-1)}{(n-1)q}$.*

*Proof.* We start with proving the unbiasedness:

$$\mathrm{E}\left[\mathcal{C}_i(x)\right] = \frac{n}{q}\sum_{j=1}^d x_je_j\mathbf{Prob}\left(\pi_i = j\right) = \frac{n}{q}\sum_{j=1}^d x_je_j\frac{q}{n} = x,$$

for all $i \in \{1, \ldots, n\}, x \in \mathbb{R}^d$.

Next, we find the second moment:

$$\mathrm{E}\left[\|\mathcal{C}_i(x)\|^2\right] = \frac{n^2}{q^2}\sum_{i=1}^d x_i^2\mathbf{Prob}\left(\pi_i = j\right) = \frac{n}{q}\|x\|^2,$$

for all $i \in \{1, \ldots, n\}, x \in \mathbb{R}^d$.

For all $i \neq j \in \{1, \ldots, n\}$ and $x, y \in \mathbb{R}^d$, we have

$$\mathrm{E}\left[\langle\mathcal{C}_i(x), \mathcal{C}_j(y)\rangle\right] = \frac{n^2}{q^2}\sum_{k=1}^d x_ky_k\mathbf{Prob}\left(\pi_i = k, \pi_j = k\right)$$

$$= \frac{n^2}{q^2}\sum_{k=1}^d x_ky_k\frac{q(q-1)}{n(n-1)} = \frac{n(q-1)}{(n-1)q}\langle x, y\rangle.$$

Thus, for all $a_1, \ldots, a_n \in \mathbb{R}^d$, the following equality holds:

$$\mathrm{E}\left[\left\|\frac{1}{n}\sum_{i=1}^n \mathcal{C}_i(a_i)\right\|^2\right] = \frac{1}{n^2}\sum_{i=1}^n \mathrm{E}\left[\|\mathcal{C}_i(a_i)\|^2\right] + \frac{1}{n^2}\sum_{i\neq j}\mathrm{E}\left[\langle C_i(a_i), C_j(a_j)\rangle\right]$$

$$= \frac{1}{nq}\sum_{i=1}^n \|a_i\|^2 + \frac{n(q-1)}{(n-1)q}\left(\frac{1}{n^2}\sum_{i\neq j}\langle a_i, a_j\rangle\right)$$

$$= \left(\frac{1}{q} - \frac{(q-1)}{(n-1)q}\right)\frac{1}{n}\sum_{i=1}^n \|a_i\|^2 + \frac{n(q-1)}{(n-1)q}\left\|\frac{1}{n}\sum_{i=1}^n a_i\right\|^2.$$

Hence, Assumption 4 is fulfilled with $A = B = 1 - \frac{n(q-1)}{(n-1)q}$.

$\square$

## J   IMPLEMENTATION DETAILS OF PERM$K$

Now, we discuss the implementation details of Perm$K$ from Definition 2. Unlike Rand$K$ and Top$K$ compressors, Perm$K$ compressors are statistically dependent. We provide a simple idea of how to manage dependence between nodes. First of all, note that the samples of random permutation $\pi$ are the only source of randomness. By Definition 2, they are shared between nodes and generated in each communication round. However, instead of sharing the samples, we can generate these samples in each node regardless of other nodes. Almost all random generation libraries and frameworks are deterministic (or pseudorandom) and only depend on the initial random seed. Thus, at the beginning of the optimization procedure, all nodes should set the same initial random seed and then call the same function that generates samples of a random permutation. The computation complexity of generating a sample from a random permutation $\pi = (\pi_1, \ldots, \pi_d)$ is $\mathcal{O}(d)$ using the Fisher-Yates shuffle algorithm (Fisher & Yates, 1938; Knuth, 1997). All other examples of compressors can be implemented in the same fashion.

## K   MORE EXAMPLES OF PERMUTATION-BASED COMPRESSORS

### K.1   BLOCK PERMUTATION COMPRESSOR

In block permutation compressor, we partition the set $\{1, \ldots, d\}$ into $m \leq n$ disjoint blocks. For each block $P_i$, $\left\lfloor \frac{n}{m} \right\rfloor$ devices sparsify their vectors to coordinates with indices in $P_i$ only.

**Definition 7.** *Let $P$ to be a partition of the set $\{1, \ldots, d\}$ into $m \leq n$ non-empty subsets, and $n = mq + r$, where $0 \leq r < m$. Define matrices $\boldsymbol{A}_1, \ldots, \boldsymbol{A}_n$ as follows: put $\boldsymbol{A}_i := 0$ if $i > mq$. Denote the subsets in $P$ as $P_1, \cdots, P_m$. Next, for any $P_i \in P$, we set $\boldsymbol{A}_{(i-1)q+1}, \boldsymbol{A}_{(i-1)q+2}, \cdots, \boldsymbol{A}_{iq}$ to $\frac{n}{q} \operatorname{Diag}(P_i)$. Here by $\operatorname{Diag}(S)$ we mean the diagonal matrix where each $i^{th}$ diagonal entry is equal to 1 if $i \in S$ and 0 otherwise. Let $\pi = (\pi_1, \ldots, \pi_n)$ be a random permutation of set $\{1, \ldots, n\}$. We define $\mathcal{C}_i(x) := \boldsymbol{A}_{\pi_i} x$. We call the set $\{\mathcal{C}_i\}_{i=1}^n$ the block permutation compressor.*

**Lemma 10.** *Compressors $\{\mathcal{C}_i\}_{i=1}^n$ belong to $\mathbb{IV}(A)$ with $A = 1 - \frac{n(q-1)}{(n-1)q}$.*

*Proof.* We start with the proof of unbiasedness:

$$\mathrm{E}\left[\mathcal{C}_i(x)\right] = \frac{1}{n} \sum_{i=1}^n \boldsymbol{A}_{\pi_i} x = \left(\frac{1}{n} \cdot \frac{n}{q} \sum_{i=1}^m q \operatorname{Diag}(P_i)\right) x = \boldsymbol{I} x = x,$$

for all $i \in \{1, \ldots, n\}, x \in \mathbb{R}^d$.

Next, we establish the second moment:

$$\mathrm{E}\left[\|\mathcal{C}_i(x)\|^2\right] = \frac{r}{n} \cdot 0 + \sum_{l=1}^m \frac{q}{n} \sum_{j \in P_l} \left|\frac{n}{q} x_j\right|^2 = \frac{q}{n} \sum_{j=1}^d \left|\frac{n}{q} x_j\right|^2 = \frac{n}{q} \|x\|^2,$$

for all $i \in \{1, \ldots, n\}, x \in \mathbb{R}^d$.

The following equality will be useful for the AB assumption:

$$
\begin{aligned}
\mathrm{E}\left[\langle \mathcal{C}_i(x), \mathcal{C}_j(y)\rangle\right] &= \mathrm{E}\left[\langle \boldsymbol{A}_{\pi_i} x, \boldsymbol{A}_{\pi_j} y\rangle\right] \\
&= \sum_{k=1}^m \left(\sum_{l \in P_k} \frac{n^2}{q^2} x_l y_l\right) \mathbf{Prob}\left(\pi_i \in P_k, \pi_j \in P_k\right) \\
&= \sum_{k=1}^m \left(\sum_{l \in P_k} \frac{n^2}{q^2} x_l y_l\right) \frac{q(q-1)}{n(n-1)} \\
&= \frac{n(q-1)}{q(n-1)} \langle x, y\rangle,
\end{aligned}
$$

for all $i \neq j \in \{1, \ldots, n\}, x, y \in \mathbb{R}^d$. Thus,

$$
\begin{aligned}
\mathrm{E}\left[\left\|\frac{1}{n}\sum_{i=1}^n \mathcal{C}_i(a_i)\right\|^2\right] &= \frac{1}{qn}\sum_{i=1}^n \|a_i\|^2 + \frac{n(q-1)}{(n-1)qn^2}\sum_{i\neq j}\langle a_i, a_j\rangle \\
&= \left(\frac{1}{qn} - \frac{n(q-1)}{(n-1)qn^2}\right)\sum_{i=1}^n \|a_i\|^2 + \frac{n(q-1)}{(n-1)q}\left\|\frac{1}{n}\sum_{i=1}^n a_i\right\|^2 \\
&= \left(\frac{(n-1)-q+1}{(n-1)q}\right)\frac{1}{n}\sum_{i=1}^n \|a_i\|^2 + \frac{n(q-1)}{(n-1)q}\left\|\frac{1}{n}\sum_{i=1}^n a_i\right\|^2 \\
&= \frac{n-q}{(n-1)q}\cdot\frac{1}{n}\sum_{i=1}^n \|a_i\|^2 + \frac{n(q-1)}{(n-1)q}\left\|\frac{1}{n}\sum_{i=1}^n a_i\right\|^2 \\
&= \left(1 - \frac{n(q-1)}{(n-1)q}\right)\frac{1}{n}\sum_{i=1}^n \|a_i\|^2 + \frac{n(q-1)}{(n-1)q}\left\|\frac{1}{n}\sum_{i=1}^n a_i\right\|^2,
\end{aligned}
$$

for all $a_1, \ldots, a_n \in \mathbb{R}^d$. Hence, Assumption 4 is fulfilled with $A = B = 1 - \frac{n(q-1)}{(n-1)q}$. $\qquad\square$

## K.2 PERMUTATION OF MAPPINGS

**Definition 8.** *Let $Q_1, \ldots, Q_n : \mathbb{R}^d \to \mathbb{R}^d$ be a collection of* deterministic *mappings $\mathbb{R}^d \to \mathbb{R}^d$. Let $\pi = (\pi_1, \ldots, \pi_n)$ be a random permutation of set $\{1, \ldots, n\}$, where $n > 1$. Define $\mathcal{C}_i := Q_{\pi_i}$. Assume that the following conditions hold:*

1. *There exists $\omega \geq 0$ such that $\mathrm{E}\left[\|\mathcal{C}_i(x)\|^2\right] \leq (\omega + 1)\|x\|^2$ for all $i \in \{1, \ldots, n\}$, $x, y \in \mathbb{R}^d$.*

2. *There exists $\theta \in \mathbb{R}$ such that $\sum_{i=1}^n \langle Q_i(x), Q_i(y)\rangle = \theta\langle x, y\rangle$ for all $x, y \in \mathbb{R}^d$.*

3. *$\frac{1}{n}\sum_{i=1}^n Q_i(x) = x$ for all $x \in \mathbb{R}^d$.*

*We call the collection $\{\mathcal{C}_i\}_{i=1}^n$ the* permutation of mappings.

**Lemma 11.** *Permutation of mappings belongs to $\mathbb{U}(A, B)$ with $A = \frac{\omega+1}{n} - \frac{1}{n-1}\left(1 - \frac{\theta}{n^2}\right)$ and $B = 1 - \frac{n}{n-1}\left(1 - \frac{\theta}{n^2}\right)$.*

*Proof.* Unbiasedness follows directly from Condition 3. Let us fix $a_1, \ldots, a_n \in \mathbb{R}^d$. We shall now establish the AB assumption.

$$
\begin{aligned}
& \mathrm{E}\left[\left\|\frac{1}{n}\sum_{i=1}^{n}\mathcal{C}_i(a_i) - \frac{1}{n}\sum_{i=1}^{n}a_i\right\|^2\right] \\
&= \mathrm{E}\left[\left\|\frac{1}{n}\sum_{i=1}^{n}(Q_{\pi_i}(a_i) - a_i)\right\|^2\right] \\
&= \frac{1}{n^2}\mathrm{E}\left[\sum_{i=1}^{n}\|Q_{\pi_i}(a_i) - a_i\|^2 + \sum_{i\neq j}\langle Q_{\pi_i}(a_i) - a_i, Q_{\pi_j}(a_j) - a_j\rangle\right] \\
&\leq \frac{1}{n^2}\mathrm{E}\left[\omega\sum_{i=1}^{n}\|a_i\|^2 + \sum_{i\neq j}\langle Q_{\pi_i}(a_i) - a_i, Q_{\pi_j}(a_j) - a_j\rangle\right] \\
&= \frac{1}{n^2}\left(\omega\sum_{i=1}^{n}\|a_i\|^2 + \mathrm{E}\left[\sum_{i\neq j}\langle Q_{\pi_i}(a_i), Q_{\pi_j}(a_j)\rangle\right] - \sum_{i\neq j}\langle a_i, a_j\rangle\right) \\
&= \frac{1}{n^2}\left(\omega\sum_{i=1}^{n}\|a_i\|^2 - \left\|\sum_{i=1}^{n}a_i\right\|^2 + \sum_{i=1}^{n}\|a_i\|^2 + \sum_{i\neq j}\mathrm{E}\left[\langle Q_{\pi_i}(a_i), Q_{\pi_j}(a_j)\rangle\right]\right) \\
&= \frac{(\omega+1)}{n}\frac{1}{n}\sum_{i=1}^{n}\|a_i\|^2 - \left\|\frac{1}{n}\sum_{i=1}^{n}a_i\right\|^2 + \frac{1}{n^2}\sum_{i\neq j}\mathrm{E}\left[\langle Q_{\pi_i}(a_i), Q_{\pi_j}(a_j)\rangle\right]
\end{aligned}
$$

Let us now compute $\frac{1}{n^2}\sum_{i\neq j}\mathrm{E}\left[\langle Q_{\pi_i}(a_i), Q_{\pi_j}(a_j)\rangle\right]$.

$$
\begin{aligned}
\frac{1}{n^2}\sum_{i\neq j}\mathrm{E}\left[\langle Q_{\pi_i}(a_i), Q_{\pi_j}(a_j)\rangle\right] &= \frac{1}{n^2}\sum_{i\neq j}\frac{1}{n(n-1)}\sum_{u\neq v}\langle Q_u(a_i), Q_v(a_j)\rangle \\
&= \frac{1}{n(n-1)}\sum_{i\neq j}\frac{1}{n^2}\sum_{u\neq v}\langle Q_u(a_i), Q_v(a_j)\rangle.
\end{aligned}
$$

Now,

$$
\begin{aligned}
\frac{1}{n^2}\sum_{u\neq v}\langle Q_u(x), Q_v(y)\rangle &= \frac{1}{n^2}\sum_{u=1}^{n}\sum_{v=1}^{n}\langle Q_u(x), Q_v(y)\rangle - \frac{1}{n^2}\sum_{u=1}^{n}\langle Q_u(x), Q_v(y)\rangle \\
&\overset{\text{Condition 3}}{=} \langle x, y\rangle - \frac{1}{n^2}\sum_{u=1}^{n}\langle Q_u(x), Q_u(y)\rangle \\
&\overset{\text{Condition 2}}{=} \left(1 - \frac{\theta}{n^2}\right)\langle x, y\rangle, \quad \forall x, y \in \mathbb{R}^d.
\end{aligned}
$$

Hence,

$$
\frac{1}{n^2}\sum_{i\neq j}\mathrm{E}\left[\langle Q_{\pi_i}(a_i), Q_{\pi_j}(a_j)\rangle\right] = \frac{1}{n(n-1)}\left(1 - \frac{\theta}{n^2}\right)\sum_{i\neq j}\langle a_i, a_j\rangle.
$$

Finally,

$$
\mathrm{E}\left[\left\|\frac{1}{n}\sum_{i=1}^{n}\mathcal{C}_i(a_i) - \frac{1}{n}\sum_{i=1}^{n}a_i\right\|^2\right]
$$

$$
\leq \frac{(\omega+1)}{n}\frac{1}{n}\sum_{i=1}^{n}\|a_i\|^2 - \left\|\frac{1}{n}\sum_{i=1}^{n}a_i\right\|^2 + \frac{1}{n(n-1)}\left(1 - \frac{\theta}{n^2}\right)\sum_{i\neq j}\langle a_i, a_j\rangle
$$

$$
= \left(\frac{\omega+1}{n} - \frac{1}{n-1}\left(1 - \frac{\theta}{n^2}\right)\right)\frac{1}{n}\sum_{i=1}^{n}\|a_i\|^2 - \left(1 - \frac{n}{n-1}\left(1 - \frac{\theta}{n^2}\right)\right)\left\|\frac{1}{n}\sum_{i=1}^{n}a^i\right\|^2.
$$

Hence, Assumption 4 is fulfilled with $A = \frac{\omega+1}{n} - \frac{1}{n-1}\left(1 - \frac{\theta}{n^2}\right)$, $B = 1 - \frac{n}{n-1}\left(1 - \frac{\theta}{n^2}\right)$. $\qquad\square$

## L   Analysis of Complexity Bounds

In this section, we analyze the complexities bounds of optimization methods, and typically these bounds have a structure of a function that we analyze in the following lemma.

**Lemma 12.** *Let us consider function*

$$f(x, y) = (x + (1 - x)y) \left( a + b\sqrt{\left( \frac{1}{x} - 1 \right) \left( \frac{1}{y} - 1 \right)} \right),$$

*where $x \in (0, 1]$, $y \in [y_{\min}, 1]$, $y_{\min} \in (0, 1/2]$, $a \geq 0$, and $b \geq 0$, then*

$$f(x, y) \geq \frac{1}{2} \min\{a, ay_{\min} + b\}, \quad \forall x, y \in (0, 1].$$

*Proof.* First, let us assume that $x \geq 1/2$. Then,

$$f(x, y) \geq \frac{a}{2}.$$

Second, let us assume that $y \geq 1/2$. Then,

$$f(x, y) \geq a \left( x + \frac{1}{2}(1 - x) \right) \geq \frac{a}{2}.$$

Finally, let us assume, that $y \leq 1/2$ and $x \leq 1/2$. Then,

$$
\begin{aligned}
f(x, y) &= (x + (1 - x)y) \left( a + b\sqrt{\left( \frac{1}{x} - 1 \right) \left( \frac{1}{y} - 1 \right)} \right) \\
&\geq a(1 - x)y + b(x + (1 - x)y)\sqrt{\left( \frac{1}{x} - 1 \right) \left( \frac{1}{y} - 1 \right)} \\
&\geq a(1 - x)y + b(x(1 - y) + (1 - x)y)\sqrt{\left( \frac{1}{x} - 1 \right) \left( \frac{1}{y} - 1 \right)} \\
&\geq a(1 - x)y + bxy \left( \left( \frac{1}{y} - 1 \right) + \left( \frac{1}{x} - 1 \right) \right) \sqrt{\left( \frac{1}{x} - 1 \right) \left( \frac{1}{y} - 1 \right)} \\
&\geq a(1 - x)y + 2bxy \left( \frac{1}{x} - 1 \right) \left( \frac{1}{y} - 1 \right) \\
&\geq a(1 - x)y + 2b(1 - x)(1 - y) \\
&\geq \frac{ay_{\min}}{2} + \frac{b}{2}.
\end{aligned}
$$

$\square$

### L.1   Nonconvex optimization

#### L.1.1   Case $n \leq d$

We analyze case, when $n \leq d$. For simplicity, we assume that $n \mid d$, $n > 1$, and $d > 1$. For $\text{Perm}K$ from Definition 2, constants $A = B = 1$ in AB inequality (see Lemma 1). We define the upper bound for the communication complexity of MARINA with $\text{Perm}K$ as $N_{\text{Perm}K}(p)$, where $p$ is a parameter of MARINA, and MARINA with $\text{Rand}K$ as $N_{\text{Rand}K}(p, k)$, where $k$ is a parameter of $\text{Rand}K$. From Theorem 4, we have that oracle complexity of MARINA with $\text{Perm}K$ is equal to

$$\mathcal{O} \left( \frac{\Delta^0}{\varepsilon} \left( L_- + \sqrt{\frac{1 - p}{p}} L_{\pm} \right) \right).$$

During each iteration of MARINA, on average, each node sends the number of bits equal to

$$\mathcal{O}\left(pd + (1-p)\frac{d}{n}\right),$$

thus, the communication complexity predicted by theory is

$$N_{\text{Perm}K}(p) := \frac{\Delta^0}{\varepsilon}\left(pd + (1-p)\frac{d}{n}\right)\left(L_- + \sqrt{\frac{1-p}{p}}L_\pm\right) \tag{31}$$

up to a constant factor.

Analogously, for $\text{Rand}K$, the communication complexity predicted by theory is

$$N_{\text{Rand}K}(p,k) := \frac{\Delta^0}{\varepsilon}(pd + (1-p)k)\left(L_- + \sqrt{\frac{1-p}{p}\frac{\frac{d}{k}-1}{n}}L_+\right) \tag{32}$$

up to a constant factor. To the best of our knowledge, this is the state-of-the-art theoretical communication complexity bound for the $\text{Rand}K$ compressor in the non-convex regime.

Finally, for $\text{Top}K$, by Theorem 6, the theoretical communication complexity is

$$N_{\text{Top}K}(k) := \frac{\Delta^0}{\varepsilon}k\left(L_- + L_+\frac{d-k+\sqrt{d^2-dk}}{k}\right) \tag{33}$$

up to a constant factor. We consider the variant of EF21, where $g_i^0$ are initialized with gradients $\nabla f_i(x^0)$, for all $i \in \{1,\ldots,n\}$, thus $\mathrm{E}\left[G^0\right] = 0$ in Theorem 6.

The following lemma will help us to choose the optimal parameters of $N_{\text{Perm}K}(p)$, $N_{\text{Rand}K}(p,k)$, and $N_{\text{Top}K}(k)$. We wish to stress that in the following sections the term *lower bound* should *not* be understood as the lower bound for the communication complexity. We merely provide lower bounds for the upper bounds predicted by the current state-of-the-art theory by optimizing their parameters.

**Lemma 13.** *For communication complexity $N_{\text{Perm}K}(p)$ of* MARINA *with PermK, communication complexity $N_{\text{Rand}K}(p,k)$ of* MARINA *with RandK and communication complexity $N_{\text{Top}K}(k)$ of* EF21 *with TopK defined in (31), (32) and (33) the following inequalities hold:*

1. *Lower bounds:*

$$N_{\text{Perm}K}(p) \geq \frac{\Delta^0}{2\varepsilon}\min\left\{dL_-, \frac{d}{n}L_- + \frac{dL_\pm}{\sqrt{n}}\right\}, \quad \forall p \in (0,1].$$

   *Upper bounds:*

$$N_{\text{Perm}K}\left(\frac{1}{n}\right) \leq \frac{2\Delta^0}{\varepsilon}\left(\frac{d}{n}L_- + \frac{dL_\pm}{\sqrt{n}}\right), \tag{34}$$

$$N_{\text{Perm}K}(1) = \frac{\Delta^0 dL_-}{\varepsilon}. \tag{35}$$

2. *Lower bounds:*

$$N_{\text{Rand}K}(p,k) \geq \frac{\Delta^0}{2\varepsilon}\min\left\{dL_-, \frac{dL_+}{\sqrt{n}}\right\}, \quad \forall p \in (0,1], \forall k \in \{1,\ldots,d\},$$

   *Upper bounds: For all $k \in \{1,\ldots,d/\sqrt{n}\}$, $p = k/d$,*

$$N_{\text{Rand}K}(p,k) \leq \frac{4\Delta^0 dL_+}{\varepsilon\sqrt{n}}. \tag{36}$$

   *Moreover, for all $k \in \{1,\ldots,d\}$, $p = 1$,*

$$N_{\text{Rand}K}(1,k) = \frac{\Delta^0 dL_-}{\varepsilon}. \tag{37}$$

3.

$$\min_{k \in \{1,\ldots,d\}} N_{\text{Top}K}(k) = N_{\text{Top}K}(d) = \frac{\Delta^0 dL_-}{\varepsilon} \tag{38}$$

*Proof.*

1. We start with the first inequality:

$$
\begin{aligned}
N_{\text{Perm}K}(p) &= \frac{\Delta^0}{\varepsilon} \left( pd + (1-p)\frac{d}{n} \right) \left( L_- + \sqrt{\frac{1-p}{p}} L_\pm \right) \\
&= \frac{\Delta^0}{\varepsilon} \left( p + (1-p)\frac{1}{n} \right) \left( dL_- + dL_\pm \sqrt{\frac{1-p}{p}} \right) \\
&= \frac{\Delta^0}{\varepsilon} \left( p + (1-p)\frac{1}{n} \right) \left( dL_- + dL_\pm \sqrt{\frac{n}{n}} \sqrt{\frac{1-p}{p}} \right) \\
&\geq \frac{\Delta^0}{\varepsilon} \left( p + (1-p)\frac{1}{n} \right) \left( dL_- + \frac{dL_\pm}{\sqrt{n}} \sqrt{\left( \frac{1}{p} - 1 \right)(n-1)} \right).
\end{aligned}
$$

Using Lemma 12 with $a = dL_-$, $b = \frac{dL_\pm}{\sqrt{n}}$, and $y_{\min} = 1/n$, we get

$$
\begin{aligned}
N_{\text{Perm}K}(p) &= \frac{\Delta^0}{\varepsilon} \left( pd + (1-p)\frac{d}{n} \right) \left( L_- + \sqrt{\frac{1-p}{p}} L_\pm \right) \\
&\geq \frac{\Delta^0}{2\varepsilon} \min \left\{ dL_-, \frac{d}{n}L_- + \frac{dL_\pm}{\sqrt{n}} \right\}.
\end{aligned}
$$

for all $p \in (0,1]$. We can obtain the bound 34 if we take $p = 1/n$:

$$
\begin{aligned}
N_{\text{Perm}K}\left( \frac{1}{n} \right) &= \frac{\Delta^0}{\varepsilon} \left( \frac{d}{n} + \left( 1 - \frac{1}{n} \right)\frac{d}{n} \right) \left( L_- + \sqrt{n-1}L_\pm \right) \\
&\leq \frac{2\Delta^0}{\varepsilon} \left( \frac{d}{n}L_- + \frac{d}{\sqrt{n}}L_\pm \right).
\end{aligned}
$$

We obtain the equality 35 by taking $p = 1$.

2.

$$
\begin{aligned}
N_{\text{Rand}K}(p,k) &= \frac{\Delta^0}{\varepsilon} (pd + (1-p)k) \left( L_- + \sqrt{\frac{1-p}{p} \frac{\frac{d}{k} - 1}{n}} L_+ \right) \\
&= \frac{\Delta^0}{\varepsilon} \left( p + (1-p)\frac{k}{d} \right) \left( dL_- + \frac{dL_+}{\sqrt{n}} \sqrt{\left( \frac{1}{p} - 1 \right)\left( \frac{d}{k} - 1 \right)} \right).
\end{aligned}
$$

Using Lemma 12 with $a = dL_-$, $b = \frac{dL_\pm}{\sqrt{n}}$, and $y_{\min} = 1/d$, we get

$$
N_{\text{Rand}K}(p,k) \geq \frac{\Delta^0}{2\varepsilon} \min \left\{ dL_-, L_- + \frac{dL_+}{\sqrt{n}} \right\} \geq \frac{\Delta^0}{2\varepsilon} \min \left\{ dL_-, \frac{dL_+}{\sqrt{n}} \right\},
$$

for all $p \in (0,1], k \in \{1,\ldots,d\}$. We can obtain the bound 36 if we take $k \in \{1,\ldots,d/\sqrt{n}\}$ and $p = k/d$:

$$
\begin{aligned}
N_{\text{Rand}K}(p,k) &\leq \frac{2\Delta^0}{\varepsilon} \left( kL_- + k\left( \frac{d}{k} - 1 \right)\frac{L_+}{\sqrt{n}} \right) \\
&\leq \frac{2\Delta^0}{\varepsilon} \left( \frac{dL_-}{\sqrt{n}} + \frac{dL_+}{\sqrt{n}} \right) \leq \frac{4\Delta^0 dL_+}{\varepsilon\sqrt{n}}.
\end{aligned}
$$

The equality 37 is obtained by taking $p = 1$.

3. This part is easily proved, using $L_- \leq L_+$ from Lemma 2, and directly minimizing (38).

$\square$

In Table 4, we summarize bounds (34), (35), (36), (37), and (38).

### L.1.2 CASE $n \geq d$

Now, we analyze case, when $n \geq d$. For simplicity, without losing the generality, we assume that $d \mid n, n > 1$, and $d > 1$. Then, PermK from Definition 3 satisfies the AB inequality with $A = B = \frac{d-1}{n-1}$.

In each iteration of MARINA, on average, PermK sends

$$\mathcal{O}\left(pd + (1-p)\right)$$

bits, thus the theoretical communication complexity is

$$N_{\text{Perm}K}(p) := \frac{\Delta^0}{\varepsilon}\left(pd + (1-p)\right)\left(L_- + \sqrt{\frac{1-p}{p}\frac{d-1}{n-1}}L_\pm\right) \tag{39}$$

up to a constant factor.

**Lemma 14.** *For communication complexity $N_{\text{Perm}K}(p)$ of MARINA with PermK, communication complexity $N_{\text{Rand}K}(p, k)$ of MARINA with RandK and communication complexity $N_{\text{Top}K}(k)$ of EF21 with TopK defined in (39), (32) and (33) the following inequalities hold:*

1. *Lower bounds:*

$$N_{\text{Perm}K}(p) \geq \frac{\Delta^0}{2\varepsilon}\min\left\{dL_-, L_- + \frac{dL_\pm}{\sqrt{n}}\right\}, \quad \forall p \in (0, 1].$$

*Upper bounds:*

$$N_{\text{Perm}K}\left(\frac{1}{d}\right) \leq \frac{4\Delta^0}{\varepsilon}\left(L_- + \frac{dL_\pm}{\sqrt{n}}\right), \tag{40}$$

$$N_{\text{Perm}K}(1) = \frac{\Delta^0 dL_-}{\varepsilon}. \tag{41}$$

2. *Lower bounds:*

$$N_{\text{Rand}K}(p, k) \geq \frac{\Delta^0}{2\varepsilon}\min\left\{dL_-, L_- + \frac{dL_+}{\sqrt{n}}\right\}, \quad \forall p \in (0, 1], \forall k \in \{1, \ldots, d\},$$

*Upper bounds:*

$$N_{\text{Rand}K}\left(\frac{1}{d}, 1\right) \leq \frac{2\Delta^0}{\varepsilon}\left(L_- + \frac{dL_+}{\sqrt{n}}\right), \tag{42}$$

*Moreover, for all $k \in \{1, \ldots, d\}, p = 1$,*

$$N_{\text{Rand}K}(1, k) = \frac{\Delta^0 dL_-}{\varepsilon}. \tag{43}$$

3.

$$\min_{k \in \{1, \ldots, d\}} N_{\text{Top}K}(k) = N_{\text{Top}K}(d) = \frac{\Delta^0 dL_-}{\varepsilon} \tag{44}$$

*Proof.*

1.

$$N_{\mathrm{Perm}K}(p) = \tfrac{\Delta^0}{\varepsilon}\,(pd + (1-p))\left(L_- + \sqrt{\tfrac{1-p}{p}\tfrac{d-1}{n-1}}L_\pm\right)$$

$$\geq \tfrac{\Delta^0}{\varepsilon}\left(p + (1-p)\tfrac{1}{d}\right)\left(dL_- + \tfrac{dL_\pm}{\sqrt{n}}\sqrt{\left(\tfrac{1}{p}-1\right)(d-1)}\right)$$

Using Lemma 12 with $a = dL_-$, $b = \tfrac{dL_\pm}{\sqrt{n}}$, and $y_{\min} = 1/d$, we get

$$N_{\mathrm{Perm}K}(p) \geq \frac{\Delta^0}{2\varepsilon}\,\min\left\{dL_-, L_- + \frac{dL_\pm}{\sqrt{n}}\right\}$$

for all $p \in (0,1]$. We can show the bound 40 if we take $p = 1/d$:

$$N_{\mathrm{Perm}K}\left(\frac{1}{d}\right) = \frac{\Delta^0}{\varepsilon}\left(1 + \left(1 - \frac{1}{d}\right)1\right)\left(L_- + \frac{d-1}{\sqrt{n-1}}L_\pm\right) \leq \frac{4\Delta^0}{\varepsilon}\left(L_- + \frac{dL_\pm}{\sqrt{n}}\right)$$

The bound 41 is obtained by taking $p = 1$.

2. As in Lemma 13 we can get, that

$$N_{\mathrm{Rand}K}(p,k) \geq \frac{\Delta^0}{2\varepsilon}\,\min\left\{dL_-, L_- + \frac{dL_+}{\sqrt{n}}\right\},$$

for all $p \in (0,1], k \in \{1, \ldots, d\}$. Moreover, if we take $p = 1/d$ and $k = 1$, we have

$$N_{\mathrm{Rand}K}\left(\frac{1}{d}, 1\right) \leq \frac{2\Delta^0}{\varepsilon}\left(L_- + \frac{dL_+}{\sqrt{n}}\right).$$

The bound 43 is obtained by taking $p = 1$.

3. For $\mathrm{Top}K$, the reasoning the same as in Lemma 13.

$\square$

In Table 4, we summarize bounds (40), (41), (42), (43), and (44).

## L.2 PŁ ASSUMPTION

### L.2.1 CASE $n \leq d$

Using the same reasoning as in Appendix L.1, Theorem 5 and Theorem 7, we can show that communication complexities predicted by theory are equal to

$$N_{\mathrm{Perm}K}(p) := \log\frac{\Delta^0}{\varepsilon}\left(pd + (1-p)\frac{d}{n}\right)\max\left\{\left(\frac{L_-}{\mu} + \sqrt{\frac{2(1-p)}{p}}\frac{L_\pm}{\mu}\right), \frac{1}{p}\right\}, \tag{45}$$

$$N_{\mathrm{Rand}K}(p,k) := \log\frac{\Delta^0}{\varepsilon}\,(pd + (1-p)k)\max\left\{\left(\frac{L_-}{\mu} + \sqrt{\frac{2(1-p)}{p}\frac{\frac{d}{k}-1}{n}}\frac{L_+}{\mu}\right), \frac{1}{p}\right\}, \tag{46}$$

$$N_{\mathrm{Top}K}(k) := \log\frac{\Delta^0}{\varepsilon}k\max\left\{\left(\frac{L_-}{\mu} + \frac{L_+}{\mu}\frac{d-k+\sqrt{d^2-dk}}{k}\right), \frac{1}{1-\sqrt{1-\frac{k}{d}}}\right\}. \tag{47}$$

up to a constant factor.

**Lemma 15.** *For communication complexity $N_{\mathrm{Perm}K}(p)$ of* MARINA *with PermK, communication complexity $N_{\mathrm{Rand}K}(p,k)$ of* MARINA *with RandK and communication complexity $N_{\mathrm{Top}K}(k)$ of* EF21 *with TopK defined in (45), (46) and (47) the following inequalities hold[9]:*

---

[9]In the lemma, we use "Big Theta" notation, which means, that if $f(x) = \Theta(g(x))$, then $f$ is bounded both above and below by $g$ asymptotically up to a logarithmic factor.

1.

$$\inf_{p \in (0,1]} N_{\text{Perm}K}(p) = \Theta\left(\max\left\{\frac{1}{\mu}\min\left\{dL_-, \frac{d}{n}L_- + \frac{d}{\sqrt{n}}L_\pm\right\}, d\right\}\right),$$

2.

$$\inf_{p \in (0,1], k \in \{1,\dots,d\}} N_{\text{Rand}K}(p,k) = \Theta\left(\max\left\{\frac{1}{\mu}\min\left\{dL_-, \frac{d}{\sqrt{n}}L_+\right\}, d\right\}\right),$$

3.

$$\min_{k \in \{1,\dots,d\}} N_{\text{Top}K}(k) = \Theta\left(\frac{dL_-}{\mu}\right).$$

*Proof.* Rearranging (45), (46) and (47), we get

$$N_{\text{Perm}K}(p) = \log\frac{\Delta^0}{\varepsilon}\max\left\{\left(pd + (1-p)\frac{d}{n}\right)\left(\frac{L_-}{\mu} + \sqrt{\frac{2(1-p)}{p}}\frac{L_\pm}{\mu}\right), d + \frac{(1-p)d}{pn}\right\},$$

$$N_{\text{Rand}K}(p,k) = \log\frac{\Delta^0}{\varepsilon}\max\left\{(pd + (1-p)k)\left(\frac{L_-}{\mu} + \sqrt{\frac{2(1-p)}{p}\frac{\frac{d}{k}-1}{n}}\frac{L_+}{\mu}\right), d + \frac{(1-p)k}{p}\right\},$$

$$N_{\text{Top}K}(k) = \log\frac{\Delta^0}{\varepsilon}\max\left\{k\left(\frac{L_-}{\mu} + \frac{L_+}{\mu}\frac{d-k+\sqrt{d^2-dk}}{k}\right), \frac{k}{1-\sqrt{1-\frac{k}{d}}}\right\}.$$

Note, that

$$\frac{k}{1-\sqrt{1-\frac{k}{d}}} \geq d, \quad \forall k \in \{1,\dots,d\},$$

thus in all complexities, the second terms inside the $\max$ brackets are at least $d$.

Analysis of first terms inside the $\max$ brackets is the same as in Lemma 13. $\square$

In Table 6, we provide complexity bounds with optimal parameters of algorithms.

### L.2.2 CASE $n \geq d$

The only difference here is that the communication complexity of Perm$K$ predicted by our theory is the following:

$$N_{\text{Perm}K}(p) := \log\frac{\Delta^0}{\varepsilon}(pd + (1-p))\max\left\{\left(\frac{L_-}{\mu} + \sqrt{\frac{2(1-p)}{p}\frac{d-1}{n-1}}\frac{L_\pm}{\mu}\right), \frac{1}{p}\right\}. \quad (48)$$

**Lemma 16.** *For communication complexity $N_{\text{Perm}K}(p)$ of* MARINA *with PermK, communication complexity $N_{\text{Rand}K}(p,k)$ of* MARINA *with RandK and communication complexity $N_{\text{Top}K}(k)$ of* EF21 *with TopK defined in (48), (46) and (47) the following inequalities hold:*

1.

$$\inf_{p \in (0,1]} N_{\text{Perm}K}(p) = \Theta\left(\max\left\{\frac{1}{\mu}\min\left\{dL_-, L_- + \frac{d}{\sqrt{n}}L_\pm\right\}, d\right\}\right),$$

2.

$$\inf_{p \in (0,1], k \in \{1,\dots,d\}} N_{\text{Rand}K}(p,k) = \Theta\left(\max\left\{\frac{1}{\mu}\min\left\{dL_-, L_- + \frac{d}{\sqrt{n}}L_+\right\}, d\right\}\right),$$

3.

$$\min_{k \in \{1,\ldots,d\}} N_{\text{Top}K}(k) = \Theta\left(\frac{dL_-}{\mu}\right).$$

The proof of Lemma 16 is the same as in Lemma 15.

Using the same reasoning as before, we provide complexity bounds in Table 6.

## M  GROUP HESSIAN VARIANCE

We showed the communication complexity improvement of MARINA algorithm with $\text{Perm}K$ under the assumption that $L_\pm \ll L_-$. In general, $L_\pm$ can be large; however, we can still use the notion of $L_\pm$ but in a different way, by splitting the functions into several groups where $L_\pm$ is small.

We split a set $\{1, \cdots, n\}$ into nonempty sets $\{\mathcal{G}_k\}_{k=1}^g$, $\bigcup_{k=1}^g \mathcal{G}_k = \{1, \cdots, n\}$, $\mathcal{G}_i \bigcap \mathcal{G}_j = \emptyset$, for all $i \neq j \in \{1, \cdots, g\}$, and $|\mathcal{G}_k| > 0$, for all $k \in \{1, \cdots, g\}$. Let us fix some set $\mathcal{G}_k$ and define functions

$$\mathcal{L}_-^{\mathcal{G}_k}(x, y) := \left\| \frac{1}{|\mathcal{G}_k|} \sum_{i \in |\mathcal{G}_k|} (\nabla f_i(x) - \nabla f_i(y)) \right\|^2,$$

$$\mathcal{L}_+^{\mathcal{G}_k}(x, y) := \frac{1}{|\mathcal{G}_k|} \sum_{i \in |\mathcal{G}_k|} \|\nabla f_i(x) - \nabla f_i(y)\|^2,$$

$$\mathcal{L}_\pm^{\mathcal{G}_k}(x, y) := \mathcal{L}_+^{\mathcal{G}_k}(x, y) - \mathcal{L}_-^{\mathcal{G}_k}(x, y)$$

and the smallest constants $L_-^{\mathcal{G}_k}, L_+^{\mathcal{G}_k}, L_\pm^{\mathcal{G}_k}$ for functions $\mathcal{L}_-^{\mathcal{G}_k}(x, y), \mathcal{L}_+^{\mathcal{G}_k}(x, y)$, and $\mathcal{L}_\pm^{\mathcal{G}_k}(x, y)$, such that

$$\mathcal{L}_-^{\mathcal{G}_k}(x, y) \leq \left(L_-^{\mathcal{G}_k}\right)^2 \|x - y\|^2, \mathcal{L}_+^{\mathcal{G}_k}(x, y) \leq \left(L_+^{\mathcal{G}_k}\right)^2 \|x - y\|^2, \mathcal{L}_\pm^{\mathcal{G}_k}(x, y) \leq \left(L_\pm^{\mathcal{G}_k}\right)^2 \|x - y\|^2,$$

for all $k \in \{1, \cdots, g\}, x, y \in \mathbb{R}^d$.

In this section, we have the following assumption about groups.

**Assumption 6.** *Compressors between groups are independent, i.e. $\mathcal{C}_i$ and $\mathcal{C}_j$ are independent, for all $i \in \mathcal{G}_k, j \in \mathcal{G}_p, k \neq p$. And Assumption 4 is satisfied with constants $A_{\mathcal{G}_k}$ and $B_{\mathcal{G}_k}$ inside each group $\mathcal{G}_k$, for $k \in \{1, \cdots, g\}$.*

Now, we prove group AB inequality.

**Lemma 17** (Group AB inequality)**.** *Let us assume that Assumptions 3 and 6 hold, then*

$$\mathrm{E}\left[\left\| \frac{1}{n}\sum_{i=1}^n \mathcal{C}_i(a_i) - \frac{1}{n}\sum_{i=1}^n a_i \right\|^2\right]$$
$$\leq \sum_{k=1}^g \frac{A_{\mathcal{G}_k}|\mathcal{G}_k|^2}{n^2} \frac{1}{|\mathcal{G}_k|} \sum_{i \in \mathcal{G}_k} \|a_i\|^2 - \sum_{k=1}^g \frac{B_{\mathcal{G}_k}|\mathcal{G}_k|^2}{n^2} \left\| \frac{1}{|\mathcal{G}_k|} \sum_{i \in \mathcal{G}_k} a_i \right\|^2. \quad (49)$$

*Proof.*

$$\mathrm{E}\left[\left\| \frac{1}{n}\sum_{i=1}^n \mathcal{C}_i(a_i) - \frac{1}{n}\sum_{i=1}^n a_i \right\|^2\right]$$
$$= \frac{1}{n^2} \sum_{k=1}^g \mathrm{E}\left[\left\| \sum_{i \in \mathcal{G}_k} \mathcal{C}_i(a_i) - \sum_{i \in \mathcal{G}_k} a_i \right\|^2\right]$$
$$+ \frac{1}{n^2} \sum_{k \neq p} \mathrm{E}\left[\left\langle \sum_{i \in \mathcal{G}_k} \mathcal{C}_i(a_i) - \sum_{i \in \mathcal{G}_k} a_i, \sum_{i \in \mathcal{G}_p} \mathcal{C}_i(a_i) - \sum_{i \in \mathcal{G}_p} a_i \right\rangle\right].$$

Due to independence and unbiasedness, the last term vanishes, and, using AB inequality, we get

$$
\mathrm{E}\left[\left\|\frac{1}{n}\sum_{i=1}^{n}\mathcal{C}_i(a_i)-\frac{1}{n}\sum_{i=1}^{n}a_i\right\|^2\right]
$$

$$
=\frac{1}{n^2}\sum_{k=1}^{g}\mathrm{E}\left[\left\|\sum_{i\in\mathcal{G}_k}\mathcal{C}_i(a_i)-\sum_{i\in\mathcal{G}_k}a_i\right\|^2\right]
$$

$$
=\sum_{k=1}^{g}\frac{|\mathcal{G}_k|^2}{n^2}\mathrm{E}\left[\left\|\frac{1}{|\mathcal{G}_k|}\sum_{i\in\mathcal{G}_k}\mathcal{C}_i(a_i)-\frac{1}{|\mathcal{G}_k|}\sum_{i\in\mathcal{G}_k}a_i\right\|^2\right]
$$

$$
\le\sum_{k=1}^{g}\frac{|\mathcal{G}_k|^2}{n^2}\left(A_{\mathcal{G}_k}\left(\frac{1}{|\mathcal{G}_k|}\sum_{i\in\mathcal{G}_k}\|a_i\|^2\right)-B_{\mathcal{G}_k}\left\|\frac{1}{|\mathcal{G}_k|}\sum_{i\in\mathcal{G}_k}a_i\right\|^2\right).
$$

From this we can get the result. □

Next, we prove analogous lemma to Lemma 5.

**Lemma 18.** *Let us consider $g^{t+1}$ from Line 8 of Algorithm 1 and assume, that Assumptions 3 and 6 hold. Moreover, if Assumption 2 holds for every group $\mathcal{G}_k$, for $k\in\{1,\cdots,g\}$, then*

$$
\mathrm{E}\left[\left\|g^{t+1}-\nabla f(x^{t+1})\right\|^2\Big|x^{t+1}\right]
$$

$$
\le(1-p)\left(\sum_{k=1}^{g}\frac{(A_{\mathcal{G}_k}-B_{\mathcal{G}_k})|\mathcal{G}_k|^2}{n^2}\left(L_+^{\mathcal{G}_k}\right)^2+\sum_{k=1}^{g}\frac{B_{\mathcal{G}_k}|\mathcal{G}_k|^2}{n^2}\left(L_\pm^{\mathcal{G}_k}\right)^2\right)\left\|x^{t+1}-x^t\right\|^2
$$

$$
+(1-p)\left\|g^t-\nabla f(x^t)\right\|^2. \tag{50}
$$

*Proof.* In the view of definition of $g^{t+1}$, we get

$$
\mathrm{E}\left[\left\|g^{t+1}-\nabla f(x^{t+1})\right\|^2\Big|x^{t+1}\right]
$$

$$
=(1-p)\mathrm{E}\left[\left\|g^t+\frac{1}{n}\sum_{i=1}^{n}\mathcal{C}_i\left(\nabla f_i(x^{t+1})-\nabla f_i(x^t)\right)-\nabla f(x^{t+1})\right\|^2\Big|x^{t+1}\right]
$$

$$
=(1-p)\mathrm{E}\left[\left\|\frac{1}{n}\sum_{i=1}^{n}\mathcal{C}_i\left(\nabla f_i(x^{t+1})-\nabla f_i(x^t)\right)-\nabla f(x^{t+1})+\nabla f(x^t)\right\|^2\Big|x^{t+1}\right]
$$

$$
+(1-p)\left\|g^t-\nabla f(x^t)\right\|^2.
$$

In the last inequality we used unbiasedness of $\mathcal{C}_i$. Using (49), we get

$$
\mathrm{E}\left[\left\|g^{t+1} - \nabla f(x^{t+1})\right\|^2 \Big| x^{t+1}\right]
$$

$$
\leq (1-p)\mathrm{E}\left[\left\|\frac{1}{n}\sum_{i=1}^n \mathcal{C}_i\left(\nabla f_i(x^{t+1}) - \nabla f_i(x^t)\right) - \nabla f(x^{t+1}) + \nabla f(x^t)\right\|^2 \Big| x^{t+1}\right]
$$

$$
+ (1-p)\left\|g^t - \nabla f(x^t)\right\|^2.
$$

$$
\leq (1-p)\left(\sum_{k=1}^g \frac{A_{\mathcal{G}_k}|\mathcal{G}_k|^2}{n^2}\frac{1}{|\mathcal{G}_k|}\sum_{i\in\mathcal{G}_k}\left\|\nabla f_i(x^{t+1}) - \nabla f_i(x^t)\right\|^2\right.
$$

$$
\left. - \sum_{k=1}^g \frac{B_{\mathcal{G}_k}|\mathcal{G}_k|^2}{n^2}\left\|\frac{1}{|\mathcal{G}_k|}\sum_{i\in\mathcal{G}_k}\nabla f_i(x^{t+1}) - \nabla f_i(x^t)\right\|^2\right)
$$

$$
+ (1-p)\left\|g^t - \nabla f(x^t)\right\|^2
$$

$$
= (1-p)\left(\sum_{k=1}^g \frac{(A_{\mathcal{G}_k} - B_{\mathcal{G}_k})|\mathcal{G}_k|^2}{n^2}\mathcal{L}_+^{\mathcal{G}_k}(x^{t+1}, x^t) + \sum_{k=1}^g \frac{B_{\mathcal{G}_k}|\mathcal{G}_k|^2}{n^2}\mathcal{L}_\pm^{\mathcal{G}_k}(x^{t+1}, x^t)\right)
$$

$$
+ (1-p)\left\|g^t - \nabla f(x^t)\right\|^2
$$

$$
\leq (1-p)\left(\sum_{k=1}^g \frac{(A_{\mathcal{G}_k} - B_{\mathcal{G}_k})|\mathcal{G}_k|^2}{n^2}\left(L_+^{\mathcal{G}_k}\right)^2 + \sum_{k=1}^g \frac{B_{\mathcal{G}_k}|\mathcal{G}_k|^2}{n^2}\left(L_\pm^{\mathcal{G}_k}\right)^2\right)\left\|x^{t+1} - x^t\right\|^2
$$

$$
+ (1-p)\left\|g^t - \nabla f(x^t)\right\|^2.
$$

$\square$

Let us define

$$
\widehat{L}_{\mathcal{G}}^2 := \left(\sum_{k=1}^g \frac{(A_{\mathcal{G}_k} - B_{\mathcal{G}_k})|\mathcal{G}_k|^2}{n^2}\left(L_+^{\mathcal{G}_k}\right)^2 + \sum_{k=1}^g \frac{B_{\mathcal{G}_k}|\mathcal{G}_k|^2}{n^2}\left(L_\pm^{\mathcal{G}_k}\right)^2\right).
$$

**Theorem 10.** *Let Assumptions 1, 2, 3 and 6 be satisfied. Let the stepsize in* MARINA *be chosen as*

$$
\gamma \leq \left(L_- + \sqrt{\frac{1-p}{p}\widehat{L}_{\mathcal{G}}^2}\right)^{-1},
$$

*then after $T$ iterations,* MARINA *finds point $\hat{x}^T$ for which* $\mathrm{E}\left[\left\|\nabla f(\hat{x}^T)\right\|^2\right] \leq \frac{2\Delta^0}{\gamma T}$.

**Theorem 11.** *Let Assumptions 1, 2, 3, 5 and 6 be satisfied and*

$$
\gamma \leq \min\left\{\left(L_- + \sqrt{\frac{2(1-p)}{p}\widehat{L}_{\mathcal{G}}^2}\right)^{-1}, \frac{p}{2\mu}\right\},
$$

*then for $x^T$ from* MARINA *algorithm the following inequality holds:*

$$
\mathrm{E}\left[f(x^T) - f^\star\right] \leq (1-\gamma\mu)^T \Delta^0.
$$

We omit proofs of this theorems as they repeat proofs from Appendix C.6 and D.2; the only difference is that we have to take $\widehat{L}^2 = \widehat{L}_{\mathcal{G}}^2$.

Let us assume that $n \leq d$, all groups have equal sizes $|\mathcal{G}_k| = G$ and constants $L_\pm^{\mathcal{G}_k} = L_\pm^G$, for all $k \in \{1, \ldots, g\}$, and in each group we use PermK compressor from Definition 2, thus communication complexity predicted by our theory is the following:

$$
N_{\mathrm{Perm}K}^G(p) := \frac{\Delta^0}{\varepsilon}\left(pd + (1-p)\frac{d}{G}\right)\left(L_- + \sqrt{\frac{(1-p)G}{pn}}L_\pm^G\right).
$$

Using the same reasoning as in Lemma 13, we can take $p = 1$ or $p = 1/G$ to get that

$$\inf_{p \in (0,1]} N^G_{\text{Perm}K}(p) = \mathcal{O}\left(\frac{2\Delta^0}{\varepsilon} \min\left\{dL_-, \frac{d}{G}L_- + \frac{d}{\sqrt{n}}L^G_\pm\right\}\right). \tag{51}$$

For the case when we have one group, we restore the communication complexity from Lemma 13.

Comparing (34) with (51), we see that $dL_-/n$ from (34) is always better than $dL_-/G$ from (51); however; if $dL_\pm/\sqrt{n}$ is a bottleneck and $L^G_\pm$ is small, then communication complexity (51) can be better.

Let us consider an example of a quadratic optimization task with two groups, wherein one group, all matrices are equal to $A$, and in another one, all matrices are equal to $B$, $A \neq B$, $A = A^\top \succcurlyeq 0$ and $B = B^\top \succcurlyeq 0$, then $G = n/2$, $L^G_\pm = 0$, and $L_\pm > 0$ (see Example 3). Hence, we get that

$$\inf_{p \in (0,1]} N^G_{\text{Perm}K}(p) = \mathcal{O}\left(\frac{\Delta^0 d}{\varepsilon n}L_-\right).$$

This bound is better than (34) by at least the factor $1 + \sqrt{n}L_\pm/L_-$.

