# OpenReview forum: "Permutation Compressors for Provably Faster Distributed Nonconvex Optimization"
_ICLR.cc/2022/Conference — ICLR 2022 Poster_

### Official Review · Reviewer_JiFh · 2021-10-27

**Correctness:** 4
**Technical Novelty And Significance:** 3
**Empirical Novelty And Significance:** 3
**Recommendation:** 6
**Confidence:** 2

**Main Review:**

Overall, the paper is interesting, and the analysis in the  main paper is careful and clear. The assumptions are not too outlandish, and it makes sense that sparsification should be easier when data / Hessians are more uniform. (I suspect that is what is happening in the MNIST example, for example.) The paper seems strong, but I am not familiar enough with this specific area to comment on its novelty or impact.

Weaknesses: Since the paper seems mainly to put forward the theoretical advantages of different schemes, I question if the proposed methods are better because of better proof techniques, or if they are really always like that in practice. For example, I suspect a fully correlated scheme where every worker uses the *same* sparsification (but randomly regenerated at each iteration) should achieve good performance as well. It is unclear to me, at least intuitively, why a permutation approach would be better than random sampling. (For this, a few toy examples would be illuminating.) I also think (and this is less easy to address, so it's not really factoring to my decision of the review) that the few experiments given may not be exhaustive enough to provide general conclusions as to the superiority of each method.

Also, since the appendix is about 40 pages long, I did not have time to read it. If the authors feel there is one specific proof that they are particularly proud of, I'd be happy to take a look and give a less lukewarm review.

**Summary Of The Paper:**

This paper investigates a permutation based sparsification technique for distributed optimization, and prove that, under a mildly tighter condition than L-smoothness, it can achieve better communication complexity than the existing random sparsification technique. The method is largely under the larger construct of MARINA, a distributed optimization scheme by Gorbunov. A (1+d/sqrt(n)) factor improvement is shown with the permutation vs randomization technique when the Hessian variance is small, and several promising numerical results are given.

**Summary Of The Review:**

See above.

---

> ### Author Response · Authors · 2021-11-22
> **Response to comments 1 (overall evaluation)**
>
> > Overall, the paper is interesting, and the analysis in the main paper is careful and clear. The assumptions are not too outlandish, and it makes sense that sparsification should be easier when data / Hessians are more uniform. (I suspect that is what is happening in the MNIST example, for example.) The paper seems strong, but I am not familiar enough with this specific area to comment on its novelty or impact.
>
> *Response:*
>
> - Thanks for the positive evaluation of the paper! This is much appreciated.
>
> - However, we wish to point out that we achieve a new theoretical SOTA for a problem of key significance in modern deep learning: **communication efficient distributed training in the nonconvex regime.** Plus, our improvements can be truly huge (in the best case scenario when Hessian variance is negligible): up to $O(\sqrt{n})$ and $O(1+d/\sqrt{n})$ in the $d\geq n$ and $d\leq n$ case, respectively. If $d\geq n$ and $n=100$, which is common in standard distributed training, this is up to $10\times$ improvement. In federated learning, one can work with $n=10^6$ million workers, and $d=10^6$ as well, we get $1000\times$ improvement in theory (i.e., in the number of communicated bits needed to solve the problem).
>
> - Moreover, our theory has predictive power, as illustrated through our experiments. We believe this is very significant since we believe theoretical SOTA is as important as empirical SOTA.
>
> We would like to kindly ask you to take this into consideration. Thank you!
>
> Authors

---

> ### Author Response · Authors · 2021-11-22
> **Response to comments 2 (just a better proof technique or actual improvement in practice?)**
>
> > Weaknesses: Since the paper seems mainly to put forward the theoretical advantages of different schemes, I question if the proposed methods are better because of better proof techniques, or if they are really always like that in practice. For example, I suspect a fully correlated scheme where every worker uses the same sparsification (but randomly regenerated at each iteration) should achieve good performance as well. It is unclear to me, at least intuitively, why a permutation approach would be better than random sampling. (For this, a few toy examples would be illuminating.)
>
> *Response:*
>
> - We believe that the fact that we obtain theoretical SOTA is significant on its own, even if we did not perform any experiments, and even if the method was not practical. (In the same way, we believe that empirical SOTA is significant on its own, even not supported by any piece of theory whatsoever).
>
> - Having said that, our new method is not better just due to an improved proof technique. We use the same method -- MARINA -- but empower it with a carefully designed correlated compressor system (e.g., PermK, or more generally, input variance compressors). The correlation is designed to provably decrease the variance of the gradient estimator, which has a positive effect on iteration complexity, and this has a positive effect on communication complexity. Our experiments clearly show this as well - the plots would not look like they do in Fig 1 if the theoretical improvement came from a better proof technique only!
>
> - The basic intuition behind (the simplest of the PermK compressors) is this. Consider the special case $n=d$. Assume that all gradients are the same, i.e., we work in the fully homogeneous regime: $\nabla f_i = \nabla f_j \forall i,j$. Consider the following three sparsification strategies (we will add this toy illuminating example to the camera-ready version of the paper):
>
> 1) All 3 nodes use **the same** Rand-1 sparsifier, but randomize it at the start of each iteration. In this case, there is no benefit coming from averaging: the average gradient estimator will be the same independent of $n$. So, such a strategy cannot possibly (in the worst case) benefit from $n$. The estimator will have the same variance as if just a single node was used to perform sparsification. This is, by the way, the key issue also with biased compressors such as Top-$1$ and with error feedback mechanisms: they do not benefit from $n$ in the fully homogeneous regime, and hence benefit can't be proved (without further assumptions) in the worst case.
>
> 2)  All 3 nodes use the Rand-1 sparsifer, but each one samples the coordinate to keep **independently**. In this case, the variance of the average of the sparsified vectors clearly improved with $n$, and this means that the average of the compressed vectors is closer to the true gradient, which means that the method behaves more like GD and less like SGD. This reduces the number of communication rounds while keeping the per-round per-node communication the same: O(1).
>
> 3) All 3 nodes use the Rand-1 sparsifier, but this time the nodes agree that **each keeps a different coordinate!** This can be done by choosing a random permutation of $(1,2,3)$, say, $\pi=(2, 3, 1)$, and telling node $i$ to keep coordinate $\pi(i)$. This is precise how PermK looks like when $n=d=3$. When this compression technique is used, we recover the true gradient exactly, with zero variance!
>
> - So, the compressors described in steps 1, 2, and 3 above get better, with 1 being the worst, 2 being fine, and 3 being the best. Of course, we do not merely consider the fully homogeneous case. Our analysis considers the arbitrary level of heterogeneity, and we get improvement in all heterogeneity regimes, except in the worst case: when Hessian variance $L_{\pm}^2$ reaches its maximal value $L_+^2$, in which case we recover the result obtained by Gorbunov et al. (In fact, we do a more careful analysis even in this case, as we explain in footnote 2 - but we believe it is fair to attribute that improvement to their work as it follows from simple observation and hence we do not wish to take credit for it).

---

> > ### Comment · Reviewer_JiFh · 2021-11-28
> > **Keep my score**
> >
> > Thanks to the authors for their response. Overall I have no reason to believe the paper is not mathematically sound, and it is an interesting contribution. However, I am not sure if the results are as groundbreaking as the authors phrase, which I think would require more detailed empirical studies and maybe a clearer explanation of the regime / assumptions / proof techniques. (I agree that there exists good theoretical results, I just think something explained a bit more clearly raises the level from borderline accept to clear accept.) Overall, I am keeping my borderline accept score.

---

> > > ### Author Response · Authors · 2021-11-28
> > > **On decision to keep the score unchanged...**
> > >
> > > Dear Reviewer JiFh,
> > >
> > > 1) You say *"Thanks to the authors for their response. Overall I have no reason to believe the paper is not mathematically sound, and it is an interesting contribution.*" **Thanks, we are glad you like the contribution!**
> > >
> > > 2) We believe **we have addressed all your concerns. However, your response does not comment on this at all. You do not say whether we failed to address any of your concerns. If we indeed managed to answer everything satisfactorily (and we are convinced of that), then we would suggest a higher score is justified. On the other hand, if you believe we did not address any of your original concerns, then please let us know what remains to be explained so that we can do so. This is the fair and right thing to do. There is only one more day for us to talk to you.**
> > >
> > > 3) Instead of commenting in our rebuttal, you now say: *"However, I am not sure if the results are as groundbreaking as the authors phrase, which I think would require more detailed empirical studies and maybe a clearer explanation of the regime / assumptions / proof techniques."* **Unfortunately, such a vague and abstract criticism** (we are sorry we need to label it as such, but I hope you agree no concrete criticism is given here) **is impossible to defend against, and as such, is not fair.**   Let us comment on this:
> > >
> > > - First, **we never used the word "groundbreaking" to describe our work.** But we do claim, justifiably, that **we obtain new theoretical SOTA for distributed nonconvex optimization in terms of communication complexity. The best-case improvement factors are huge - and are clearly mentioned in the abstract already.** Do you believe this is *not* the case? If you believe so, we would expect you to point to a method with better theoretical communication complexity. If, however, you do believe our claim, and as experts in the field we know it is correct, then what do you mean by saying *"I am not sure if the results are as groundbreaking as the authors phrase"*? Does this mean you do not value clear theoretical advances in the field leading to new theoretical SOTA (we do not claim you mean to say this, but we really do not know what you are saying here)? We believe that this is a vague and unjustified statement that should not be used in a scientific review.
> > >
> > > - **What we study is an important and highly studied problem: communication efficient distributed optimization / training in the smooth nonconvex regime. We ask: how can a new theoretical SOTA method for a problem central to modern machine learning be judged as "weak accept" only?**
> > >
> > > - We believe our explanations are clear, all proofs are included in detail, and all assumptions are included in detail. Whatever was not clear to other reviewers and yourself, we addressed in a very detailed rebuttal. We do not need more assumptions than the previous SOTA of Gorbunov (2021), and we clearly state so in the paper (we only require each $f_i$ to have $L_i$-Lipschitz gradient, and we require $f$ to be lower bounded!). Our empirical studies are sufficient to show that our theoretical predictions translate into practice. Having said that, **our work should be judged through its theoretical contributions as this is where the heart of our paper is. We were asked by you and others to perform some more empirical studies, and we did so. We included a table with extra insights.**
> > >
> > > 4) You say "*(I agree that there exists good theoretical results, I just think something explained a bit more clearly raises the level from borderline accept to clear accept.) Overall, I am keeping my borderline accept score."* **We believe the level of exposition in our paper is of a very high standard. What exactly do you want us to explain a bit more clearly? Even assuming that some small clarity issues exist, why should a new theoretical SOTA result be judged as weak accept only?**
> > >
> > >
> > > ---
> > >
> > > We deeply apologize for the somewhat assertive tone of our response. But we believe the issues we raised above with your latest comment and final decision are serious. We are glad that you view our paper positively, but we do not think it is justified to keep your score of weak accept given the fact that we addressed in detail all the issues you raised, as well as the issues of other reviewers. We believe that theoretical SOTA should command a much higher than a borderline accept score.
> > >
> > > We very much hope you will re-evaluate your final decision. Thanks for considering our plea! And please accept our apologies if this message seems inappropriate to you. We are merely expressing, honestly, our professional opinion here. That is all. We value your opinion and effort even if we disagree with it!
> > >
> > > Authors

---

> ### Author Response · Authors · 2021-11-22
> **Experiments not exhaustive and 40 pages of appendix**
>
> > I also think (and this is less easy to address, so it's not really factoring to my decision of the review) that the few experiments given may not be exhaustive enough to provide general conclusions as to the superiority of each method. Also, since the appendix is about 40 pages long, I did not have time to read it. If the authors feel there is one specific proof that they are particularly proud of, I'd be happy to take a look and give a less lukewarm review.
>
> *Response:*
>
> - We agree that no finite set of experiments can possibly be exhaustive. However, we provide robust and strong theory, and our theoretical communication complexity is the current SOTA for the problem we consider. This alone, we believe, is significant, irrespective of whether the practical performance follows the theory.
>
> - However, we provide a few experiments which do illustrate that this is the case. We did not need to fish for the problem to get the desired result: all we had to do is to set up a problem where we can control for Hessian variance, and then observe if the communication complexity of MARINA with PermK improves as $L_{\pm}^2$ decreases, and whether it improved compared to MARINA used with RandK. We indeed observed this in all experiments we performed. This is not exhaustive, true, but no finite set of experiments ever can be.
>
> - We wish to point out that our work is not primarily empirical. If it was, and there was no supporting theory, then we would agree that a much higher bar should be used to judge whether the experimental findings are robust enough to justify that there is some underlying phenomenon at play or not. However, our theory is conclusive: the theoretical effect is clear. In some sense, one theorem equals an infinity of experiments. Of course, worst-case theoretical predictions are necessarily conservative. Still, the qualitative predictive power of our theory is, we believe, clearly demonstrated nevertheless.
>
> - Re Appendix: We are happy about all our results. While we managed to summarize all key results in the main body of the paper, we had to move the proofs to the appendix, and we also included a few additional results there, too. For example, we consider the Polyak-Lojasiewicz condition, and in that case, we show that our method always converges linearly.  You may also wish to look at the proof of the main theorem to see how the AB inequality and the Hessian variance beautifully come together to yield the improvement. We also recommend additional examples of permutation-based compressors - we did not have space to describe these in the main body of the paper. We believe there are some interesting constructions there going beyond PermK. Finally, we recommend reading the analysis of the complexity bounds supporting the results summarized in Table 4.
>
> Thank you very much again for reading our work and engaging with us through your review.
>
> Authors

---

### Official Review · Reviewer_YEbP · 2021-10-31

**Correctness:** 3
**Technical Novelty And Significance:** 2
**Empirical Novelty And Significance:** 1
**Recommendation:** 5
**Confidence:** 3

**Main Review:**

The proposition of studying correlated compressors in a distributed environment is good, but I'm a little confused on the motivation -- why is the hypothesis limited on MARINA but not on other algorithms? To me the hypothesis is fairly simple and interesting: in a distributed setting, workers follow parameter server architecture and communicate via compressed gradients, how will the correlation in the compressors affect the convergence? This should be independent to the algorithm itself. Actually, some analysis have already been done in the literature, for instance (Acharya et al., 2019)(Acharya et al., 2020), which are analyzed in more general cases. Same question for the Hessian variance part: could you elaborate why this is specific for MARINA? Understanding a newly proposed algorithm is great, but the contribution of a paper seems limited if the results only hold on one algorithm.

The idea of PermK, where unbiased sparsifier is constructed via some random state and selected coordinates are enlarged, seems to be overlapped with (Jianqiao et al., 2017). In that paper, unbiased gradient sparsification is guaranteed in a similar manner. The similar idea can also be found in (Wang et al., 2018). It's unclear to me about the novelty on this method.

The experiments are done on linear tasks, but it's unclear what hypothesis they are verifying. In the main paper, a new complexity bound on the iteration is shown, but in the experiments all the X-axis are number of bits. They only match when the same compressors are used in each run and number of bits in each case grow linearly with the iteration at the same rate.


- Reference

Jianqiao Wangni, Jialei Wang, Ji Liu, and Tong Zhang. *Gradient Sparsification for Communication-Efficient Distributed Optimization* https://arxiv.org/pdf/1710.09854.pdf

Jayadev Acharya, Clément L. Canonne, Himanshu Tyagi. *Inference under Information Constraints I: Lower Bounds from Chi-Square Contraction* https://arxiv.org/pdf/1812.11476.pdf

Jayadev Acharya, Clément L. Canonne, Himanshu Tyagi. *Communication-Constrained Inference and the Role of Shared Randomness* http://proceedings.mlr.press/v97/acharya19a/acharya19a.pdf

Hongyi Wang, Scott Sievert, Zachary Charles, Shengchao Liu, Stephen Wright, Dimitris Papailiopoulos *ATOMO: Communication-efficient Learning via Atomic Sparsification* https://arxiv.org/pdf/1806.04090.pdf

**Summary Of The Paper:**

This paper extends the analysis to an existing algorithm called MARINA, and proposes two results: 1) Correlated compressors among the workers, and one realization called PermK; In the analysis, an inequality called AB inequality is used to study the compression variance. 2) A new metric called Hessian variance to refine the results of MARINA.

**Summary Of The Review:**

Could you take a look at my main review and address the concerns on motivation, PermK and experiments?

---

> ### Author Response · Authors · 2021-11-22
> **On correlated compressors 1**
>
> > The proposition of studying correlated compressors in a distributed environment is good, but I'm a little confused on the motivation -- why is the hypothesis limited on MARINA but not on other algorithms? To me the hypothesis is fairly simple and interesting: in a distributed setting, workers follow parameter server architecture and communicate via compressed gradients, how will the correlation in the compressors affect the convergence? This should be independent to the algorithm itself.
>
> *Response:*
>
> We can provide several answers to your question. All are valid, but each addresses your question from a different point of view. We will be happy to update the text of the paper with suitable comments along these lines as we agree the answers are potentially interesting to the reader. On the other hand, we do not think that this is necessary as this does not address the heart of our contribution.
>
> 1) Our main object of study in this paper is **distributed nonconvex smooth optimization/learning in a regime where communication is the main bottleneck.** We argue that distributed optimization is of critical importance as it is the main paradigm for training supervised ML models with a large-enough number of training examples and parameters. Also, modern DNN models are nonconvex. So, the problem is clearly of the highest interest in modern ML literature. In this problem class, the MARINA method of Gorbunov et al (2021) is currently the method achieving the SOTA theoretical communication complexity. In this paper, we provide a more general (due to the AB assumption, which can model arbitrarily correlated unbiased estimators and not merely independent unbiased estimators) and refined (due to the inclusion of the effect of Hessian variance, which to the best of our knowledge is a new notion first defined in this paper) analysis of MARINA, without any additional assumptions on the problem class (as Gorbunov et al, we only require Lipschitzness of the gradients $\nabla f_i$, and lower-boundedness of $f$). We are able to show theoretically that suitably designed correlated compressors (e.g., PermK) improve upon the uncorrelated (in fact, independent) compressors used by Gorbunov et al. We improve upon the theoretical SOTA, and our improvement is very substantial: by the factors of $O(\sqrt{n})$ or $O(1+d/\sqrt{n})$ in the $d\geq n$ and $d\leq n$ regimes, respectively. These are huge factors, and this can be observed in our experiments as well. **So, at the moment, MARINA with the PermK compressor is the new theoretical SOTA in terms of communication complexity for the important problem class we consider.** In *this* sense, it does not matter whether the notions of AB inequality or Hessian variance have uses outside of the scope of our paper. **Hence, we are of the opinion that the criticism raised here is not really relevant to our key contribution. We believe it should be merely seen as an admittedly interesting, yet ultimately tangential question, one which we are happy to answer, but one that is orthogonal to our contributions, and hence should not have a bearing on whether our paper should or should not be accepted.** We say this with the utmost respect to the reviewer. We believe the suggestion to answer the questions the reviewer is asking is a good one, and we are most happy to do so. However, we felt the need to be very clear about our view of the importance of these questions.

---

> ### Author Response · Authors · 2021-11-22
> **On correlated compressors 2**
>
> 2) You are right that the idea that correlated compressors might help to improve the theoretical communication complexity of distributed optimization methods has very broad applicability. This is an advantage of our proposal and not a disadvantage. We have uncovered something new, a phenomenon that has the potential to advance the theoretical SOTA for different problem classes from the one we focus on in this paper. For example, when we started our research, we first studied distributed gradient descent with compressed communication in the strongly convex regime -- a problem class very different from the one we ultimately cover in this paper. It is known that gradient descent with compressed gradients converges linearly to a neighborhood of the optimal solution (Khirirat et al, 2018). We designed a very naive (our first) variant of PermK (which arises as a special case of our PermK in the $n=d$ setting) and proved the convergence of the new method. We observed improvement in the theory. While the PermK correlation helped for this problem, the analysis does not depend on the notion of Hessian variance. After several months of additional research, strongly encouraged by these early results, we turned our attention to the much more challenging and timely (and relevant to the ICLR community) problem of **nonconvex** optimization. In the non-distributed setting, SGD methods that use specifically designed **biased** gradient estimators (e.g., SARAH, SPIDER) have better complexity than those that do not (e.g., SVRG). We knew that if we wanted to advance the theoretical SOTA in this regime, we needed to rely on biased estimators as well. Indeed, prior to our work, the theoretical SOTA in this regime is obtained by the MARINA method, and MARINA indeed uses a biased gradient estimator. Moreover, MARINA relies on the idea of **compressing gradient differences.** The idea of compressing gradient differences first appeared in the MARINA paper and is very different from the more simplistic approaches in the papers you cite. A somewhat similar strategy was proposed before in the DIANA algorithm of Mishchenko et al (2019), but their gradient estimator is unbiased, and one compresses the difference of the true gradient and a certain gradient estimate as opposed to a true gradient. When proposing MARINA, Gorbunov et al (2021) improved upon the previous theoretical SOTA, which happened to be DIANA. So, we knew that if we wanted to make a theoretical breakthrough in the very important area of communication efficient **nonconvex** distributed optimization, we knew we had to improve upon MARINA. We studied MARINA deeply and tried to see whether it can, too, benefit from correlated compressors such as PermK. We succeeded in the end, but along the way, we discovered the general class of arbitrarily correlated unbiased compressors (defined via the AB inequality) and the notion of Hessian variance. Both of these notions are new and are critical in our analysis.

---

> ### Author Response · Authors · 2021-11-22
> **On correlated compressors 3 (plus Hessian variance)**
>
> > Same question for the Hessian variance part: could you elaborate why this is specific for MARINA? Understanding a newly proposed algorithm is great, but the contribution of a paper seems limited if the results only hold on one algorithm.
>
> *Response:*
>
> 3) Note that **the notion of Hessian variance directly encodes the behavior of gradient differences. This is not a coincidence. Indeed, MARINA operates by compressing gradient differences, and hence it is expected that some properties of gradient differences will play a role.** However, the original analysis of MARINA by Gorbunov et al did not rely on any special structural insight into the problem class that would depend on gradient differences. Yet, reliance on gradient differences is critical behind the theoretical success of MARINA - it is precisely this that lends the method its variance reduction property.
>
> In summary, **we are fully convinced that the scope of many distributed methods relying on communication compression can be extended from independent to correlated compressors and that the AB inequality will play a key role.** Yet, we can't analyze dozens of algorithms in a single paper. We picked a method (MARINA) and setting (nonconvex) which we believe is most important to the ML community in general and the ICLR community in particular. We stand behind our choice. **Moreover, we believe the novel notion of Hessian variance will also have a wider appeal.** Again, this is not a weakness of our paper. Instead, we believe this is a strength as besides establishing new theoretical SOTA for an important problem of interest we give inspiration to other researchers to apply the insights and tools (e.g., the AB inequality and Hessian variance) proposed in our work in other domains.

---

> ### Author Response · Authors · 2021-11-22
> **The literature you cite 1**
>
> > Actually, some analysis have already been done in the literature, for instance (Acharya et al., 2019)(Acharya et al., 2020), which are analyzed in more general cases...
>
> *Response:*
>
> - **The paper "Jianqiao Wangni, Jialei Wang, Ji Liu, and Tong Zhang. Gradient Sparsification for Communication-Efficient Distributed Optimization" does not contain any convergence theory and this alone means that this work is really not directly relevant to our contributions at all.** It is an interesting paper in the broader field of communication compression, but its focus is on forming sparsification strategies via solving certain optimization subroutines. These strategies are theoretically studied on their own and are not shown to work in conjunction with any optimization method. So, their work does not predict any theoretical improvement. Moreover, they study compression of gradients (or stochastic gradients), which is strictly suboptimal to the reliance on gradient differences, and relies on unbiased sparsification. None of this is surprising - the paper is from 2017, and hundreds of papers have been written since. In our work, we focus on improving the **current theoretical SOTA** instead.
>
> - **The paper "Jayadev Acharya, Clément L. Canonne, Himanshu Tyagi. Inference under Information Constraints I: Lower Bounds from Chi-Square Contraction" is not even a paper from the same field at all.** These authors study "sample complexity of learning and testing discrete distributions in this information-constrained setting". The authors say in their abstract: "Underlying our bounds is a characterization of the contraction in chi-square distance between the observed distributions of the samples when information constraints are placed." There is only a very superficial connection here (they are also concerned with communication complexity, but very different problems), and we do not see how this paper is relevant to our work at all. If the reviewer believes some parts are relevant, we would want to hear which parts and exactly how. If there is sufficient relevance, we will be most happy to mention the paper in our work. However, we wish to stress again that this paper is addressing a non-optimization problem, and hence its results are irrelevant to distributed optimization, which is what we study here. Having had a read, we believe though it seems to be a nice paper.
>
> -  The paper **Jayadev Acharya, Clément L. Canonne, Himanshu Tyagi. Communication-Constrained Inference and the Role of Shared Randomness** is of the same category as the above paper. Not relevant to our work.

---

> ### Author Response · Authors · 2021-11-22
> **The literature you cite 2**
>
> > The idea of PermK, where unbiased sparsifier is constructed via some random state and selected coordinates are enlarged, seems to be overlapped with (Jianqiao et al., 2017). In that paper, unbiased gradient sparsification is guaranteed in a similar manner. The similar idea can also be found in (Wang et al., 2018). It's unclear to me about the novelty on this method.
>
> *Response:*
>
> - The first comment offered above is very superficial and misses the point. Yes, we work with **unbiased compressors**, and this class includes **sparsification** operators as well. Unbiased sparsification (e.g., Rand-$K$) is constructed in a standard way: i) by zeroing out $d-K$ coordinates chosen uniformly at random, and ii) scaling up the remaining $K$ coordinates by the factor $d/K$, which makes the compressor *unbiased*. Virtually every paper that works with unbiased sparsification operators, and there are hundreds if not thousands of them, will use this standard construction. Pointing out this similarity amounts to pointing out something that is really irrelevant. The differences is what matters! Note that one of the key differences in our paper compared to prior theoretical work on distributed nonconvex optimization is the use of carefully designed **correlated** unbiased sparsifiers and their combination with the current SOTA method in terms of theoretical communication complexity: MARINA. Correlation is what is important. But not *any* kind: the first part of Lemma 1 points to what happens when the correlation is bad: $A$ will be large, and $B$ will be small, the exact opposite of what we want the parameters of the AB inequality to be.
>
> - The work (Jianqiao et al., 2017) not only **does not contain any convergence analysis** for the simplistic (from the point of view of what we know now - 4 years later) method they study (Alg 1 in their paper = distributed sparsified SGD), but they also design their compressor for each node of the network **independently** in the sense that the optimization problem they solve to define their compressor (problem (4) in their paper) on each node is independent of what is happening on the remaining compute nodes. They optimize the expected sparsity given a budget on variance, and they do this separately on each node. This is a very different strategy from ours. Indeed, our goal is almost orthogonal: we do not want each node to be greedy and optimize their own individual variance. Instead, we take a global view, and require the variance of the average of the compressed vectors to be small. We do this without relying on any surrogate optimization procedure, and we specifically want the variance to be exactly zero in the regime when all aggregated vectors are identical.
>
> - **The ATOMO paper, which we are very well aware of, also **does not establish any convergence let alone communication complexity bounds** and in *this* sense is irrelevant to the main contribution of our paper.** ATOMO merely focuses on designing a particular compression strategy based on atomic decomposition and sparsification. Unlike the above paper, their strategy *will* in general lead to some correlation among the compressors. However, their sparsifier remains a heuristic since no convergence or complexity guarantees are offered. **Their compressor is independent of whether one considers convex or nonconvex optimization - they do not consider optimization, instead, they focus on designing a compression strategy independent of any optimization method.** In sharp contrast, our goal is to establish a new theoretical SOTA in terms of communication complexity for distributed nonconvex optimization.
>
> **In summary, we are quite surprised about the proposed/cited papers brought up by the reviewer. They are all irrelevant to our key contributions, and are at best very broadly relevant, in the same way, hundreds of other papers on the broad topic of communication efficiency is relevant.**
>
> The most relevant paper to our work is the current theoretical SOTA in terms of communication complexity for distributed nonconvex optimization - and that method is MARINA. If the reviewer believes a different method is the current theoretical SOTA, please let us know which paper you have in mind. **If you agree MARINA is the SOTA, then since we can improve on it, MARINA with PermK becomes the new SOTA, and that is highly significant.**

---

> ### Author Response · Authors · 2021-11-22
> **On Experiments**
>
> > The experiments are done on linear tasks, but it's unclear what hypothesis they are verifying. In the main paper, a new complexity bound on the iteration is shown, but in the experiments, all the X-axis are number of bits. They only match when the same compressors are used in each run and the number of bits in each case grows linearly with the iteration at the same rate.
>
> *Response:*
>
> - First, we believe that a strong theoretical work does not require any experiments in the same way as strong empirical work does not require any theory. Of course, it is even better when theoretical papers can be accompanied with at least illustrating experiments showcasing the prediction power of the theory, and when empirical work can be supported by at least some theoretical justification. We believe our work offers very strong theoretical advances (new theoretical SOTA for one of the most important problems in modern supervised machine learning: communication efficient distributed training in the nonconvex regime), and as such, does not need empirical evaluation. Nevertheless, we provided empirical evaluation which does support our theoretical predictions, and that means that the bounds we obtain and quantities we study (e.g., Hessian variance) are truly insightful and meaningful.
>
> - The $x$ axis in the plots shows the quantity that is most relevant to our paper: the total number of communicated bits per 1 node (= number of iterations times the number of bits communicated per node per iteration). This is relevant since we are precisely focusing on communication efficient optimization in the regime when the network/communication speed is the major bottleneck. In such a regime, it is the total number of communicated bits that is the most relevant quantity - and this is also the theoretical quantity we study: see Table 4. Note that the communication complexities in Table 4 arise by multiplying the theoretical number of iterations (from Table 3) by the number of bits sent in a single iteration per node (calculations are done in Appendix L - we state this just before Section 4.1). **So, unlike what the reviewer claims, what we show in the plots has a perfect correspondence to our theory.** **Each plot represents the dependence between the norm of the gradient (or function value) and the total number of transmitted bits per node.**

---

> ### Author Response · Authors · 2021-11-28
> **We have responded with 6 substantial posts... We did not hear from you so far.**
>
> Dear Reviewer YEbP,
>
> Thanks again for your review.
>
> - **We have responded to your concerns with 6 substantial posts. As you will see, no criticism related to the heart of our paper (our theoretical contribution) is valid. We have also explained what our experiments show - we did test a hypothesis, and the test was successful. Again, this criticism is not valid.**
>
> - I addition, we responded to the concerns of all other reviewers, and we believe we addressed all misunderstands. Please read them, we will be thankful!
>
> - **Please let us know if you agree that we cleared up your concerns. If yes, then please raise the score appropriately. If not, please let us know what concern we did not address and why. There is just one day left for us to discuss these issues with you.**
>
> Thank you!
>
> Authors
>
> PS: Unfortunately, it is clear to us from your review that you did not understand the essence of our work. You suggested comparing to irrelevant work (we argued in detail that this is the case), and raised truly bizarre concerns about basic concepts from the literature of unbiased compressors (It seems you think we reinvented the notion of an unbiased sparsifier, and that this is our contribution? No one even remotely familiar with the field would make such a basic misjudgment.). In our view, you failed to understand our work and engage with it scientifically. We kindly request that you either change the self-reported confidence to "educated guess", or, if you do understand our response and change your mind, please consider raising your score substantially. **Our work established new theoretical SOTA for an important problem, and as such deserves, in our view, substantially higher scores.**
>
> We apologize for the assertive tone, but we need to defend our work against what we believe is a very superficial, uninformed, and misleading review. Please do not mistake this for an ad hominem attack. What we are unhappy with is merely the review text and its low quality.

---

### Official Review · Reviewer_wJjy · 2021-11-02

**Correctness:** 4
**Technical Novelty And Significance:** 2
**Empirical Novelty And Significance:** 2
**Recommendation:** 6
**Confidence:** 3

**Main Review:**

Improving communication complexity is an important topic in distributed optimization. The paper tried to extend the analysis of the SOTA algorithms in a more detailed case to refine the complexities. The paper is clearly written and easy to read.

My main concern are as follow:
1. The analysis (Theorem 4) here seems to be heavily relied on the AB inequality and correspondently Hessian variance assumption, but I am not sure whether the Hessian variance is motivated enough. Now authors verify the assumption holds on only identical functions and quadratic functions, I am not sure whether it is applicable to more general problems.

2. For the experiments, does the autoencoder example satisfy the extra assumptions?

3. For the proof, the difference seems to lie in the Lemma 5 here and derivation above Eq (21) in MARINA paper, where AB inequality and Hessian variance work to bring some difference in the coefficients in my opinion, and the claimed difference of using correlated compressor is incorporated into the coefficient computation of AB inequality, the remaining things are similar, e.g., the Lyapunov function $\Phi$. Can the authors kindly provide more intuition on the benefits of the new analysis, with the additional assumption?

Some minor thoughts:
1. Even though closely related to MARINA paper, but as a separate independent submission, to make it self-contained, I may suggest that at least authors can consider to add a formal description (e.g., an \algorithmic environment or the main update rule) of the algorithm into the main content (rather than mentioning it in Appendix B), I think, at least now, readers should not be required to be familiar with MARINA as they are with SGD before reading the submission.

All in all, now I view the contributions here are a little marginal, and skeptical on whether it is suitable as a separate work. I hope to have more insights from authors on the importance and practicality of the additional assumptions. Please definitely indicate here if I misunderstand any point. I will appreciate the authors to address my confusions, and definitely reconsider my decision. Thank you very much.

**Summary Of The Paper:**

This paper studied MARINA method (proposed in a previous work) in distributed nonconvex optimization problems. With a new assumption on the Hessian variance, the paper extend the original analysis of MARINA to accomodate correlated compressor, which avoid the independent compressor setting and can lead to improvement to the communcation complexity in a case when the number of dimension is larger than that of component.

**Summary Of The Review:**

The paper improves the analysis of MARINA algorithm, but the introduced new assumptions need to be further justified.

---

> ### Author Response · Authors · 2021-11-22
> **Response to Concern #1**
>
> > The analysis (Theorem 4) here seems to be heavily relied on the AB inequality and correspondently Hessian variance assumption, but I am not sure whether the Hessian variance is motivated enough. Now authors verify the assumption holds on only identical functions and quadratic functions, I am not sure whether it is applicable to more general problems.
>
> *Response:*
>
> - **The fact that our analysis relies on the AB inequality is not an issue, but an advantage**, because it **broadens** the class of compressors we can analyze in conjunction with MARINA. Indeed, virtually all papers on unbiased compressors, including the MARINA paper of Gorbunov et al (2021) we improve in this paper, work with **independent** compressors across the nodes. The AB inequality includes these as a special case, but allows to work with **arbitrary dependence structures**. We give examples of provably useful dependence: this leads to the **new notion of input variance compressors** in general, and permutation (PermK)  compressors in particular.
>
> - We wish to stress, and we already mentioned this in the paper, that the **Hessian variance assumption** is **not** an additional assumption we need to make in the sense that this would limit the applicability of MARINA when contrasted with its scope from Gorbunov et al (2021). Quite to opposite is true! **We show that under the exact same assumptions on the functions used by Gorbunov et al (2021) in their MARINA paper, there is a well defined hidden quantity, which we call Hessian variance, which can be used/tapped to design theoretically better compression mechanisms (e.g., PermK).** So, the Hessian variance assumption is **not** limiting the scope of our paper in any way.
>
> - Our analysis of MARINA is both more general (in the sense that we prove it works with all compressors satisfying the AB inequality, which is a larger class of compressors than the unbiased independent compressors studied before), and more accurate/precise (in the sense that it can take advantage of the situation when the Hessian variance is better than its worst-case value of $L_+^2$ dictated by Lemma 2) - contrast the first two rows of Table 3 (our result includes the Gorbunov et al result as a special case). Admittedly, we get most advantage when Hessian variance is zero - but this is merely an extreme case we mention for pedagogical/clarity reasons. We do not claim this is what will happen in practice. However, it is important to understand this extreme regime and the potential benefits it can provide before one can appreciate what happens in the more realistic regime when Hessian variance is neither too large, not too small.
>
> - **It is (obviously!) not true that we only prove that the Hessian variance "assumption holds on only identical functions and quadratic functions"**. First of all, this is not an extra assumption, as we explained above. Indeed, Lemma 2 says that $L_{\pm}^2$ is well defined and finite whenever each function $f_i$ has $L_i$-Lipschitz gradient, in which case it is bounded above by $L_+^2$ (and $L_+^2$ is itself bounded above by $\frac{1}{n}\sum_i L_i^2$). So, the Hessian variance is a  "hidden" quantity that always has **some** finite value between $0$ and $L_+^2$, and depending on how small it is (the smaller the better), we get more or less improvement. So, it is **not** true that we only show that the Hessian variance "assumption holds on only identical functions and quadratic functions". It holds automatically whenever each $f_i$ has $L_i$-Lipschitz gradient. Moreover, in Theorem 3 we give expressions for Hessian variance in the case of twice continuously differentiable functions, shedding more light on this new quantity.
>
> **The criticism raised here is factually incorrect.** We hope we explained why in a satisfactory manner. However, if any further questions remain, please do not hesitate to reach out to us. We will be most happy to explain.
>
> Authors

---

> ### Author Response · Authors · 2021-11-22
> **Response to Concern #2**
>
> > For the experiments, does the autoencoder example satisfy the extra assumptions?
>
> *Response:*
>
> - We explained in our response to Concern #1 that the reviewer is factually incorrect when he/she thinks that we introduce stronger assumptions in our paper compared to those used in the MARINA paper of Gorbunov et al (2021). **We do not need any extra assumptions**. In light of this, the question raised by the reviewer here does not make sense.
> - In summary, Hessian variance is well defined as soon as each $f_i$ has Lipschitz gradient. This is a standard assumption made in virtually all (but not all) papers on gradient-type methods.
>
> **This remark is not an issue with our paper but comes from a misunderstanding by the reviewer.** We hope we clarified this in our response.
>
> Authors

---

> ### Author Response · Authors · 2021-11-22
> **Response to Concern #3**
>
> > For the proof, the difference seems to lie in the Lemma 5 here and derivation above Eq (21) in MARINA paper, where AB inequality and Hessian variance work to bring some difference in the coefficients in my opinion, and the claimed difference of using correlated compressor is incorporated into the coefficient computation of AB inequality, the remaining things are similar, e.g., the Lyapunov function  $\Phi$. Can the authors kindly provide more intuition on the benefits of the new analysis, with the additional assumption?
>
> *Response:*
>
> - **Our analysis is similar to the analysis of MARINA. It differs in a few but very critical places and this difference leads to a very large effect in the generality of MARINA (we allow for arbitrarily correlated unbiased compressors) and in the theoretical complexity of MARINA through the use of our newly proposed PermK compressors** (see also Tables 3 and 4). The new notions of AB inequality and Hessian variance work in a **beautiful tandem** with the original MARINA proof and uncover a new regime when theoretical complexity can be substantially improved. The best-case improvement factors (see the abstract) are enormous.
>
> - This is the kind of result we as theory researchers appreciate a lot. The small but critical change to what we know, leading to a massive improvement. We propose that our results be judged according to the "distance between the complexity results" we obtain, i.e., by the improvement in the theoretical communication complexity, rather than by the "distance" between the proofs. **We believe we achieved something mathematically beautiful here**: we achieve a very large effect through a deep observation which ultimately (after we cleaned our proof techniques several times, the last of which you see in our paper) necessitated only small differences in the MARINA proof. **On the way to achieve this, we discovered a new notion (Hessian variance) that can be of independent interest and discovered a new and very broad class of unbiased compressors (those satisfying the AB inequality), which can also be of independent interest.** In fact, we already know that both of these concepts are applicable in other settings, are actively pursuing these research directions.
>
> - The basic intuition is as follows. It is not optimal for the $n$ nodes to apply independent compressors (as done in original MARINA) in the regime when the data stored across the $n$ nodes is nearly homogeneous. **Our proposal is that the nodes should instead collaborate and that a "smart" collaboration can lead to a much better gradient estimator** (i.e., with smaller variance). In the extreme case when the data is fully homogeneous (i.e., when $f_i=f_j$ for all $i,j$), PermK is able to reconstruct the gradient exactly, with zero error! This is where its strength lies. The more similar the data is, the better PermK becomes when compared to RandK.
>
> - Please note that **we do not require any additional assumptions** to obtain our improved communication complexity results. This is an important strength of our paper which seems to have been misunderstood.
>
> We are most happy to add more clarifications to the camera-ready version of the paper!
>
> Authors

---

> ### Author Response · Authors · 2021-11-22
> **On "Minor thoughts"**
>
> > Even though closely related to MARINA paper, but as a separate independent submission, to make it self-contained, I may suggest that at least authors can consider to add a formal description (e.g., an \algorithmic environment or the main update rule) of the algorithm into the main content (rather than mentioning it in Appendix B), I think, at least now, readers should not be required to be familiar with MARINA as they are with SGD before reading the submission.
>
> **We completely agree!**
>
> - We indeed wanted to include MARINA in the main body of the paper, but ultimately decided to move it to the appendix due to the page limitations. This way we were able to better explain what is novel in our paper.
> - However, in the camera-ready version of the paper, should it be accepted, we will certainly add MARINA to the main body. This would be the first change we would make with the added space provided.
>
> Authors

---

> ### Author Response · Authors · 2021-11-22
> **On "All in all"**
>
> > All in all, now I view the contributions here are a little marginal, and skeptical on whether it is suitable as a separate work. I hope to have more insights from authors on the importance and practicality of the additional assumptions. Please definitely indicate here if I misunderstand any point. I will appreciate the authors to address my confusions, and definitely reconsider my decision. Thank you very much.
>
> *Response:*
>
> **We truly appreciate the reviewer saying** "Please definitely indicate here if I misunderstand any point. I will appreciate the authors to address my confusions, and definitely reconsider my decision. Thank you very much." **We do, indeed, believe that there are some serious misunderstandings here, and we did our best to explain the source of the misunderstanding, and to clarify things.**
>
> Please do not hesitate to ask more questions!
>
> We shall now address the "marginal contributions" claim:
>
>
>
> **We respectfully disagree that the contributions are marginal.**
>
> - **An improvement of up to $O(\sqrt{n})$ (when $d\geq n$) or up to $O(1+d/\sqrt{n})$ (when $d\leq n$) is huge!** In typical situations, the number $d$ of weights in a NN is much larger than the number of data points (they are over-parameterized), and the number of machines $n$ is much smaller than the number of data points. So, we are typically in the $d\gg n$ regime. However, **if the number of machines is, for example, $n=100$, we get an order of magnitude improvement. In federated learning, $n$ is much larger (tens of thousands or even millions), and the improvement will become $100\times$ to $1000\times$.**
>
> - In experimental deep learning,  an improvement (in generalization) of a few percentage points is considered massive. We get theoretical improvement which can be arbitrarily large (depending on the values of $n$ and $d$).
>
> Authors

---

> ### Comment · Reviewer_wJjy · 2021-11-26
> **Thank you and further clarification**
>
> I truly appreciate authors for the effort of addressing my confusions, while I still have some confusions.
>
> ---
>
> First I think I need to further clarify that the "additional assumption" in my previous review should be that *the Hessian variance is **zero** (or **significantly smaller** than $L_+$, so that we can drop the $L_\pm$ term in table 4)*. **With this setting on $L_\pm$**, the claimed improvement will be attained as mentioned in Section 4.1, i.e., $O(\sqrt{n})$ when $d\geq n$ and $O(1+d/\sqrt{n})$ when $\sqrt{n}\leq d \leq n$.
>
> In fact authors have responded to my question in the rebuttal to Reviewer mZxC ("Addressing Concern #1"), but it is still unclear that what will be the value of $L_\pm$. Also as theory researchers, not only the case $L_\pm^2=L_+^2$, **but also the case $L_\pm^2=O(1)L_+^2$** is least favorable to us in theory, they should be different in order, rather than only the "maximum possible value".
>
> ---
>
> Also that is why I still have questions 2 on Example 2, in fact what I want indeed is that, **what are the values of $L_\pm$ and $L_+$ of the objective function?** It would be better if it can be provided (derived) explicitly (or try to empirically estimate it), similar to that in Example 1 (Section 5.1). With the value of $L_\pm$, I believe the paper will definitely be enhanced.
>
> ---
>
> In summary, I value the submission in terms that authors proposed an new quantity $L_\pm$ that may plays a role in the convergence analysis, and improves the complexity with additional settings on $L_\pm$ (i.e., $L_\pm$ significantly smaller than $L_+$). While authors provided some clues that the new quantity will truly make sense (table 2), but the clue looks preliminary because it is based on simple examples, so now it is unclear that $L_\pm$ will really play a role (i.e., avoid the $L_\pm^2=O(1)L_+^2$ regime).
>
> I am not sure whether authors agree, but In my opinion, the flow of the submission is, we found Hessian variance, and it affects/improves complexity. While Hessian variance, as the core concept in the submission, should be the main focus of the paper and studied more in depth, e.g., add more discussion on deriving (or estimating) $L_\pm$ on more non-trivial and important examples in machine learning. As authors mentioned, *"hope that our work will motivate other researchers to study this quantity further"*, but as the first work, it would be better to contain more nontrivial examples to justify $L_\pm$ significantly smaller than $L_+$, to truly further motivate the research.
>
> As an example (may not be totally appropriate, but the flow should be similar I think), in *Mei, Jincheng, et al. "Leveraging non-uniformity in first-order non-convex optimization." ICML 2021*, they identified nonuniform smoothness/PL/KL property and proposed corresponding algorithms, while they verified the condition holds in some RL and GLM problems, which looks to be non-trivial. With the comparison, that's why I think the current submission can be further improved.
>
> ---
>
> So with the clarification, I respectfully still have some confusions on the Example 2. Also my question 1 is already answered in the rebuttal to Reviewer mZxC, while I am not sure whether my confusion is fully addressed, because I think $L_\pm=O(1)L_+$ case is also least favorable in theory.
>
> Thank you very much.

---

> > ### Author Response · Authors · 2021-11-27
> > **Response to "Thank you and further clarification" - part I**
> >
> > > I truly appreciate authors for the effort of addressing my confusions, while I still have some confusions.
> >
> > Answering your questions is our job! Please ask away!
> >
> > > First I think I need to further clarify that the "additional assumption" in my previous review should be that the Hessian variance is zero...
> >
> > Thanks for clarifying what you meant. We are glad that our response to Reviewer mZxC ("Addressing Concern #1") addressed the concern you had.
> >
> > Your remaining concern here is: "but it is still unclear that what will be the value of $L_{\pm}$". Let us address it now. We offer several answers, from different point of view:
> >
> > 1) **Hessian variance can attain virtually any value - quadratic functions example.** Let us give a simple illuminating "pedagogical" example that will show that it is possible for $L_{\pm}$ to attain virtually any value smaller than $L_+$. Consider functions $f_i(x)=\frac{1}{2}a_i x^2$, where $x\in \mathbb{R}^d$, $d=1$, and $a_i \geq 0$ are the Hessian matrices (in this case, $1\times 1$ matrices). This is a special case of Example 3 from our paper. Theorem 3 from our paper says that $L_{\pm}^2 = \frac{1}{n}\sum_{i=1}^n a_i^2 - \left(\frac{1}{n}\sum_{i=1}^n a_i\right)^2$ and $L_{+}^2 = \frac{1}{n}\sum_{i=1}^n a_i^2$. So, while $L_{\pm}^2$ is the *variance* of the values $a_1,\dots, a_n$, $L_{+}^2$ is the *average of their squares*. We will now show that it is possible  for $L_{\pm}^2$ to attain *any* value in the interval $[0,(1-\frac{1}{n})L_{+}^2]$.
> >
> > **Proposition.** Let $f_i(x)=\frac{1}{2}a_i x^2$ for $i = 1,2,\dots,n$. Then $0 \leq L_{\pm}^2/L_+^2 \leq 1-\frac{1}{n}$ (and both bounds are tight). Moreover, choose any $0\leq \theta \leq 1- \frac{1}{n}$. Then there exist nonnegative constants $a_1,\dots, a_n$  such that $L_{\pm}^2/L_+^2 = \theta$.
> >
> >
> > **Proof of the Proposition:**  Since the values $a_1,\dots, a_n$ are nonnegative, it is easy to see that $\left(\frac{1}{n}\sum_{i=1}^n a_i\right)^2 \geq \frac{1}{n^2}\sum_{i=1}^n a_i^2,$ which implies that $ L_{\pm}^2/L_+^2 \leq 1 - \frac{1}{n}$. It is possible to argue that this bound is tight (we have the proof but do not include it here - ask if you need to see it!). Let us continue to the second part of the statement. Let $e\in \mathbb{R}^n$ be the vector of all ones. It is easy to see that as long as $\frac {1}{\sqrt{n}} \leq \delta \leq 1$, it is possible to choose vector $a = (a_1,\dots,a_n)\in \mathbb{R}^d$ with nonnegative entries such that  $a^\top  e= \delta ||a|| \cdot ||e||$ (we can argue this in detail if you do not see this as obvious - let us know!).  Now,
> >
> > $$\frac{L_{\pm}^2}{L_+^2} = \frac{\frac{1}{n}\sum_{i=1}^n a_i^2 - \left(\frac{1}{n}\sum_{i=1}^n a_i\right)^2}{\frac{1}{n}\sum_{i=1}^n a_i^2} = 1 - \frac{(\frac{1}{n} a^\top e)^2}{\frac{1}{n} || a||^2} = 1 - \frac{(\frac{1}{n} \delta ||a|| \cdot ||e||)^2}{\frac{1}{n} || a||^2} =  1 - \frac{\delta^2  ||e||^2}{n}.$$
> >
> > Since $||e||^2=n$, we have $\frac{L_{\pm}^2}{L_+^2}=1 - \delta^2$.  QED
> >
> >
> > **Commentary:** Let us comment on the above proposition, which we are happy to add to the paper. We have shown in our paper that $0\leq L_{\pm}^2 \leq L_+^2$. Hence, $0 \leq  L_{\pm}^2/L_+^2 \leq 1$. So, the first part of the proposition actually says that in the above simple setting with 1$d$ convex quadratics, $L_{\pm}^2$ must always be *strictly* better than $L_{+}^2$. Of course, the ratio $1-\frac{1}{n}$ is "close"' to $1$ (unless $n=1$, which is a trivial case we need not consider). However, this is not the main point of the above result. The main point is the second part of the statement, which says that the ratio $L_{\pm}^2/L_+^2$ can take *arbitrary* values in the interval $[0,1- \frac{1}{n}]$. In particular, it *can* be arbitrarily small. We believe this addresses a part of your concern.
> >
> > 2) **Random partitioning leads to lower Hessian variance.** To address your point in a different way, we took the "mg" dataset from https://www.csie.ntu.edu.tw/~cjlin/libsvmtools/datasets/regression.html
> >
> > This dataset as $1,385$ datapoints and $d=6$ features.  We partitioned the dataset into $n=2$ parts (i.e., among 2 machines) randomly and consider the problem $\min_w \frac{1}{2} ||X_1 w - y_1||^2 + \frac{1}{2} ||X_2 w - y_2||^2 $. $(X_1, y_1)$ and $(X_2, y_2)$ refer to the split. In this case, we got $L_+ = 3609.6653$ while $L_\pm = 34.58$. Clearly, it is possible for $L_\pm$ to be much smaller than $L_+$.
> >
> > 3) **Your question is too broad.** We wish to point out that your question "what will be the value of $L_{\pm}^2$" is too broad in our view. We have shown above that there are problems where it *can* have almost any value smaller than $L_+^2$, and that it can be small in practice. We have shown in our paper through synthetic experiments that when it is small, we get an improvement on the original MARINA method. Please let us know what else you would like to us to explain or shed light on.

---

> > ### Author Response · Authors · 2021-11-27
> > **Response to "Thank you and further clarification" - part II**
> >
> > > Also that is why I still have questions 2 on Example 2, in fact what I want indeed is that, what are the values of
> > $L_{\pm}$ and $L_+$ of the objective function? It would be better if it can be provided (derived) explicitly (or try to empirically estimate it), similar to that in Example 1 (Section 5.1). With the value of $L_{\pm}$, I believe the paper will definitely be enhanced.
> >
> > - For difficult-enough architectures and models, it is impossible to derive explicit expressions for even the classical smoothness constants such as $L_-$ and $L_i$, let alone $L_\pm$. So, it is not reasonable to expect that we could do this with Hessian variance and $L_+$, which are somewhat more complicated concepts. We do not view this as a weakness of our results.
> >
> > - Estimation of $L_{\pm}^2$ in the experiment from Section 5.2 is certainly possible. We will add it to the paper (we are running the code right now - but this will take some time; hopefully we can report on this soon) - but we do not share the view that this is very important to do. But it certainly will not hurt! In general, estimation of smoothness constants and of Hessian variance may be costly. However, please note that our method does *not* depend on this estimate in any other way than through the learning rate. It is standard in the literature to tune the learning rate even if all constants defining it are known (as typically such constants are conservative). So, in some sense, one can avoid the estimation problem and instead rely on learning rate tuning.

---

> > ### Author Response · Authors · 2021-11-27
> > **Response to "Thank you and further clarification" - part III**
> >
> > > In summary, I value the submission in terms that authors proposed a new quantity ...
> >
> > - We understand your concern. Let us explain: we proved in our paper that $L_\pm^2$ lies in the interval $[0,L_+^2]$. Or analysis always improves on that of Gorbunov et al (2021), irrespective of what value Hessian variance assumes. But it will be very small (or nil) if it is close to the upper bound, and massive (as advertised in the abstract, for example) if it is close to the lower bound. In general, Hessian variance can essentially take *any* value in this interval, and this depends on the *problem* at hand (e.g., quadratics, logistic regression, NNs), and the *data*. And so our improvement will be somewhere in between. We illustrated on the example of quadratics of the form $f_i(x) = \frac{1}{2}a_i x^2$ that $L_{\pm}^2$ can have any value in the interval $[0,(1-1/n)L_+^2]$. So, there are (quadratic) problems for which our complexity bounds provide massive improvements, and there are also (quadratic) problems for which our complexity bounds merely match (in fact, very slightly improve on) the previous bound of Gorbunov et al (2021). We expect the situation with more complicated problem classes to be similar.
> >
> > - However, not that the values of all of these constants, $L_-$, $L_+$ and $L_{\pm}$, are data-dependent! Indeed, in the quadratic example, we had to construct the values $a_1,\dots, a_n$ (which in an ERM problem depend on the training data!) in order to obtain $L_+$ and $L_\pm$ with desired values. So, **it is not in general possible to characterize problem classes** in which would expect $L_\pm$ to be much smaller than $L_+$. Problem class is not enough: data is what matters here. So, the problem is more difficult than perhaps you appreciated (correct us if we are wrong). Note that we provided **formulas** for $L_\pm$ and $L_+$ for arbitrary twice continuous differentiable functions in Theorem 3. These expressions depend on the problem class and the data. In some sense, an answer to your question could simply be: when $L_{\pm}^2\ll L_+^2$ (using the expressions from Theorem 2), then we expect to get a lot of improvement. However, characterizing *when* this happens is difficult as any potential answer to this must necessarily depend on the properties of the data.
> >
> > - However, you are right that we did not perform any *theoretical* analysis that would *quantify* how much and in what situations will Hessian variance be much smaller than $L_+^2$ beyond Examples 1 and 2  from Section 3, and now also the example of quadratic functions we gave in a previous response. This is an interesting subject of study for future research - but we do not think it is necessary for us to answer this difficult question in *this* paper. This is one of the many directions of research our work opens. We are aware of a dozen other directions which need to be explored, and which are best explored in new papers other than included in our paper. Please note that our paper already has 50+ pages including appendices.
> >
> > > I am not sure whether authors agree, but in my opinion...
> >
> > We do not agree here. All the following aspects of our paper are equally important:
> >
> > - AB inequality (Section 2)
> > - input variance and permutation based compressors which achieve good $A$ and $B$ constants in the AB inequality (Section 2 + appendix)
> > - Hessian variance (Section 3)
> > - theory supporting Hessian variance (Section 3 + appendix)
> > - convergence analysis of MARINA in this general setting (Section 4)
> > - communication complexity analysis of MARINA with PermK compressors (Section 4)
> > - experiments which illustrate that our ideas and bounds have predictive power: they translate to improvements in toy experiments (Section 5)
> >
> > They are all equally important parts of our paper, as hinted by our choice of sectioning. We take steps to develop all these directions. Did we go **all the way** in all of these directions? Certainly not. Perhaps more can be said about AB inequality and input variance compressors, perhaps more can be said about Hessian variance, and so on. But we believe what we did is convincing, and forms a coherent and unified story. We hope that our work will inspire future research.
> >
> > No paper is ever the final word on a topic. This is how science works (we know you know this, of course!).

---

> > ### Author Response · Authors · 2021-11-27
> > **Response to "Thank you and further clarification" - part IV**
> >
> > > As an example (may not be totally appropriate, but the flow should be similar I think) ...
> >
> > - Please note we *also* verified that our conditions hold: we do not require any extra assumptions than those of Gorbunov et al (2021). The authors in the paper you mention need to verify their conditions since they *generalize* the standard notions of smoothness and KL inequality. So, they need to provide nontrivial examples for when their conditions are satisfied. However, we do not generalize the applicability scope of MARINA: our analysis applies to the same setting of lower bounded and differentiable $f$ with $L_i$-Lipschitz gradient for each $f_i$ considered by Gorbunov et al. So, in this sense, the paper you cite is not a good analogy. We just wanted to make sure this point is clear. However, we know what you mean, and we will address this next.
> >
> > - In our previous response (part I) we i) give an example of a quadratic which shows that $L_{\pm}^2$ could be arbitrarily smaller than $L_+^2$. So, the notion of Hessian variance is not interesting merely in the cases when $L_\pm=0$, as is the case in Examples 1 and 2 from Section 3.1. Providing closed form expressions for Hessian variance for non-quadratics is possible, and we do so already in Theorem 3. However, its is difficult to use these to calculate the ratio $L_\pm^2/L_+^2$ explicitly and characterize when it will be small. The key difficulty comes from the fact that the ratio is data dependent - so the answer will not be merely the question of identifying appropriate problem classes (e.g., generalized linear models).
> >
> > In some sense, we feel you ask us to do something that is perhaps not even possible.  We would appreciate if you could shed more light on what exactly you would want us to do. If we can do it, we certainly will!

---

> > ### Author Response · Authors · 2021-11-27
> > **Response to "Thank you and further clarification" - part V**
> >
> > > Also that is why I still have questions 2 on Example 2...
> >
> > We've now managed to run more computations w.r.t. the experiment from Section 5.2, as you requested. Instead of working with the notions $L_+^2$ and $L_{\pm}^2$ as defined in the paper, we work with their "adaptive" / "instantaneous" variants, which we call $\left(L^t_+\right)^2$ and $\left(L^t_{\pm}\right)^2$, which are easier to estimate. For example, $\left(L^t_{\pm}\right)^2$ is defined as the constant for which
> >
> > $$ \frac{1}{n}\sum_{i=1}^n || \nabla f_i(x^t) -  \nabla f_i(x^{t+1}) ||^2 - || \nabla f(x^t) - \nabla f(x^{t+1})||^2 = \left(L^t_{\pm}\right)^2 ||x^{t} - x^{t+1}||^2.$$
> >
> > We obtain this by choosing $x=x^t$ and $y=x^{t+1}$ in inequality (8) defining $L_\pm^2$.
> >
> > In particular, we estimate how the "instantaneous" ratio $\left(L^t_+\right)^2/\left(L^t_{\pm}\right)^2$ changes during the optimization process of MARINA with PermK on the autoencoder task (see Section 5.2 and Figure 2).
> >
> >  We have the plots showing the evolution for each iteration. However, since we can't show the plots here, we instead show a coarse summary of these plots in the form of a table:
> >
> >
> > |               |                         |  |            |    Iteration number        |            |
> > |-----------------|:-----------------------:|:----------------:|:----------:|:----------:|:----------:|
> > |                 |                         |        10        |     300    |     600    |     900    |
> > | $\left(L^t_+\right)^2 / \left(L^t_\pm\right)^2$ |  Homogeneous (p = 0.9)  |   104.73591768   | 9.96024738 | 5.69375897 | 3.61622289 |
> > |  $\left(L^t_+\right)^2 / \left(L^t_\pm\right)^2$               | Heterogeneous (p = 0.0) |    18.83973209   | 2.31226298 | 1.69947195 |  1.3082626 |
> >
> > Looking at this table, we see that in the homogeneous case (first row), $\left(L^t_\pm\right)^2$ is about $10\times$ times  smaller than $\left(L^t_+\right)^2$ (on average), and is much larger at the start of the process than towards the end. On the other hand, in the heterogeneous case, the difference is smaller. The same dynamics applies here: it is much larger at the start than towards the end.
> >
> > The above table provides further numerical justification that $L_{\pm}^2$ can be effectively much smaller than $L_+^2$, and that this drives the improvement of our new PermK compressor.

---

> > ### Comment · Reviewer_wJjy · 2021-11-27
> > **Response**
> >
> > Thank the authors for addressing my confusion and extra efforts.
> >
> > *"Hessian variance can attain virtually any value"*:
> > yeah sure, I asked about the value of $L_\pm$, of course it is the lowest value (e.g., if a function is $L$-smooth, then it will also be $2L$-smooth, while we should care about the lowest $L$), also that is the value authors tried to estimate in the experiment mentioned in the response just now.
> >
> > *"it is not reasonable to expect that we could do this with Hessian variance"*:
> > I agree that the deriving the value of $L_\pm$ is data-dependent (as estimating the smoothness parameter $L$ or $L_+$ in classical nonconvex optimization) and not easy, so that's why I mentioned empirical estimation. But because both authors and I hope to see that $L_\pm$ is really smaller enough than $L_+$ (while generally we have no such requirements on $L$ or $L_+$), so I personally think that it is reasonable that we make some extra efforts on $L_\pm$ itself to compare with $L_+$. Also I mentioned Mei et al., (2021) work because it derived the closed-form expression of the nonuniform smoothness parameter in some RL and GLM examples, so I have the desire that your work can try to do something similar.
> >
> > *"Managed to run more computations"*: I appreciate that authors provided more details for the experiment from an empirical perspective. From a retrospective viewpoint, it suggests some interesting dynamics on $L_+/L_\pm$ over iterations, $L_\pm$ will be much smaller at the beginning, while possibly it may not be able keep holding till the end. Even it is preliminary results, but it truly reveal some evidence that $L_\pm$ will be much smaller in some cases at the beginning, which may motivate others to improve algorithms or analysis.
> >
> > With that, I will raise my evaluation. Maybe I would suggest that authors can consider to add the extra computation above in the next version. Thank you very much for the effort during the weekend :)

---

> > > ### Author Response · Authors · 2021-11-27
> > > **Thanks!!! (plus a few more clarifications)**
> > >
> > > Thanks for engaging with us, and allowing us explain our results. Much appreciated! And thanks for raising your score!
> > >
> > > There seems to be a bit of confusion in your latest response, and hence we would like to explain as this may influence your opinion (or the opinion of other reviewers) further.
> > >
> > > > "Hessian variance can attain virtually any value": yeah sure, I asked about the value of
> > > $L_{\pm},$ of course it is the lowest value (e.g., if a function is $L$-smooth, then it will also be
> > > $2L$-smooth, while we should care about the lowest $L$), also that is the value authors tried to estimate in the experiment mentioned in the response just now.
> > >
> > > Please note that our proposition "Hessian variance can attain virtually any value", which we proved in an earlier post, talks about the **exact value of Hessian variance.** That is, about the smallest value possible for which the inequality defining it is satisfied. **So, your above summary of our result is incorrect** - our result is more interesting that simply making an observation of the type "if a function is $L$-smooth, then it will also be
> > > $2L$-smooth, while we should care about the lowest $L$".
> > >
> > > **We would really want to make sure this message gets across.  The proposition says that the ratio $L_\pm^2/L_+^2$, with both of these constants chosen optimally, i.e., using their smallest values, can achieve any value in the interval $[0,1-\frac{1}{n}]$.**
> > >
> > > > Also I mentioned Mei et al., (2021) work because it derived the closed-form expression of the nonuniform smoothness parameter in some RL and GLM examples, so I have the desire that your work can try to do something similar.
> > >
> > > We did look at this paper. In Lemma 10 of the arXiv version of their paper they indeed obtain a closed form expression for their nonuniform smoothness parameter $\beta(\theta)$ (which is really a function of $\theta$). That expression depends on the loss at $\theta$ and the gradient of the loss at $\theta$.
> > >
> > > **The closest analogue to obtaining a similar closed form expression in our setting is what our Theorem 3 is about. However, the nature of $L_{\pm}^2$ happens to be such that Theorem 3 only gives a variational expression (unlike the function $\beta(\theta)$ in their work). This is because Hessian variance needs to satisfy a certain inequality for *all* $x$ and $y$. However, we can alternatively think of a related notion: Hessian variance seen as a function of $x$ and $y$. In that case, please note that our Theorem 3 gives the closed form expression you are looking for.** Indeed, please look at $L_{\pm}(x,y)$ defined in that theorem. Plus this expression is not valid for GLMs only; it is valid for all twice continuously differentiable functions.  This is what it is quite involved. Alternatively, we could define $L_{\pm}(x,y)$ as the ratio
> > > $$L_{\pm}^2(x,y) = \frac{\frac{1}{n} \sum_{i=1}^{n} || \nabla f_{i}(x)-\nabla f_{i}(y) ||^{2}-||\nabla f(x)-\nabla f(y)||^{2}}{||x-y||^{2}}.$$
> > > This is a closed form expression in the form of a function, in a similar way as $\beta(\theta)$ in Mei et al. (2021) is a closed form expression in the form of a function. If Mei et al. (2021) were interested in the largest value of
> > > $\beta(\theta)$, they would need to consider the problem $\sup_\theta \beta(\theta)$. In a similar way, we are interested in Hessian variance, given by $$L_{\pm}^2 = \sup_{x,y \in \mathbb{R}^d, x\neq y} L_{\pm}^2(x,y).$$
> > >
> > > **As you can see, we already obtained a result/expression similar to the one you wanted us to obtain. We believe this should settle your concern.**
> > >
> > > > "Managed to run more computations": I appreciate that authors provided more details for the experiment from an empirical perspective. From a retrospective viewpoint, it suggests some interesting dynamics on
> > > $L_+/L_\pm$ over iterations, $L_\pm$ will be much smaller at the beginning, while possibly it may not be able keep holding till the end. Even it is preliminary results, but it truly reveal some evidence that $L_\pm$ algorithms or analysis.
> > >
> > > **We are glad we were able to provide the empirical evidence you asked for!**
> > >
> > > Yes, something interesting is happening with the dynamics here, and our analysis does not really shed light on this. This is a similar situation to the phenomenon of stepsize scheduling in training: sometimes large stepsizes work fine, and sometimes one needs to decrease... The optimal choice depends on local geometry, which is hard to capture theoretically. Capturing the adaptive/dynamic behavior of $L^t_{\pm}$ is an interesting but hard problem to take in the future.
> > >
> > > > With that, I will raise my evaluation.
> > >
> > > **Thank you, much appreciated!**
> > >
> > > > Maybe I would suggest that authors can consider to add the extra computation above in the next version.
> > >
> > > **Yes, of course, we shall add this to the camera ready version of this paper should it be accepted. The extra experiment is useful.**
> > >
> > > > Thank you very much for the effort during the weekend :)
> > >
> > > **Thanks you as well for the exact same thing!**

---

### Official Review · Reviewer_mZxC · 2021-11-02

**Correctness:** 4
**Technical Novelty And Significance:** 3
**Empirical Novelty And Significance:** 3
**Recommendation:** 6
**Confidence:** 3

**Main Review:**

### Strength
1. The paper proposes some new techniques (e.g. AB inequality and input variance compressors) to relax the original assumption of independent compressors.
2. The paper proposes a new quantity called Hessian variance to refine the analysis of communication complexity.


### Concerns
1. The advantages of the new theoretical results are based on the regime Hessian variance is 0 or small. To illustrate, the paper gives some examples of Hessian variance equal 0, i.e. identical functions and linear perturbation. But these seem to be very simple cases. For more complicated models, it is not clear if and when the Hessian variance is small.
2. In the big data case ($d \leq n$), when $d \ll n$, there is no improvement.
3. In Figure 1, PermK does not outperform other algorithms, although the paper claims the theoretical improvement of $\sqrt{n}$. In Figure 2, the difference between PermK and TopK is very small, which does not show the theoretical improvements.

**Summary Of The Paper:**

This paper extends the theory of MARINA to support potentially correlated compressors and refines the original analysis of MARINA based on a new quantity called Hessian variance. They show their proposed compressors, random permutations, have improvements in theoretical communication complexity in the low Hessian variance regime.

**Summary Of The Review:**

Although not perfect and build upon existing works, the paper proposes some new techniques and quantity to refine the analysis of distributed non-convex optimization algorithm. The theoretical novelty is sufficient.

---

> ### Author Response · Authors · 2021-11-22
> **Addressing Concern #1**
>
> >The advantages of the new theoretical results are based on the regime Hessian variance is 0 or small. To illustrate, the paper gives some examples of Hessian variance equal 0, i.e. identical functions and linear perturbation. But these seem to be very simple cases. For more complicated models, it is not clear if and when the Hessian variance is small.
>
> *Response:*
>
> - Please note that **our theoretical results are an improvement to the previous results for MARINA by Gorbunov et al (2021) in all regimes**. The only regime in which we do not get a strict improvement is when the Hessian variance is maximal, i.e., when $L_{\pm}^2 = L_+^2$ (please look at Lemma 2 to see that this is an upper bound on Hessian variance), since in that case we get the same result. However, even in this case --- which, we stress, is the least favorable to us --- we compare against a result that is better than that provided in Gorbunov et al (2021). Please see footnote 2 on page 2 of our paper. Indeed, we observed (see the footnote) that their analysis can be modified slightly to obtain a stronger result, and this is the result we recover in the least favorable scenario to us.
>
> - We also wish to point out that the advantage of our approach gets stronger as the Hessian variance starts decreasing from its largest value, which is $L_{+}^2$ (see Lemma 2), to its lowest possible value, which is 0, and is **most pronounced when Hessian variance is zero**. We stress this extreme case for pedagogical reasons only as we believe it is most important for the reader to understand the best-case improvement obtainable by our refined method and analysis first, before appreciating that in real-life settings, the improvement will likely be somewhat subdued when compared to this ideal-case improvement. Please note that our experiments support this very clearly. Further, we give some illustrating examples of when Hessian variance is zero in Section 3 and summarize them in Table 2. Indeed, Hessian variance is zero in the two cases you mentioned, which corresponds to the first 2 lines in Table 2. However, please note that we point out these extreme cases for illustration only. If you look at the 3rd line of Table 2, you will see a formula for the Hessian variance of arbitrary quadratics, and in the 4th line, we give a formula for arbitrary twice continuously differentiable functions. Admittedly, it is not immediately obvious when Hessian variance will be small and when it will be large, but it is clear that it will be small in many cases and large in many cases as well. No method is best in all regimes, and we admit that our improvements get stronger as Hessian variance improves. However, **we always get some improvement on the analysis of Gorbunov et al (2021), except for the extreme case when the Hessian variance attains is maximum possible value**.
>
> - Please note we do not claim the Hessian variance is small or large. Clearly, mathematically speaking, it is possible to come up with situations when Hessian variance is anywhere on the spectrum from 0 to $L_{+}^2$. Our techniques always improve but improve much more in situations when Hessian variance is closer to 0. When it is zero, we get massive theoretical improvement factors: we mention them in the abstract.
>
> - We wish to point out that **the notion of Hessian variance is new**, and it may be of independent interest. We did not borrow it from any other paper; it appeared naturally through our attempts at improving the theory of the theoretical SOTA method, which is MARINA. Since this is a new notion, naturally it will take time to understand it well. we hope that our work will motivate other researchers to study this quantity further. For example, it will be interesting to study its relation to other notions of function or gradient similarity proposed in the literature. We wish to stress one key advantage of our notion to the more classical notion called "Hessian similarity" used in some recent works on Federated Learning (but unrelated to our setup or method). Our notion can be defined using first-order information only (see Definition 8). So, it is much less restrictive.
>
> We believe this concern is fully addressed above, and if not, we would be happy to respond to any further questions!
>
> Authors

---

> > ### Comment · Reviewer_mZxC · 2021-11-28
> > **Question about the reply to concern #1**
> >
> > Hi, I am not sure why the authors claim that "our theoretical results are an improvement to the previous results for MARINA by Gorbunov et al (2021) in all regimes.". In Table 4, in the big model case ($d \geq n$), when $L_{\pm}$ is large, it seems not easy to compare the PermK's $\frac{d}{n}L_- + \frac{d}{\sqrt{n}}L_{\pm}$ with RandK's $\frac{d}{\sqrt{n}}L_{+}$.
> >
> > Besides, I still think it is not clear if and when the Hessian variance is small for more complicated models. The paper will definitely be improved if can settle this problem. Otherwise, we will not know if the theory will be helpful in practice.

---

> > > ### Author Response · Authors · 2021-11-28
> > > **Answer to Question about the reply to concern #1**
> > >
> > > Thank you for your comment!
> > >
> > > > Hi, I am not sure why the authors claim that "our theoretical results are an improvement to the previous results for MARINA by Gorbunov et al (2021) in all regimes.". ...
> > >
> > > By Lemma 2, we know that $L_\pm \leq L_+$ and $L_- \leq L_+$. When $L_\pm$ is large, it means that $\frac{d}{n}L_- + \frac{d}{\sqrt{n}}L_\pm \leq \frac{d}{n}L_+ + \frac{d}{\sqrt{n}}L_+ \leq 2 \frac{d}{\sqrt{n}}L_+,$ thus the complexity bound of PermK is not worse than the complexity bound of RandK! Even if $L_\pm$ is large! But in case when $L_\pm$ is small, PermK is much better.
> > >
> > > Please, ignore all constants factors when you compare complexities because all complexities are provided in terms of "Big-O".
> > >
> > > > Besides, I still think it is not clear if and when the Hessian variance is small for more complicated models. The paper will definitely be improved if can settle this problem. Otherwise, we will not know if the theory will be helpful in practice.
> > >
> > > To the Reviewer wJjy we provided strong evidences that $L_\pm$ can be small, while $L_+$ is large.
> > >
> > >
> > > 1)
> > > We took the "mg" dataset from https://www.csie.ntu.edu.tw/~cjlin/libsvmtools/datasets/regression.html This dataset has $1,385$ datapoints and $d=6$ features.  We partitioned the dataset into $n=2$ parts (i.e., among 2 machines) randomly and consider the problem $\min_w \frac{1}{2} ||X_1 w - y_1||^2 + \frac{1}{2} ||X_2 w - y_2||^2 $. $(X_1, y_1)$ and $(X_2, y_2)$ refer to the split. In this case, we got $L_+ = 3609.6653$ while $L_\pm = 34.58$. Clearly, it is possible for $L_\pm$ to be much smaller than $L_+$.
> > >
> > > 2)
> > > We've managed to run more computations w.r.t. the experiment from Section 5.2. Instead of working with the notions $L_+^2$ and $L_{\pm}^2$ as defined in the paper, we work with their "adaptive" / "instantaneous" variants, which we call $\left(L^t_+\right)^2$ and $\left(L^t_{\pm}\right)^2$, which are easier to estimate. For example, $\left(L^t_{\pm}\right)^2$ is defined as the constant for which
> > >
> > > $$ \frac{1}{n}\sum_{i=1}^n || \nabla f_i(x^t) -  \nabla f_i(x^{t+1}) ||^2 - || \nabla f(x^t) - \nabla f(x^{t+1})||^2 = \left(L^t_{\pm}\right)^2 ||x^{t} - x^{t+1}||^2.$$
> > >
> > > We obtain this by choosing $x=x^t$ and $y=x^{t+1}$ in inequality (8) defining $L_\pm^2$.
> > >
> > > In particular, we estimate how the "instantaneous" ratio $\left(L^t_+\right)^2/\left(L^t_{\pm}\right)^2$ changes during the optimization process of MARINA with PermK on the autoencoder task (see Section 5.2 and Figure 2).
> > >
> > >  We have the plots showing the evolution for each iteration. However, since we can't show the plots here, we instead show a coarse summary of these plots in the form of a table:
> > >
> > >
> > > |               |                         |  |            |    Iteration number        |            |
> > > |-----------------|:-----------------------:|:----------------:|:----------:|:----------:|:----------:|
> > > |                 |                         |        10        |     300    |     600    |     900    |
> > > | $\left(L^t_+\right)^2 / \left(L^t_\pm\right)^2$ |  Homogeneous (p = 0.9)  |   104.73591768   | 9.96024738 | 5.69375897 | 3.61622289 |
> > > |  $\left(L^t_+\right)^2 / \left(L^t_\pm\right)^2$               | Heterogeneous (p = 0.0) |    18.83973209   | 2.31226298 | 1.69947195 |  1.3082626 |
> > >
> > > Looking at this table, we see that in the homogeneous case (first row), $\left(L^t_\pm\right)^2$ is about $10\times$ times  smaller than $\left(L^t_+\right)^2$ (on average), and is much larger at the start of the process than towards the end. On the other hand, in the heterogeneous case, the difference is smaller. The same dynamics applies here: it is much larger at the start than towards the end.
> > >
> > > The above table provides further numerical justification that $L_{\pm}^2$ can be effectively much smaller than $L_+^2$, and that this drives the improvement of our new PermK compressor.

---

> ### Author Response · Authors · 2021-11-22
> **Addressing Concern #2**
>
> >In the big data case ($d\leq n$), when $d\ll n$, there is no improvement.
>
> *Response:*
>
> This is factually incorrect.
>
> - We have strict improvement when $n \leq d^2$. However, when $n > d^2$, we get no theoretical improvement.
>
> - Note that modern ML models are typically heavily over-parameterized, which means that $d$ is typically much larger than the number of training examples. Moreover, in our notation, $n$ is the number of machines each of which owns many data examples, and hence $n$ is much smaller (by orders of magnitude) than the number of training examples. So, in a typical situation when our method would be applied, $d$ would be much larger than $n$. Moreover, communication compression makes the most sense when $d$ is very large.
>
> - Please note that no method can possibly be better than previous theoretical SOTA in all regimes. So, simply pointing out that that there exists a regime when a method is not better than some other methods should *not* be seen as an issue with a paper. In fact, when such a claim of absolute superiority across all regimes is made in some paper, the theory of that paper is almost surely wrong.
>
> In summary, we believe we fully addressed this concern, which simply seems to be just a misunderstanding.   We are happy to provide any further clarifications!

---

> ### Author Response · Authors · 2021-11-22
> **Addressing Concern #3**
>
> > In Figure 1, PermK does not outperform other algorithms, although the paper claims the theoretical improvement of $\sqrt{n}$. In Figure 2, the difference between PermK and TopK is very small, which does not show the theoretical improvements.
>
> *Response:*
>
> This is factually incorrect.
>
> - First, note that in Figure 1 we vary Hessian variance across the columns: it is 0 in the first column and grows in subsequent columns. Qualitatively, our theory predicts that MARINA with PermK (our contribution) will outperform MARINA with RandK (the result of Gorbunov et al) in all regimes, except when the Hessian variance is maximal (in which case we recover their result; so we are never worse), and that the difference will be more pronounced when Hessian variance is small. Please note that this is exactly what the experiments in Figure 1 show. We get the biggest advantage in the columns on the left, and the advantage is **1-2 orders of magnitude**  in the 2nd-3rd rows (when $n=d$ or $n=10d$) in the first four columns. This is a **massive improvement which corresponds to our theoretical predictions from Table 4**,  as we explain in the commentary in Section 5.1.
>
> - Please note that **whenever we apply the TopK compressor (which is biased, and hence not compatible with MARINA), we use the current theoretical SOTA error feedback method called EF21.** Error feedback is necessary when biased compressors are used as otherwise one may experience divergence (Seide et al 2014, Beznosikov et al 2020). We include comparisons to EF21 because biased compressors often lead to very good performance in practice, and we thought such a comparison might be interesting. However, such a comparison has nothing to do with our theoretical results. Please note that the theoretical communication complexity of EF21 is **the same as that of gradient descent** (see Table 4). That is, **EF21 does not benefit from compression in theory! On the other hand, MARINA with RandK does improve on GD in theory, and MARINA with our PermK compressor is even better, as we show in our paper.**
>
> - So, when you say, for example, that "the difference between PermK and TopK is very small, which does not show the theoretical improvements", you are making a wrong conclusion. **We prove that MARINA with PermK is better than MARINA with RandK, and all our experiments support that. However, biased compressors are currently massively misunderstood in theory, as even the best theory (EF21) is very conservative. This is not an issue with our paper, but an issue with the current understanding of biased compressors and error feedback.** Let us shed some light on this. EF21 with TopK suffers from the following drawbacks: a) it is only understood that its communication complexity is not worse than gradient descent (in theory), b) it does NOT improve with $n$.
>
> In conclusion, **we believe that this third criticism is also based on a misunderstanding by the reviewer, and we hope our explanation clarifies this.** If any further feedback is needed, please do ask!

---

> > ### Comment · Reviewer_mZxC · 2021-11-25
> > **Question about concern #3**
> >
> > Thanks for the author's reply. Sorry, I did not make it clear.
> >
> > In Figure 1 first row, the performance of the three algorithms seems similar. If I understand correctly, this is the "Big model case" you discussed in Section 4.1. In this case, you claim a theoretical improvement of at least $\sqrt{n}$ (when $L_{\pm} = 0$).

---

> > > ### Author Response · Authors · 2021-11-25
> > > **Response to Question about Concern #3**
> > >
> > > Indeed, in the case when $L_\pm = 0$ and $n \leq d$, we have a theoretical improvement factor of $\sqrt{n}$. The first row represents the case when $n = 10$, so $\sqrt{n} = \sqrt{10} \approx 3.16$. It is not easy to capture (= observe in experiments) the theoretical factor of $3.16$ in some practical optimization tasks for many reasons: i) we could have chosen a slightly suboptimal learning rate, which may erase these gains (the theoretical learning rate requires the estimation of some parameters and this estimation can easily be at least a bit inaccurate), ii) the optimization algorithms considered are nondeterministic and the noise has some (small, but nevertheless present) influence on the plots, iii) the improvement is in worst case *upper bound* on the behaviour of the method, and naturally prsctical runs may not behave in this worst case manner.
> > >
> > > In other words, when $n$ is small, then indeed, the improvement we would expect to observe in practice would be small. But for that reason exactly we also perform distributed optimization experiments where the number of nodes $n$ is big ($n = 1000$). The only change between the first and the second row is $n$. Relative to RandK, we see that the performance of PermK increases with $n$. We added the first line with $n = 10$ for pedagogical reasons to show that the method improves with $n$, which is a particular qualitative prediction our theory makes which we confirm in this experiment.
> > >
> > > If you need any other clarification, we are ready to provide them! Thank you for engaging with us!

---

### Author Response · Authors · 2021-11-22
**To all reviewers**

Dear reviewers,

Thank you for your evaluations!

**We believe we addressed all concerns, and we believe all were minor as they come from misunderstandings which we clarified.**  **However, there is a significant mismatch between the quality of our results (theoretical SOTA for an important problem) and the evaluations and scores we have obtained.**

We are convinced this mismatch is due to some basic and often even elementary misunderstandings which we clear up in our rebuttal. We would be most happy if you could read our response to all authors to see that, indeed, we managed to answer all concerns satisfactorily.

**If any further questions or issues arise, we will be most happy to respond promptly!**

**Thank you very much for the time and effort you spent reading and evaluating our work! We truly appreciate this.**

Authors

---

### Author Response · Authors · 2021-11-27
**To all reviewers: please engage with us, discussion period ends in 2 days**

Dear reviewers,

Thanks again for your reviews. We would be most happy if you could engage with our responses to your reviews. You have raised some concerns, and we believe we addressed them. If anything is not addressed properly in your view, please let us know. If all is addressed to your liking, also let us know! We are at your disposal - ready to clarify any points that remain unanswered in full.

Thank you!

Authors

---

### Author Response · Authors · 2021-12-01
**Dear reviewers: we addressed all your cocerns**

Dear reviewers,

Thanks again for your reviews!

**Our work offers a new SOTA in theoretical communication complexity for distributed optimization/training in the smooth nonconvex regime, which is a key problem in modern machine learning (neural network training always leads to nonconvex problems, and for most activation functions, the problem is also smooth).**

We believe that such work should command substantially higher scores. We addressed all concerns in a lot of detail. However, we have got minimal engagement. Thanks to the reviewers who responded to us, and to the reviewer who increased their score.

**If all issues are addressed to your liking, and we believe we addressed them all, please let us know! In such a case, please do consider increasing your score appropriately.** We do not think any issues remained. We are at your disposal - ready to clarify any points that remain unanswered in full.

Thank you!!!

Authors

---

### Decision · Program_Chairs · 2022-01-20

**Decision:**

Accept (Poster)

**Comment:**

The reviewers have agreed that the paper is in borderline. Although the reviewers are not really convinced about the authors’ responses, they still acknowledge that the paper is interesting and developed some new techniques for the analysis of distributed optimization.

The following concerns are raised by the reviewers from their discussions:

1) The paper is heavily based on existing work.
2) The theoretical advantages are based on the regime Hessian variance is 0 or small, but it is not clear if and when the Hessian variance is small for more complicated models, which means we will not know if the theory will be helpful in practice. Although the authors provide some experimental results in the rebuttal showing that $(L_+)^2 / (L_{\pm})^2$ can be large at initial iterations, it is still not clear how long will this advantage keep during training and how much the advantage is.
3) Reviewer wjjy increased the score from 5 to 6, considering that the additional result truly suggests that the implicit setting can hold in some case at the beginning of iteration, which makes the submission a complete story for him/her now, to some extent. But if we are more strict on the evaluation, the experiment result also suggests that the implicit assumption will not hold anymore over iterations, because the ratio is approaching 1 quickly, i.e., $ L_\pm$ is about the same order as $ L_+$, so there is a mismatch between theory and practice, which even brings out the risk that the paper will fail from the beginning because Sec 4.1 will not make sense anymore.
4) On the theory side, two main contributions of this paper is relaxation on the compressors used in MARINA and a new assumption to refine the analysis. These two contributions seem rather limited if only used to analyze this specific algorithm -- it's unclear what the authors mean in practice or how they correlated to the MARINA. For example, it is still hard for us to compare or understand MARINA with another algorithm as we wouldn't know if the improvement from MARINA is due to a better design, or this additional assumption. The reviewer also finds the authors statement that "their analysis focuses on MARINA because it is SOTA" confusing. Different from NLP and vision community where standard benchmarks are usually used to evaluate new models, He/she is confused by what it means for a newly proposed optimization algorithm to be SOTA.
5) Since the paper proposes a specific algorithm named PermK, it's quite reasonable to question how it relates to some previously proposed sparsification methods with similar design such as (Jianqiao et al., 2017). However, the authors insist their main contribution is in theory, and the small-scale experiments comparing with TopK and RandK are sufficient. The reviewer disagrees about this. As communication compression is usually need in larger scales (at least beyond MNIST), and TopK/RandK are not SOTA baselines of sparsification.

The authors are expected to address them for the clarifications in the final version.